# Sweepstakes reproductive success via pervasive and recurrent selective sweeps

Einar Árnason[1,2]*, Jere Koskela[3], Katrín Halldórsdóttir[1], Bjarki Eldon[4]

[1]Institute of Life- and environmental Sciences, University of Iceland, Reykjavik, Iceland; [2]Department of Organismal and Evolutionary Biology, Harvard University, Cambridge, United States; [3]Department of Statistics, University of Warwick, Coventry, United Kingdom; [4]Leibniz Institute for Evolution and Biodiversity Science, Museum für Naturkunde, Berlin, Germany

**Abstract** Highly fecund natural populations characterized by high early mortality abound, yet our knowledge about their recruitment dynamics is somewhat rudimentary. This knowledge gap has implications for our understanding of genetic variation, population connectivity, local adaptation, and the resilience of highly fecund populations. The concept of sweepstakes reproductive success, which posits a considerable variance and skew in individual reproductive output, is key to understanding the distribution of individual reproductive success. However, it still needs to be determined whether highly fecund organisms reproduce through sweepstakes and, if they do, the relative roles of neutral and selective sweepstakes. Here, we use coalescent-based statistical analysis of population genomic data to show that selective sweepstakes likely explain recruitment dynamics in the highly fecund Atlantic cod. We show that the Kingman coalescent (modelling no sweepstakes) and the Xi-Beta coalescent (modelling random sweepstakes), including complex demography and background selection, do not provide an adequate fit for the data. The Durrett–Schweinsberg coalescent, in which selective sweepstakes result from recurrent and pervasive selective sweeps of new mutations, offers greater explanatory power. Our results show that models of sweepstakes reproduction and multiple-merger coalescents are relevant and necessary for understanding genetic diversity in highly fecund natural populations. These findings have fundamental implications for understanding the recruitment variation of fish stocks and general evolutionary genomics of high-fecundity organisms.

*For correspondence:
einararn@hi.is

**Competing interest:** The authors declare that no competing interests exist.

## Editor's evaluation

This fundamental work significantly advances our understanding of genetic diversity in highly fecund organisms by showing that the Atlantic cod genome is prone to recurrent selective sweeps. The evidence supporting these conclusions is compelling, with rigorous analysis of the site frequency spectrum providing support for models of selective sweepstakes reproduction and multi-merger coalescents. This work will be of broad interest to evolutionary geneticists and should stimulate future work to determine whether random sweepstakes interact with selective sweepstakes.

## Introduction

Individual reproductive success, the number of reproducing offspring, is a fundamental demographic parameter in ecology and evolution. The distribution of individual reproductive success affects the distribution and abundance of organisms (the subject of ecology) and the genotypic and phenotypic changes resulting from the major processes of evolution. Individual reproductive success determines individual fitness, the currency of natural selection. Many marine organisms are highly fecund,

producing vast numbers of juvenile offspring that experience high mortality (type III survivorship) as they go through several developmental stages, fertilization, zygote, larvae, fry, etc. until finally recruiting as adults into the next generation. Sweepstakes reproductive success (*Hedgecock, 1994*), suggested having 'a major role in shaping marine biodiversity' (*Hedgecock and Pudovkin, 2011*, p. 971), is a key to understanding the mechanism behind individual reproductive success. Sweepstakes reproduction has few winners and many losers leading to very high variance and skew in individual reproductive output. High fecundity alone does not lead to sweepstakes absent a mechanism for generating high variance in and highly skewed distribution of offspring numbers.

Two main ecological mechanisms can turn high fecundity into sweepstakes reproduction that produces 'reproductive skew': a random and a selective mechanism. The first is the chance matching of reproduction to a jackpot of temporally and spatially favourable conditions, a case of random sweepstakes (*Hedgecock and Pudovkin, 2011*). The match/mismatch hypothesis (*Cushing, 1969*) often explains the dynamics of recruitment variation and year-class strength by the timing of reproduction with favourable but erratic environmental conditions. For example, climatic variability leads to random temporal and spatial shifts in planktonic blooms that are food for developing fish larvae, a match means reproductive success, a mismatch means reproductive failure (*Cushing, 1969*). By chance, a random individual hits the jackpot of favorable environmental conditions that result in a very large reproductive output of reproducing offspring (*Schweinsberg, 2003*; *Eldon and Wakeley, 2006*).

The second mechanism is selective sweepstakes in which the genetic constitution of survivors differs from that of non-survivors (*Williams, 1975*). Under the second scenario, an organism's developmental stages pass through numerous independently acting selective filters with the cumulative effect of producing a high-variance high-skew offspring distribution. Here, the winning genotypes are ephemeral and must be continuously reassembled; they are the Sisyphean genotypes in a race that *Williams, 1975* argued could pay the selective cost of sexual reproduction (after Sisyphus from Greek mythology, punished with forever pushing a boulder up a hill). By analogy, the population climbs a local selective peak by positive selection, but the environment changes continuously because the sequence of selective filters changes. Only a new or recombined genotype can climb the selective peak the next time around (*Williams, 1975*). The population forever tracks an elusive optimum by climbing an ephemeral adaptive peak. The selective filters can arise from abiotic factors, and biotic density- and frequency-dependent effects arising from inter- and intraspecific competition and from predation and predator avoidance (*Reznick, 2016*).

The prevailing view in evolutionary ecology is that highly fecund populations evolve without sweepstakes reproduction. Random mortality is seen as hitting every family, the offspring of every pair, to the same degree. High fecundity simply compensates for high mortality and there is no mechanism for turning high fecundity into high-variance high-skew offspring distribution. Juvenile mortality might even be compensatory and reduce the variance in offspring number via density-dependent competition or predation. In this scenario reproduction does not match favourable conditions by chance, no individual hits the jackpot, nor does selective filtering happen. The resulting offspring distribution has a much smaller variance than in the sweepstakes models, with the same low and unchanged coefficient of variation in the distribution of zygotes and the distribution of adult offspring (*Nunney, 1996*). This mode of reproduction is expected to result in a similar distribution of reproducing offspring as in the assumed mode of reproduction of low fecundity and model organisms (*Wright, 1931*; *Fisher, 1930*). A low variance in individual reproductive success modelled through the Wright–Fisher model (or similar models) is nearly universally assumed in population genetics (*Wakeley, 2007*). This is the hypothesis of no sweepstakes.

Genomics and coalescent theory offer powerful tools to test three hypotheses: non-sweepstakes versus sweepstakes reproduction and two sweepstakes hypotheses, random and selective sweepstakes. Conducting similar tests with ecological methods would be a daunting task, requiring one to follow the fate of the offspring of different individuals (*Grant and Grant, 2014*). The first hypothesis can be tested by identifying the footprint of non-sweepstakes versus sweepstakes reproduction in population genomic data. The second and third hypothesis tests for random versus the selective sweepstakes, given evidence of sweepstakes reproduction in the data.

The classical Kingman coalescent (*Kingman, 1982*) models the reproduction of low-fecundity organisms (Appendices 1 and 2). Multiple-merger coalescents (*Donnelly and Kurtz, 1999*; *Pitman,*

*1999*; *Sagitov, 1999*; *Schweinsberg, 2000*; *Schweinsberg, 2003*) describe the genealogies for the random and the selective sweepstakes reproduction. The Xi-Beta coalescent (*Schweinsberg, 2000*; *Birkner et al., 2018*) models the genealogy of a population with large reproductive events in which a random individual has enormous reproductive success and well approximates the random or jackpot sweepstakes hypothesis (*Hedgecock and Pudovkin, 2011*; see Appendix 3). The Durrett–Schweinsberg model of recurrent selective sweeps (*Durrett and Schweinsberg, 2005*), implying an ever-changing environment that continuously favors new mutations, well approximates selective sweepstakes (*Williams, 1975*) (see Appendix 4). The multiple-merger Durrett–Schweinsberg coalescent describes the genealogy of a neutral site linked to a site hit by a favorable mutation that rapidly sweeps to fixation. The neutral site can escape via recombination (see Appendix 5).

The empirical evidence for reproductive sweepstakes leading to reproductive skew is limited (e.g. *Árnason, 2004*; *Árnason and Halldórsdóttir, 2015*; *Niwa et al., 2016*). Empirical evidence for variance in reproductive success due to life-table characteristics has been found using genome-wide polymorphism data in marine fishes (*Barry et al., 2021*). Reproductive skew needs to be studied using gene genealogies on a genome-wide scale. Multiple-merger coalescents occur in models of rapidly adapting populations (*Neher and Hallatschek, 2013*; *Schweinsberg, 2017*), under both directional selection (*Neher, 2013*; *Sackman et al., 2019*) and possibly strong purifying (background) selection (*Irwin et al., 2016*; *Cvijović et al., 2018*). However, background selection is not, in general, expected to mimic selective sweeps (e.g. *Durrett and Schweinsberg, 2005*; *Schrider, 2020*). Sweepstakes reproduction may apply to many organisms and may be more prevalent than previously thought. There is, therefore, a need for a critical examination of the contrasting hypotheses.

Here, we compare genomic sequences for the highly fecund Atlantic cod (*Gadus morhua*) to predictions of three coalescent models: the *Kingman, 1982* with arbitrary demographic histories, the neutral Xi-Beta or formally the $\Xi$-Beta$(2 - \alpha, \alpha)$ coalescent (*Schweinsberg, 2000*; *Schweinsberg, 2003*; *Birkner et al., 2018*) modelling random jackpot sweepstakes in diploid, highly fecund organisms, and the Durrett–Schweinsberg coalescent derived from a population model with recurrent selective sweeps (*Durrett and Schweinsberg, 2005*; Appendix 1). We analyze whole-genome sequences (at 16× and 12× coverage, respectively) of Atlantic cod sampled from two localities in Iceland, with the localities serving as statistical replicates (*Appendix 6—figure 1*). We also consider whether other mechanisms can explain the observed patterns by examining the effects of population expansion, cryptic population structure, balancing and background selection, and the joint action of several processes.

## Results

### Neutrality under no sweepstakes?

Genomic scans of Tajima's $D$ and Fay and Wu's $H$ neutrality test statistics (for GL1 and GL2 genotype likelihoods in both populations; *Figure 1a*, *Figure 1—figure supplement 1*, and *Appendix 6—figure 2a, b*, and *Appendix 7—table 1* and *Appendix 7—table 2*) showed extensive and genome-wide deviations from expectations of neutral equilibrium under the classical theory, including indications consistent with selective sweeps occurring throughout the genome (*Fay and Wu, 2000*; *Zeng et al., 2006*; *Przeworski, 2002*; see Appendix 8). The McDonald–Kreitman test (*McDonald and Kreitman, 1991*) and the neutrality index derived from it (*Rand and Kann, 1996*), also indicated positive selection. The neutrality index $NI = 1$ under neutrality, negative values of $-\log(NI)$ indicate negative or purifying selection, and positive values indicate positive selection. Our estimates showed both negative and positive selection effects distributed throughout each chromosome (*Figure 1—figure supplement 2*). On a local genomic scale, the distribution of the neutrality index was heavier on the side of positive selection, although only a minority of individual tests reached nominal significance and none was significant after taking multiple testing into account (*Figure 1b*). On a genome-wide scale, the mean and the median of $-\log NI$ were 0.27 and 0.21, respectively, and the estimated proportion of adaptive non-synonymous substitutions $\alpha = 1 - NI$ (*Smith and Eyre-Walker, 2002*) was 19–24%.

The classic no-sweepstakes model with population growth (such as post-Pleistocene population expansion, *Hewitt, 2004*) is known to affect primarily the singleton class and left tail of the site-frequency spectrum. Fitting a no-sweepstakes with population growth model to the data under the Kingman coalescent provided indication of historical expansion (see *Figure 2* and *Figure 2—figure*

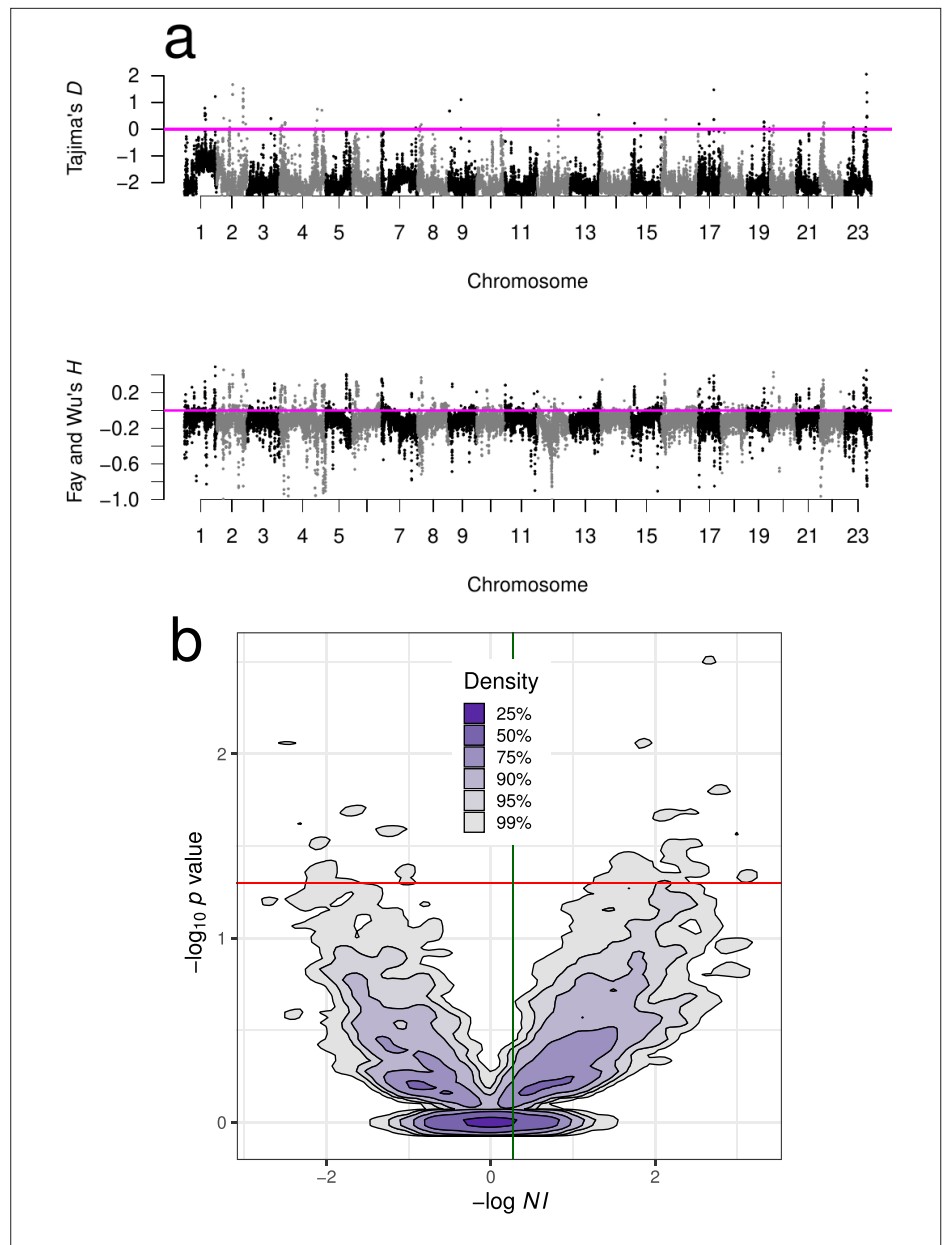

**Figure 1.** Neutrality test statistics and distribution of the neutrality index. (**a**) Manhattan plots of Tajima's D (**Tajima, 1989**) and Fay and Wu's H (**Fay and Wu, 2000**) showed mostly negative values at all chromosomes implying deviations from neutrality. Sliding window estimates (window size 100 kb with 20 kb step size) using GL1 genotype likelihoods for the South/south-east population. Value of the statistic under Kingman coalescent neutrality equilibrium indicated with magenta horizontal line. (**b**) Kernel density contours (**Duong, 2022**) of the $-\log_{10} p$ value significance of Fisher's exact test associated with the McDonald–Kreitman test (**McDonald and Kreitman, 1991**) plotted against the neutrality index (**Rand and Kann, 1996**) $-\log NI$. $NI = (P_n/P_s)/(D_n/D_s)$ where $P_n$, $P_s$, $D_n$, and $D_s$ are the number of non-synonymous and synonymous polymorphic and fixed sites, respectively, for all genes of each chromosome. Negative values of $-\log NI$ imply purifying (negative) and background selection and positive values imply positive selection (selective sweeps). The outgroup is Pacific cod (Gma). Overall, the cloud of positive values is denser than the cloud of negative values. The red horizontal line is at nominal significance level of 0.05 for individual tests; no test reached the $0.05/n$ Bonferroni adjustment for multiple testing. The mean (green vertical line) and the median of $-\log NI$ were 0.27 and 0.21, respectively, and imply that the proportion of adaptive non-synonymous substitutions $\alpha = 1 - NI$ (**Smith and Eyre-Walker, 2002**) is 19–24%. **Figure 1—figure supplement 1** shows neutrality statistics for the Þistilfjörður population. **Figure 1—figure supplement 2** shows distribution and violin plot of $-\log NI$ across each chromosome from the South/south-east population.

*Figure 1 continued on next page*

*Figure 1 continued*

The online version of this article includes the following figure supplement(s) for figure 1:

**Figure supplement 1.** Neutrality tests for Þistilfjörður population.

**Figure supplement 2.** Neutrality Index and violin plot of neutrality index across chromosomes.

*supplement 1*). However, the plausible demographic growth scenarios did not markedly improve the fit of neutral models without sweepstakes. We, therefore, reject the no-sweepstakes hypothesis.

## Random versus selective sweepstakes?

The life table of cod (Appendix 9 and *Appendix 7—table 3*), showing an exponential decay of the number of survivors with age and an exponential increase in fecundity with age, implies that fewer and fewer individuals produce a larger and larger number of eggs. A few females may live 25 years or more and still increase fecundity with age. Thus, the life-table results in a large variance and skew in offspring number. Old surviving females may be the lucky few to be alive or they may be the very fit that have passed all selective filters.

We next compared our observations to predictions of the $\Xi$-Beta $(2 - \alpha, \alpha)$ coalescent, which models random jackpot sweepstakes reproduction in a diploid highly fecund population. Here, the parameter $\alpha \in (1, 2)$ determines the skewness of the offspring distribution, in essence, the jackpot size. A smaller $\alpha$ essentially means a larger jackpot. We used a range of approximate Bayesian computation (ABC) posterior estimates of the $\alpha$ parameter (Appendix 3). The observed site-frequency spectra were overall more V-shaped than the U-shape of the expected normalized site-frequency spectrum predicted by this model (Appendix 3; *Appendix 6—figure 3a, b*). Singletons and low-frequency variants were the closest to expectations of an $\alpha = 1.35$ (*Appendix 6—figure 3*). However, as the derived allele frequency increases, the observations were closer to a smaller and smaller $\alpha$ (as small as $\alpha = 1.0$) predictions. The expected site-frequency spectrum of this model shows local peaks at intermediate allele frequencies, which represent the expected simultaneous multiple mergers of two, three, and four groups, corresponding to the four parental chromosomes involved in each large reproduction event. In diploid highly fecund populations, a single pair of diploid parents may occasionally produce huge numbers of juvenile offspring (*Möhle and Sagitov, 2003*; *Birkner et al., 2013a*; *Birkner et al., 2018*). The observations did not show these peaks (*Appendix 6—figure 3a, b*). The expectations of this model were also mainly outside the bootstrap error bars of the observations (*Figure 3*). However, comparing the observed site-frequency spectra to expectations of the haploid $\Lambda$-Beta$(2 - \alpha, \alpha)$ coalescent, a haploid version of random sweepstakes (*Appendix 6—figure 4*), showed a better fit. Low-frequency variants fit reasonably well to an $\alpha = 1.35$. However, as the derived allele frequency increased, a smaller and smaller $\alpha$ (as small as $\alpha = 1.0$, the Bolthausen–Sznitman coalescent) gave a good fit. This is likely a signal of either positive or negative natural selection. Rare alleles (less than 10–12%) contribute little to the variance in fitness. Once an allele (a site) reaches an appreciable and intermediate frequency it can contribute significantly to the variance in fitness such that selection quickly moves it out of intermediate frequency ranges. Negative selection moves it to a low frequency, and positive selection moves it to a high frequency so alleles spend a short time at intermediate frequencies (sojourn times are short). The fact that a haploid $\Lambda$-coalescent model fits a diploid organism better than the corresponding diploid $\Xi$-coalescent is suggestive of natural selection, where fitter offspring descend from one particular parental chromosome out of the available four. The parameter $\alpha$ determines the skewness of the offspring distribution in the $\Lambda$-Beta-coalescent. But that model has no known interpretation for an explicitly selection-driven skewness (except in the particular case $\alpha = 1$, the Bolthausen–Sznitman coalescent (*Neher, 2013*; *Neher and Hallatschek, 2013*), which did not adequately fit our data). Hence, the $\Lambda$-Beta-coalescent is not an appropriate model for a diploid organism and remains difficult to intepret. Furthermore, we used ABC to estimate jointly the parameters $\alpha$ and $\beta$, where $\beta$ denotes a population size rescaled rate of exponential growth of the population forward in time, using the $\Xi$-Beta$(2 - \alpha, \alpha)$ coalescent (Appendix 3). The processes generating reproductive skew and population growth, can account for some features of the site-frequency spectrum. Thus, by jointly estimating the skew parameter $\alpha$ and the growth parameter $\beta$ we hope to obtain a more accurate understanding of the observed data. The resulting posterior distribution showed small values of both parameters (*Figure 2* and *Figure 2—figure supplement 1*)

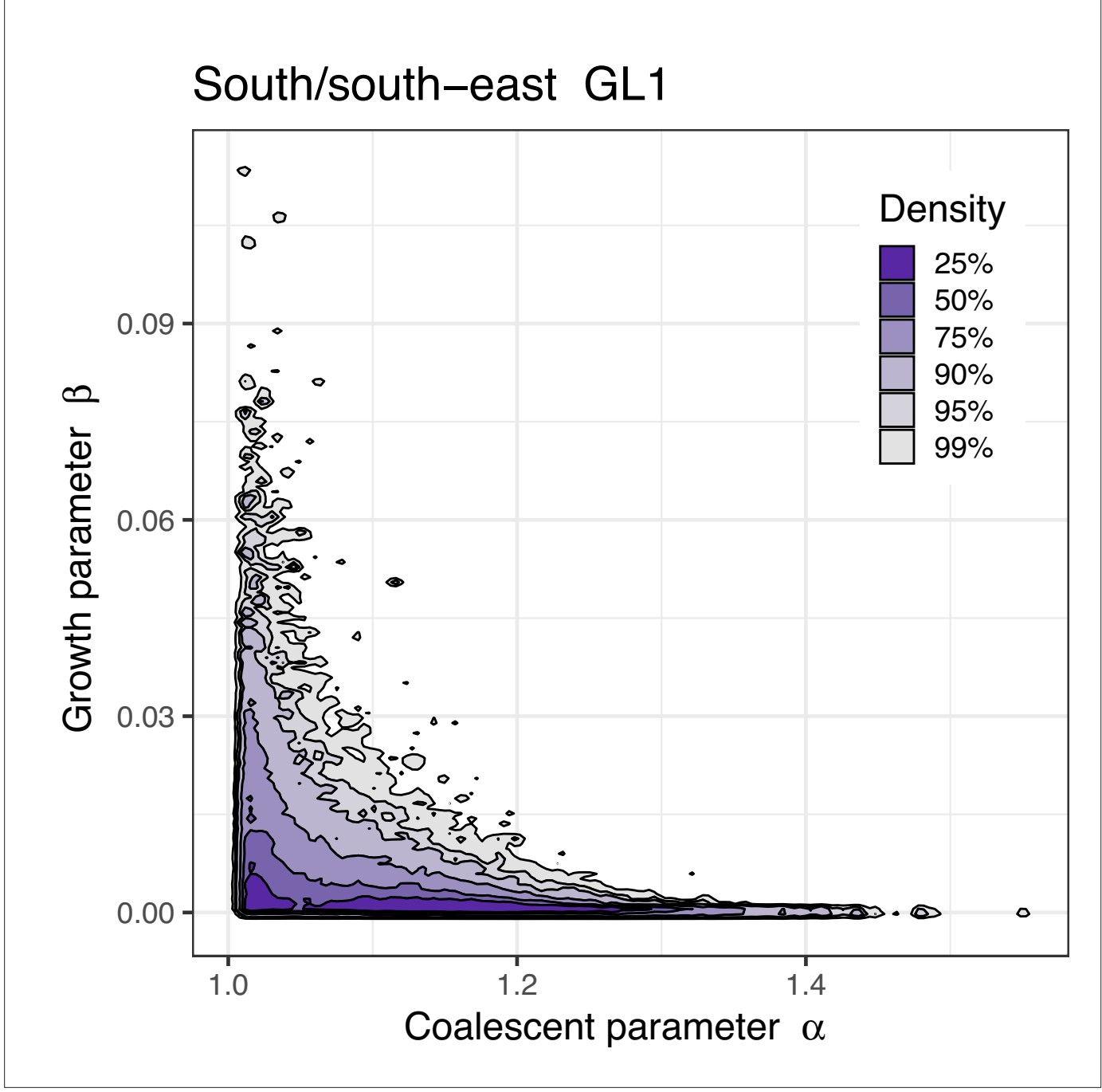

**Figure 2.** Approximate Bayesian computation (ABC) joint estimation of parameters of the neutral $\Xi$-Beta$(2 - \alpha, \alpha)$ coalescent (random sweepstakes) and of population growth. A kernel density estimator (**Duong, 2022**) for the joint ABC-posterior density of $(\alpha, \beta) \in \Theta_\mathbf{B}$. The parameter $\alpha$ determines the skewness of the offspring distribution in the neutral $\Xi$-Beta$(2 - \alpha, \alpha)$ coalescent model, and the parameter $\beta$ is a population-size rescaled rate of exponential population growth. Estimates using GL1 for the South/south-east population. A bivariate model-fitting analysis adding exponential population growth to the $\Xi$-Beta$(2 - \alpha, \alpha)$ coalescent does not improve model fit for random sweepstakes. The population growth parameter ($\beta$) only has an effect under maximal sweepstakes (low values of $\alpha$). **Figure 2—figure supplement 1** explores the random sweepstakes model with population growth using both GL1 and GL2 likelihood estimates of site-frequency spectra for both the South/south-east and Þistilfjörður populations, and for different ranges of parameter values.

The online version of this article includes the following figure supplement(s) for figure 2:

**Figure supplement 1.** Joint estimate of growth and coalescent parameter for other situations.

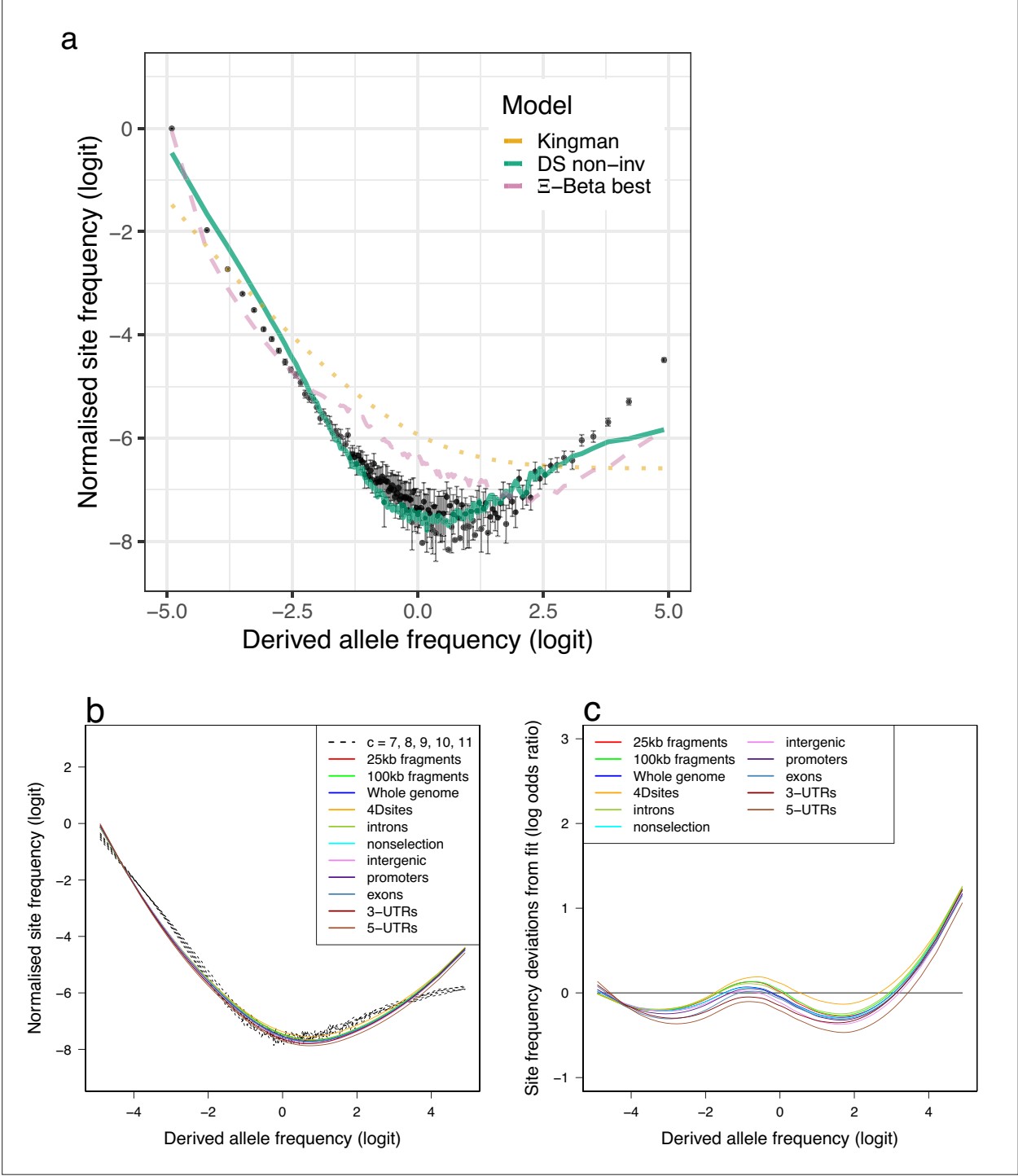

**Figure 3.** Fit of observations to models: the no-sweepstakes model, the random sweepstakes model, and the selective sweepstakes model. (**a**) Mean observed site-frequency spectra for the 19 non-inversion chromosomes combined estimated with GL1 likelihood for the South/south-east populations (sample size $n = 68$). Error bars of observed data (dots) are ±2 standard deviations of the bootstrap distribution with 100 bootstrap replicates. Expected site-frequency spectra are the Kingman coalescent modelling no sweepstakes, the best approximate maximum likelihood estimates (**Eldon et al., 2015**) of the Ξ-Beta coalescent modelling random sweepstakes, and the approximate Bayesian computation (ABC) estimated Durrett–Schweinsberg coalescent (DS) modelling selective sweepstakes. (**b**) The observed site-frequency spectra of different sized fragments and various functional classes compared to expectations of the Durrett–Schweinsberg coalescent (DS) ABC estimated for the non-inversion chromosomes for the South/south-east population. The compound parameter $c$ ranges from 5 to 11. Fragment sizes of 25 and 100 kb. The different functional groups are fourfold degenerate sites (4Dsites), intronic sites, non-selection sites (sites more than 500 kb away from peaks of selection scan, **Appendix 6—figure 8**), intergenic sites,

*Figure 3 continued on next page*

*Figure 3 continued*

promoters, exons, $3'$-UTR sites (3-UTRs), and $5'$-UTR sites (5-UTRs) in order of selective constraints. (**c**) Deviations from expectations of the Durrett–Schweinsberg model of recurrent selective sweeps of different sized fragments and functional groups for the South/south-east population. ***Figure 3—figure supplement 1*** shows comparable results for the Þistilfjörður population. ***Figure 3—figure supplement 1*** shows site-frequency spectrum polarized with 100% consensus of walleye pollock (Gch), Pacific cod (Gma), and Arctic cod (Bsa) to minimize potential effects of SNP misorientation and low-level ancestral introgression (Appendix 10). ***Figure 3—figure supplement 4*** shows site-frequency spectrum for transversions only removing transition sites that are more likely to be at mutation saturation to adddress potential SNP misorientation. ***Figure 3—figure supplement 4*** shows site-frequency spectrum truncated by removing singletons and doubletons and the $n-1$ and $n-2$ classes that are most sensitive to SNP misorientation and low-level ancestral introgression.

The online version of this article includes the following figure supplement(s) for figure 3:

**Figure supplement 1.** Site-frequency spectra and model fit for the replicate Þistilfjörður population.

**Figure supplement 2.** Site-frequency spectra polarized using a 100% consensus of three outgroup taxa.

**Figure supplement 3.** Site-frequency spectra of transversions excluding transitions.

**Figure supplement 4.** Site-frequency spectra excluding singletons and doubletons.

implying strong reproductive skew and little population growth. That the distribution of the growth parameter spread more with greater reproductive skew (as $\alpha \to 1$) is not surprising, as population size is known to affect the model only weakly when the reproductive skew is pronounced. Furthermore, the impact of variable population size vanishes entirely when reproductive skew is maximum ($\alpha = 1$) (***Freund, 2020***; ***Koskela and Wilke Berenguer, 2019***). Earlier work (***Matuszewski et al., 2017***), using a model in which a single individual reproduces each time and occasionally wins the jackpot whose size is constant over time, also found reproductive skew over demographic expansion in Japanese sardines. We used a more realistic model (***Schweinsberg, 2003***), in which the whole population reproduces simultaneously, however, a single random female occasionally hits a jackpot, whose size will vary over time.

$$(2 - \alpha, \alpha)$$
$$(2 - \alpha, \alpha)$$

The $\Xi$-Beta$(2 - \alpha, \alpha)$ model of random sweepstakes showed that reproductive skew is a more likely explanation than demographic expansion under the classical Kingman model and the model predicts an upswing, as observed at the right tail of the site-frequency spectrum. It nevertheless cannot adequately explain our data. There were systematic deviations from expectations of the model (see residuals in ***Figure 4a, b*** and ***Figure 4—figure supplement 1a, b***). The deviations were nearly symmetrical around a derived allele frequency of 50% (logit of 0), and rare (less than 12%, logit of $-2$) and common alleles (greater than 88%, logit of 2) were too frequent. In contrast, intermediate alleles were too few compared to model expectations. The deviations immediately suggest the action of positive natural selection by selective sweeps.

Therefore, we investigated the hypothesis of selective sweepstakes by comparing our observations to predictions of the Durrett–Schweinsberg coalescent derived from the Durrett–Schweinsberg model (Appendix 4). In the Durrett–Schweinsberg model, a random site on a chromosome is hit by a beneficial mutation that, with a certain probability, goes to fixation in a time measured in $\log N$ coalescent time units, where $2N$ is the population size. The beneficial mutation sweeps with its neutral sites that are some recombinational distance from the selected site (***Durrett and Schweinsberg, 2005***; ***Nielsen, 2005***). Distant sites are more likely to escape this hitchhiking effect than neighbouring sites because of larger recombination rates. Even though the model is built from a whole chromosome undergoing recurrent selective mutations, the resulting coalescent only describes a single site under the joint effect of hitchhiking and recombination (***Nielsen, 2005***). Thus, the model cannot make joint predictions about several sites, such as measures of linkage disequilibrium. In Appendix 4, we propose a two-site extension of the Durrett–Schweinsberg model in the restricted case of two sampled sequences, facilitating predictions of linkage disequilibrium. This model of recurrent selective sweeps explained our results for all subsets of the data (and GL1 and GL2 in both populations ***Figure 3***, ***Figure 3—figure supplement 1***, and ***Appendix 6—figure 5***). We also considered the potential effects of SNP misorientation and low-level ancestral introgression (***Baudry and Depaulis, 2003***; ***Hernandez et al., 2007***; ***Schumer et al., 2018***) (Appendix 10). Polarizing the site-frequency

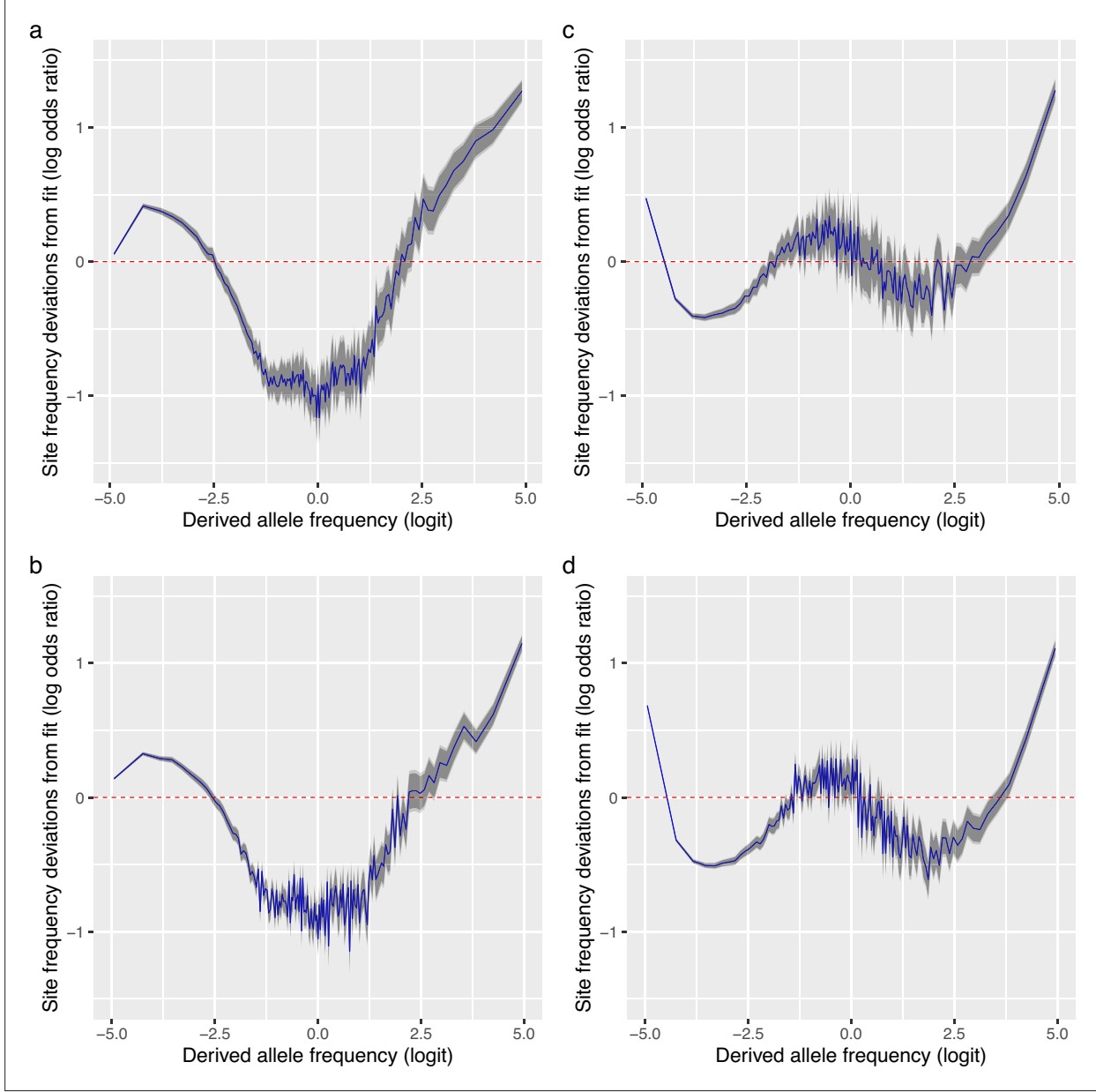

**Figure 4.** Deviations from fit to the random sweepstakes model and the selective sweepstakes model. (**a, b**) Deviations of site frequencies from approximate maximum likelihood best-fit expectations of the neutral $\Xi$-Beta$(2 - \alpha, \alpha)$ coalescent modelling random sweepstakes. Deviations of the mean site frequencies of non-inversion chromosomes 3–6, 8–11, and 13–23 estimated with genotype likelihoods GL1 from best-fit expectations of the $\Xi$-Beta$(2 - \alpha, \alpha)$ coalescent with $\hat{\alpha} = 1.16$ for the South/south-east population (sample size $n = 68$) (**a**) and with $\hat{\alpha} = 1.16$ for the Þistilfjörður population (sample size $n = 71$) (**b**). Deficiency of intermediate allele frequency classes and excess mainly at right tail of site-frequency spectrum. (**c, d**) Deviations of GL1 estimated site frequencies from expectations of the Durrett–Schweinsberg model of recurrent selective sweeps for the South/south-east population with a compound parameter $c = 8.25$ and the Þistilfjörður population with a compound parameter $c = 6.3$, respectively. Better fit than random model but also with excess at right tail of site-frequency spectrum. Deviations reported as the log of the odds ratio (in blue), the difference of the observed and expected logit of site frequencies. The dashed red line at zero represents the null hypothesis of no difference between observed and expected. The darker and lighter shaded gray areas represent the 95% and the 99% confidence regions of the approximately normally distributed log odds ratio. *Figure 4—figure supplement 1* shows comparable deviation from fit for the GL2 genotype likelihood data.

The online version of this article includes the following figure supplement(s) for figure 4:

**Figure supplement 1.** Deviations from fit to the random sweepstakes model and the selective sweepstakes model for GL2 genotype likelihood data.

spectra with a 100% consensus of several outgroup sequences (*Figure 3—figure supplement 2*), did not change the overall pattern. Considering transversions only (*Figure 3—figure supplement 4*) (avoiding mutational saturation of transitions, *Agarwal and Przeworski, 2021*) also did not change the overall pattern. Finally, truncating the site-frequency spectra (by removing the singleton and doubleton and $n-1$ and $n-2$ class of sites most affected by SNP misorientation) also did not change the overall results (*Figure 3—figure supplement 4*). Linkage disequilibrium decays rapidly to background values (*Appendix 6—figure 6a*) as the Durrett–Schweinsberg model requires. The decay of linkage disequilibrium observed in the data was also consistent with predicted results from our two-site, two-sample extension of the Durrett–Schweinsberg model (Appendix 4 and *Appendix 6— figure 6b*), provided that small fractions of sweeps (on the order of 10%) are taken to affect the whole chromosome regardless of recombination. Such 'sweeps' are characteristic of e.g. population

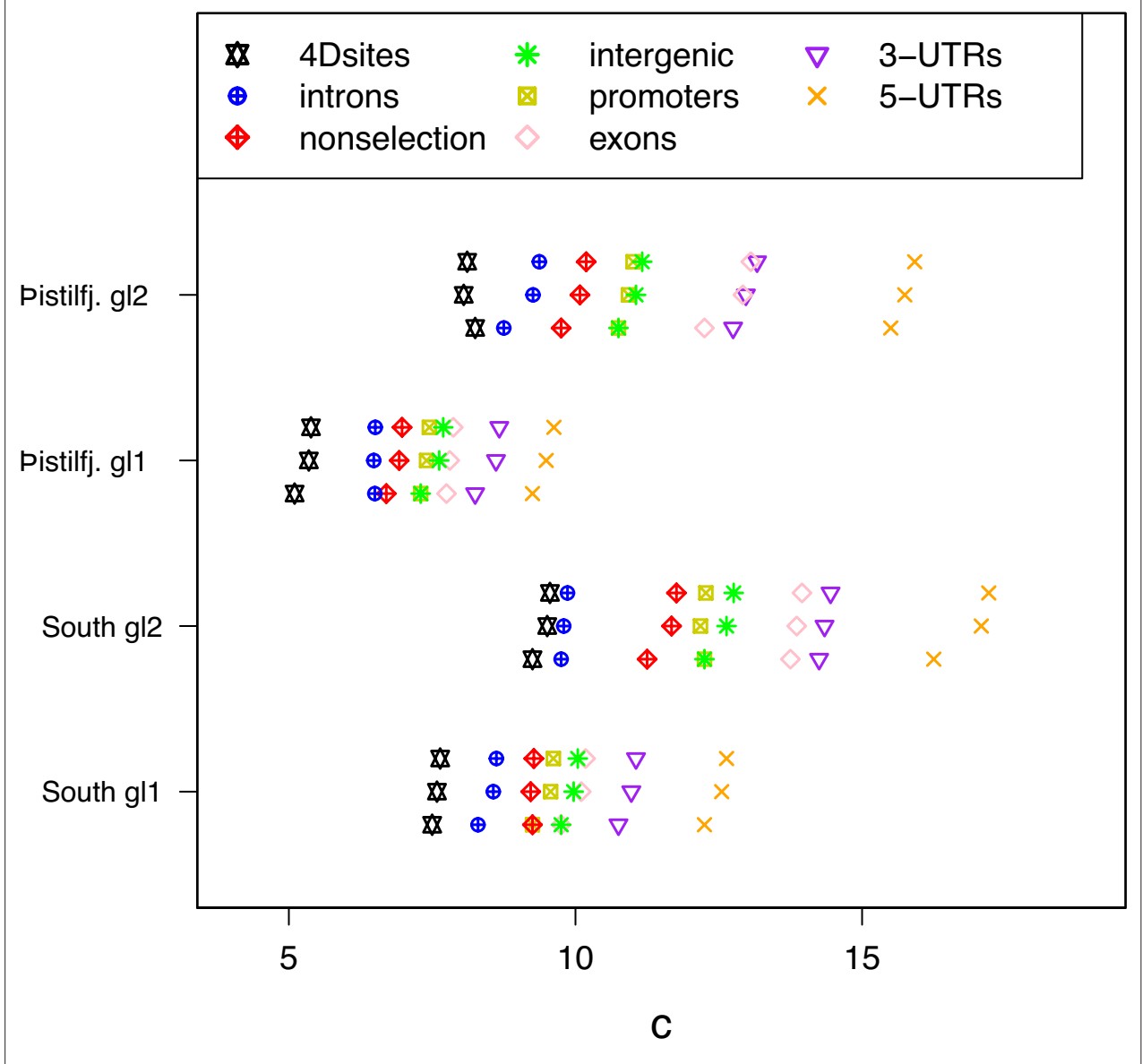

**Figure 5.** Approximate Bayesian computation (ABC) estimation of parameters of the Durrett-Schweinsberg coalescent (*Durrett and Schweinsberg, 2005*) (the selective sweepstakes model) for various functional regions of the genome. For each category from top to bottom the mean, the median, and the mode of the ABC-posterior distribution of the compound parameter $c \in \Theta_{DS}$ using site-frequency spectra computed from likelihood GL1 and GL2 for the South/south-east (South) and Þistilfjörður (Þistilfj.) populations. The different functional groups are fourfold degenerate sites (4Dsites), intronic sites, non-selection sites (sites more than 500 kb away from peaks of selection scan, *Appendix 6—figure 8*), intergenic sites, promoters, exons, $3'$-UTR sites (3-UTRs), and $5'$-UTR sites (5-UTRs), regions ranging from less to more constrained by selection.

bottlenecks. The compound parameter $c = \delta s^2/\gamma$ of the Durrett–Schweinsberg model measures the rate of selective sweeps ($\delta$) times the squared selection coefficient ($s^2$) of the beneficial mutation over the recombination rate ($\gamma$) between the selected site and the site of interest. The compound parameter is essentially the density of selection per map unit along the chromosome (*Aeschbacher et al., 2017*). ABC estimates yielded similar values across all replicated data sets, an average of about 10 that is 10 times more frequent than the coalescence rate of a sample with a low variance mode of reproduction described by the classical Kingman coalescence (*Figure 5* and *Appendix 6—figure 7*). The estimated compound parameter was correlated with functional constraints and importance of sites indicating a higher selection density per generation per genetic map unit in exons and UTRs (*Figure 5*). The residuals of the fit to the Durrett–Schweinsberg coalescent (*Figure 4* and *Figure 4—figure supplement 1c, d*) showed deviations that were both smaller and opposite the deviations of those of the neutral $\Xi$-Beta$(2 - \alpha, \alpha)$ model (*Figure 4a, b* and *Figure 4—figure supplement 1a, b*) with intermediate frequency classes being too frequent. The Durrett–Schweinsberg model is essentially haploid. We suggest that a diploid model, where dominance generates two phenotypes such that selection acts on pairs of chromosomes jointly rather than single chromosomes as in the Durrett–Schweinsberg model would provide an even better fit. However, developing a diploid multi-locus version of the Durrett–Schweinsberg model is outside the scope of the present work. Nevertheless, a comparison of our data with predictions of the Durrett–Schweinsberg model, in particular in comparison with our additional analysis, is perfectly valid. Overall, the selective sweepstakes hypothesis embodied in the Durrett-Schweinsberg coalescent (*Durrett and Schweinsberg, 2005*) modelling recurrent selective sweeps, in essence, explained our data, whereas the hypothesis of low-variance reproduction and one of random sweepstakes did not.

We took several steps to investigate and consider the effects of selection and recombination on the observed patterns of allele frequencies. We did a principal component (PC)-based genome-wide scan of selection (using PCangsd; *Meisner and Albrechtsen, 2018*) and detected several peaks (*Appendix 6—figure 8*). We used sites that are at least 500 kb away from selective peaks. We refer to these as non-selection sites. We extracted sites from the genome that are likely under different selective constraints. We thus extracted fourfold degenerate sites (referred to as 4Dsites), intron sites, intergenic sites, promoter sites, $5'$ UTR sites, $3'$ UTR sites, and exon sites. The less constrained sites are not necessarily neutral to selection. For example, although silent at the protein level, mutations at fourfold degenerate sites could affect transcriptional and translational efficiency and mRNA stability, thus giving rise to selection for or against such sites. However, the first three sets of sites are generally considered less constrained and the other sets are more constrained by selection. The resulting site-frequency spectra and parameter estimates ranked according to selective constraints (*Figures 3 and 5*, and *Appendix 6—figure 7*).

Furthermore, we used OmegaPlus (*Alachiotis et al., 2012*) and RAiSD (*Alachiotis and Pavlidis, 2018*) to detect selective sweeps genome-wide. Both methods use local linkage disequilibrium to detect sweeps (*Nielsen, 2005*). In addition, RAiSD uses a local reduction in levels of polymorphism and shifts in the frequencies of low- and high-frequency derived alleles affecting, respectively, the left and right tails of the site-frequency spectrum. Both methods indicated pervasive selective sweeps on all chromosomes (*Figure 6*). We also used SLiM (*Haller and Messer, 2019*) to simulate positive selection under the no-sweepstakes Wright–Fisher model and a random sweepstakes model in the domain of attraction of a Xi-Beta coalescent (*Appendix 6—figure 9*). We tried various forms of dominance of selection among diploid genotypes (semidominance, $h = 0.5$ and full dominance, $h = 1.0$) with different strengths of selection (selection coefficient $s$). The model of the successive selective pass or fail filters suggests that lacking a function (a derived allele) is a failing genotype while having a function (derived allele) is a passing genotype as modelled by full dominance. The observation of the heavy mortality of immatures (type III survivorship, *Appendix 7—table 3*) therefore suggests a model of selection against a recessive lethal and for a dominant. This is a two-phenotype model for a diploid organism. The results of the SLiM simulations of positive selection (*Appendix 6—figure 9*) gave site-frequency spectra that were qualitatively similar to the observed spectra. Selection for a semidominant phenotype produced more U-shaped spectra while selection for a dominant produced more V-shaped spectra similar to those observed. Recurrent hard sweeps interrupting the standard Kingman coalescent (simulated using `msprime`; *Baumdicker et al., 2021*) produced U-shaped

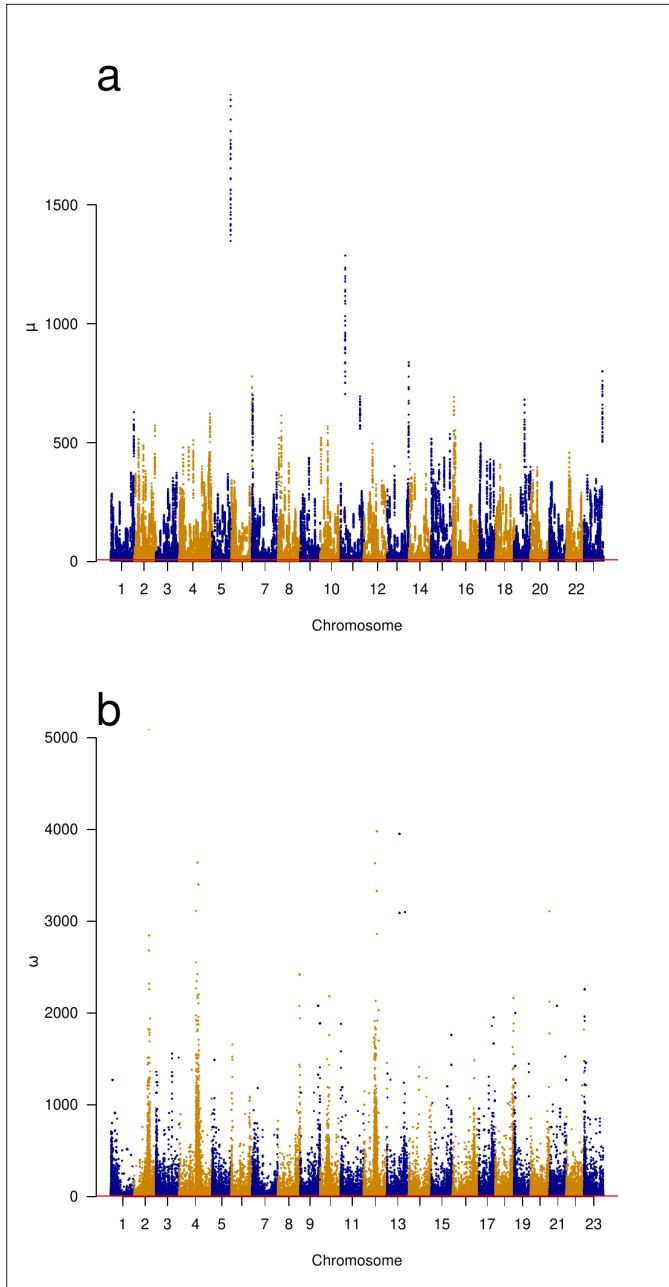

**Figure 6.** Genomic scans of selective sweeps by two methods. (**a**) Manhattan plots from detection of selective sweeps using RAiSD (**Alachiotis and Pavlidis, 2018**) and (**b**) by using OmegaPlus (**Alachiotis et al., 2012**). The $\omega$ statistic of OmegaPlus (**b**) measures increased linkage disequilibrium in segments on either the left or the right sides of a window around selected site and a decrease in linkage disequilibrium between the segments across the selected site (**Kim and Nielsen, 2004**; **Alachiotis and Pavlidis, 2018**). The $\mu$ statistic of RAiSD (**a**) is a composite measure based on three factors, a reduction of genetic variation in a region around a sweep, a shift in the site-frequency spectrum from intermediate- towards low- and high-frequency derived variants, and a factor similar to $\omega$ that measures linkage disequilibrium on either side of and across the site of a sweep. Chromosomes with alternating colours. Indications of selective sweeps are found throughout each chromosome.

site-frequency spectra (*Appendix 6—figure 10*) that are qualitatively similar to our data from the South/south-east coast.

## Synopsis of results

We have shown that the Durrett–Schweinsberg coalescent modelling recurrent selective sweeps affecting linked sites gives the best fit for our observations (*Figure 3*). By extension, the hypothesis of reproduction by selective sweepstakes is best supported by our data. The Kingman coalescent and Wright–Fisher model of reproduction, without a strong positive selection of recurrent strongly beneficial mutations (*Appendix 6—figure 9* and *Appendix 6—figure 10*), cannot explain our data. Similarly, the model of random sweepstakes, the Xi-Beta coalescent, in which a random individual has windfall reproductive success, although fairing better than the Kingman coalescent neverthe-less cannot explain the observations. In Appendix 11, we ask if other processes can better explain the observed patterns and provide a detailed analysis of alternatives. Through analysis and forward and backward simulations, we consider historical demography under low-variance reproduction (see Appendix 12), the potential confounding due to cryptic population structure (see Appendix 13), the effects of balancing selection of large inversions (see Appendices 14 and 15), the effects of negative and background selection (see Appendix 15), the joint action of several evolutionary mechanisms (see Appendix 16), and the potential effects of SNP misorientation from parallel mutation and low-level ancestral introgression (see Appendix 16). Although some alternative mechanisms can come close to observations under some parameter values, they did not provide a satisfactory fit overall (see Appendix 11). Historical demographic expansions or contractions do not explain our data (*Appendix 6—figure 11* and *Appendix 6—figure 12*). Analysis of potential cryptic population structure does not provide answers to our patterns (*Appendix 6—figure 13*). Similarly, modelling sampling from divergent popu-lations, a combination of extreme parameter values can produce patterns similar to the observed patterns (*Appendix 6—figure 14* and *Appendix 6—figure 15*). However, a leave-one-out anal-ysis of our sample shows that our sample was not produced under such extreme parameter values (*Appendix 6—figure 16*). There are clear signals of balancing selection of large inversions at four chromosomes (see Appendix 14, *Appendix 6—figure 17*, and *Appendix 6—figure 18*). However, balancing selection does not change the overall shape of the site-frequency spectrum of these chro-mosomes, which is the summary statistic we use for our analysis. Simulations of background selection show that a narrow window of parameter space can resemble observed patterns. Still, in general, background selection does not fit our results (*Appendix 6—figure 9d* and *Appendix 6—figure 19*). Finally, simulations of the joint action of several evolutionary mechanisms, notably of demography and background selection with or without selective sweeps, do not produce qualitatively accurate U-shaped site-frequency spectra similar to the observed except in simulations that included selective sweeps (*Appendix 6—figure 19*).

## Discussion

Understanding recruitment dynamics and what shapes the distribution of individual reproductive success is a fundamental challenge in evolutionary genomics of high-fecundity organisms. It is key to further understanding metapopulation and community dynamics, predicting response to anthro-pogenic challenges, for conservation and management, and further development of the ecological and evolutionary theory (*Eldon, 2020*). We show that selective sweepstakes, modelled by a partic-ular example of the Durrett–Schweinsberg multiple-merger coalescent derived from a population model of recurrent selective sweeps (*Durrett and Schweinsberg, 2005*), essentially explain our data. Even a model of recurrent but incomplete selective sweeps (*Coop and Ralph, 2012*) similarly leads to U-shaped site-frequency spectra generated by a multiple-merger coalescent model similar to the Durrett–Schweinsberg model. We further show that neither non-sweepstakes reproduction nor random-sweepstakes reproduction can explain our data. Other biologically plausible scenarios (e.g. historical demographic changes, cryptic breeding structure, and background selection) show a much poorer fit. Our results indicate that strong pervasive positive natural selection is pivotal in reproduc-tive sweepstakes, more so than in windfall sweepstakes (*Hedgecock and Pudovkin, 2011*).

The random sweepstakes $\Xi$-Beta$(2-\alpha,\alpha)$ model assumes a single-pair mating with enormous reproductive output. However, cod is a batch spawner in which a female may pair with a different male for each batch, with potential sneaker males participating in fertilization as well (*Hutchings et al., 1999*; *Nordeide, 2000*). This mating system may result in larger fertilization success than in monogamous broadcast spawning. The $\Xi$-Beta$(2-\alpha,\alpha)$ models the simultaneous coalescence of the four parental chromosomes involved in a large reproductive event, the random sweepstakes. The cod mating system implies that the two maternal chromosomes of a female combine with many pairs of paternal chromosomes with more genetic diversity than in a high-fecundity monogamous system. However, how such a mating system affects the coalescent and the shape of the site-frequency spectrum is unclear.

We have considered models based on haploid reproduction, or diploid reproduction with monogamous pairs. It is possible that a two-sex model which more accurately reflects the mating traits of cod, in which many different males can fertilize the eggs of one female (*Hutchings et al., 1999*; *Nordeide, 2000*), may further improve the fits we have obtained. *Birkner et al., 2018* provide a framework for such modelling. We have chosen to use simpler, monogamous models as a starting point for our analysis and leave the development of a parsimonious mating structure model for future work.

By the time an advantageous mutation reaches the exponential phase of the Durrett–Schweinsberg process, recombination during the lag phase will have broken up the initial linkage disequilibrium of a new mutation to a haplotype composed of a chromosomal segment. The evolution of that mutation is haplotype specific. In contrast, random sweepstakes would increase the frequency of genomes of the reproducing pair. The Durrett–Schweinsberg model assumes a Kingman coalescent interrupted by a selective sweep. However, the Durrett–Schweinsberg model needs to better capture the low-frequency singleton and doubleton class of sites. In contrast, the random sweepstakes $\Xi$-Beta$(2-\alpha,\alpha)$ very well captures the low-frequency singleton and doubleton class of sites. It is possible that mutations are entering the population under random sweepstakes, many being lost but some drifting to a high enough frequency that they contribute sufficiently to the variance in fitness to be grabbed by the selective process and swept to fixation. Can random sweepstakes possibly also increase the frequency of variants and thus facilitate selective sweepstakes? There is a larger variance of allele frequencies under random sweepstakes, so that many variants will be lost. We leave for future work the question of whether random sweepstakes interact with selective sweepstakes in this way. Interpreting the Durrett–Schweinsberg model as approximating selective sweepstakes, we conclude that our findings are strong evidence for selective sweepstakes (*Williams, 1975*) characterizing the distribution of individual reproductive success of the highly fecund Atlantic cod. Under the Durrett–Schweinsberg coalescent of recurrent selective sweeps, the rise in frequency of new mutations each time, happens rapidly compared to the coalescent timescale. The continuous input of new beneficial mutations represents the Sisyphean genotypes that forever climb a selective peak under Williams' concept of selective sweepstakes (*Williams, 1975*). By extension, selective sweepstakes are the life history of highly fecund organisms with skewed offspring distribution.

Recurrent bottlenecks may mimic the effects of recurrent selective sweeps (*Galtier et al., 2000*). The duration, depth, and rate of recovery of a bottleneck (*Nei et al., 1975*) relative to the $\log N$ timescale of recurrent sweeps under the Durrett–Schweinsberg model is an important issue. A small number of individuals having large numbers of descendants due to a bottleneck and rapid recovery or due to a selective sweep will in both cases lead to multiple mergers in the genealogy. Our simulations of random sweepstakes with recurrent bottlenecks yield roughly a U-shaped site-frequency spectrum, but the fit is not as good as for the selective sweepstakes model. In the Durrett–Schweinsberg model, interpreting a small fraction of sweeps (on the order of 10%) as chromosome-wide sweeps or population bottlenecks resulted in a model which was able to explain the decay of linkage disequilibrium observed in Atlantic cod, without affecting the good fit of the site-frequency spectrum. Overall, therefore, the Durrett–Schweinsberg model explains our data although it is formally only applicable to single-locus data from a haploid species (the resulting coalescent process traces the genealogy of a single site), assumes a constant population size, disallows competing, simultaneous sweeps (*Kim and Stephan, 2003*), and only models hard sweeps. Both soft and incomplete sweeps and recombinational breakups likely occur in cod. Our estimator will pick up the effects of such sweeps. Incomplete sweeps and LD-based measures (*Nielsen, 2005*; *Sabeti et al., 2006*; *Sabeti et al., 2007*) may be

neccessary, particularly in connection with the further extensions of the Durrett–Schweinsberg model that we present in *Appendix 6—figure 6*. Further extending the model is an avenue for future work.

High-fecundity matters in two ways in this process. First, each round of replication results in many new mutations in the genome of a new gamete. Even though the probability of a positive mutation is very small, the millions of gametes produced by each female multiplied by the billions of individuals in a population ensure a steady input to the population of new positive mutations to each generation. Second, high fecundity makes available a large reproductive excess which permits substitutions to occur at high rates by natural selection without the population going extinct (*Felsenstein, 1971*). The reproduction of a high-fecundity organism compares with the replication of a virus in an epidemic. Each infected individual produces hundreds of billions of viral particles. Even with a tiny proportion of positive mutations, the numbers of new mutations are so enormous that it is all but certain that an epidemic produces a steady stream of more contagious and fitter viral variants that sweep to fixation by selection. If the population crashes (*Hutchings, 2000*) the mutational input of adaptive variation diminishes. The population may run out of fuel for responding to environmental challenges via selective sweeps and go extinct (*Felsenstein, 1971*). Kimura's neutral theory of molecular evolution and polymorphisms (*Kimura, 1968*) relied on excessive genetic load based on Haldane's dilemma (*Haldane, 1957*) that the cost of adaptive substitution would limit the rate of evolution lest the population go extinct (*Felsenstein, 1971*). Truncation selection of continuously distributed characters, where genetic and nongenetic factors independently affect the probability of survival and act cumulatively in each individual (*Williams, 1975*), mitigates the genetic load (*King, 1967*; *Sved et al., 1967*). Our considerations of full dominance with selection against a lethal homozygote would entail a large genetic load. However, there can be strong selection in one habitat patch and near neutrality in another due to differences in competition and predation. The marginal fitness differences would then be less but such soft selection (*Wallace, 1975*; *Reznick, 2016*) would not drive the population to extinction (*Charlesworth, 2013*). Marginal fitness would still preserve full dominance and a two phenotype selection scheme and thus behave similar to the haploid Durrett–Schweinsberg model. The high fecundity and consequent excessive reproductive capacity in our study organism may also alleviate the genetic load problem. However, both loss of mutational input and genetic load (a case of selective extinction) may nevertheless be a factor in the non-recovery of a population following a crash (*Hutchings, 2000*). Does cod have the reproductive capacity to substitute adaptive alleles at a high rate without going extinct from the substitution load (*Kimura and Maruyama, 1969*)? If each high-fitness fish has $k$ offspring, which survive long enough to reproduce, and the selective sweep starts from a single individual, then after one generation, there are $k$ fit fish; after 2 there are $k^2$; after 3 there are $k^3$, and so on. At sweep's completion, there are $k^{(\log N)}$ fit descendants after $\log N$ generations. For the sweep to complete, we thus need $k^{(\log N)} = N$, or $k = N^{(1/\log N)} = e$, the base of the natural logarithm. As a numerical example, it is immaterial whether we assume a population size of a billion ($N = 10^9$) and duration of a sweep 20.7 generations or a population size of a trillion ($N = 10^{12}$) fish and a sweep 27.6 generations. The reproductive excess required is 2.71 or approximately three fit offspring that make it to reproduction. In practice, the number will have to be larger because not all fit offspring will survive to reproduce and because our estimated frequency of sweeps was large enough that 20–30 generations might be a bit too long. However, there is no reason to think that the cod population would not have the reproductive capacity to support selective substitution at the estimated rate.

Our estimate of the rate of selective sweeps (Appendix 17) amounts to mergers of ancestral lineages of our sample happening because sweeps occur at 5–18 times larger rates than mergers due to ordinary low-variance reproduction (*Figure 5*). In the classical model, the coalescence rate is on the order of the population size, or $N$ generations, but the duration of selective sweeps is on the order of $\log N$ generations. If we assume that there is a billion cod in the Icelandic population, this is some 20 generations or about 100 years from when a beneficial mutation arises until fixation. With the sigmoid nature of the positive selection curve, with a lag phase followed by an exponential phase and ending in a stationary phase, the main action of selection bringing an allele from a low frequency to a high frequency during the exponential phase may only take a few generations, perhaps 15–20 years. Erratic climatic variability, such as the great salinity anomalies (*Cushing, 1969*; *Dickson et al., 1988*) in the North Atlantic, which can greatly affect cod reproduction and ecology, is detectable over decadal timescales, similar time span as the exponential phase of our estimated selective sweeps.

We estimate that each chromosome in Atlantic cod is affected by a selective sweep every 23–50 years on average (Appendix 17). Since we also see evidence of rapid recombination (*Appendix 6—figure 6*), we expect that any single sweep will not strongly affect a large region of a chromosome. The rapid recombination will modulate the genomic footprints of sweeps. There is clear evidence that sweeps occur everywhere along the genome (in chromosomal fragments of different sizes, different functional groups, and on all chromosomes *Figure 5* and *Appendix 6—figures 3 and 17* and *Appendix 6—figure 18*). It is, therefore, likely that the true rate of sweeps is even faster than our estimate. For example, if an average sweep were to affect 10% of a chromosome, we would expect to see sweeps every 3–4 years or roughly once a generation to explain our results. Building a fully quantitative, data-informed picture of the rate of sweeps requires developing a diploid, genomic version of the Durrett–Schweinsberg model, which is currently absent from the literature, and for which task our results provide strong applied motivation.

The higher positive than negative selection rate is similar to findings in *Drosophila* and different from humans and yeast, where negative selection predominates (*Li et al., 2008*). Similarly, the proportion of adaptive non-synonymous substitutions is lower but in the direction of the results of *Drosophila* (*Bierne and Eyre-Walker, 2004*; *Sella et al., 2009*). Our study is of a locally circumscribed population compared to a more geographically diverse sampling of the fly. A global sample of cod would likely show an even higher proportion of advantageous mutations.

Is the large substitution rate of one sweep per year even possible? If we accept a 3.5 Mya split of Atlantic cod and walleye pollock (*Vermeij, 1991*; *Vermeij and Roopnarine, 2008*; *Coulson et al., 2006*; *Carr and Marshall, 2008*) the rate of one substitution per year (*Appendix 7—table 4*) would translate into 3.5 M site difference between the taxa. *Appendix 7—table 4* also shows that the $p$ distance (proportion of nucleotide differences per nucleotide) is 0.005, and with a 685-Mb genome, yields a 3.4-M site difference between the taxa, a fair agreement. But it is unlikely that all substitution is by selection or hitchhiking. Although the proportion of adaptive substitutions ($\alpha$) is substantial, there is also a role for random genetic drift in substitution. Our findings highlight genetic hitchhiking as a key driver of substitutions in cod. The fitted value $c \approx 10$ can be thought of as a rate with which hitchhiking drives a given (neutral) mutation towards fixation, in contrast to a rate of 1 for genetic drift as modelled by the Kingman coalescent. However, as a compound parameter, $c$ does not carry direct information about the abundance of neutral versus selective mutations. This is comparable to *Drosophila*, for example, millions of differences between *melanogaster* and *simulans*, in which many adaptive substitutions occur (e.g. *Fay et al., 2002*; *Smith and Eyre-Walker, 2002*; *Andolfatto, 2005*; *Eyre-Walker, 2006*). We can ask (*Eyre-Walker, 2006*) what for is all this adaptive variation? Where are the camel's hump and elephant's trunk of cod? We answer that the optimal phenotype is mostly ephemeral (although balanced polymorphic inversions may tie up some long-duration adaptive variations). The population is not going anywhere in particular. This is evolution by selective sweepstakes, metaphorically a Red Queen race (*Van Valen, 1973*; *Strotz et al., 2018*) to stay in the game against nature (*Lewontin, 1961*).

Our findings provide a new perspective on coalescent models in population genetics and genomics. For the first time, a test involving genomic data, that is, using copies of chromosomes from several pairs of homologous chromosomes, was made on the contrasting hypotheses of reproduction using multiple-merger coalescents in a diploid organism. It is also the first time multiple-merger coalescent models based on neutral evolution and selection are contrasted. Previously, two neutral $\Lambda$-coalescents have been compared to data of outbreaks of the tuberculosis bacterium and the Bolthausen–Sznitman coalescent ($\alpha = 1$) used to model rapid selection (*Menardo et al., 2019*). Our findings have repercussions for and give impetus to further theoretical development of multiple-merger coalescents, particularly for multiple-merger coalescent models of strong selection. Our work motivates the construction of joint models featuring neutral and selective sweepstakes. As a starting point, we expect selective sweeps akin to the Durrett–Schweinsberg model could be incorporated into the Schweinsberg (*Schweinsberg, 2003*) pre-limiting population models giving rise to the Beta-coalescent. To affect the infinite-population limit, selective sweeps would have to occur on the same fast timescale of $N^{\alpha-1}$ generations as neutral multiple mergers. Even on this timescale, selective sweeps lasting $\log N$ generations will be instantaneous resulting in multiple mergers in the coalescent limit. An intriguing possibility is that the Durrett–Schweinsberg selective sweeps could account for some of the observed deviation from the Kingman coalescent, the combined model, might yield substantially higher best-fit

estimates of $\alpha$ than those obtained from the more restrictive Beta-coalescent. Low values of $\alpha$ result in implausibly short timescales for evolution, and a combined neutral-and-selective sweepstakes model has the potential to avoid this defect.

We suggest that sweepstakes reproduction is much more common than previously thought. It is essential to understand sweepstakes and the natural and anthropogenic ecological processes conducive to sweepstakes (*Hedgecock and Pudovkin, 2011*; *Williams, 1975*). Are selective sweepstakes (*Williams, 1975*) the rule, or is there a role for random sweepstakes (*Hedgecock and Pudovkin, 2011*; *Vendrami et al., 2021*)? It is possible that big-bang, the semelparous reproductive strategy of reproducing once before dying, is a sweepstakes reproduction if ecological mechanisms generate a high-variance, highly skewed offspring distribution. This mode of reproduction characterizes many annual plants, a myriad of insects, and vertebrates such as Pacific salmon (*Oncorhynchus*) and Arctic cod (*Boreogadus saida*), a close relative of Atlantic cod. We further posit that sweepstakes may be the mode of reproduction of viruses (*Timm and Yin, 2012*) as inferred from the overdispersion of offspring distribution from superspreader individuals and events (*Endo et al., 2020*), some cancer cells (*Kato et al., 2017*), and various bacteria (*Wright and Vetsigian, 2019*; *Menardo et al., 2019*; *Ypma et al., 2013*). Fungi and plant pathogens, which cause extensive crop losses of great economic importance (*Pimentel et al., 2000*), may also reproduce by sweepstakes. Similarly, many repeat reproducers, the iteroparous reproductive strategy, produce vast numbers of tiny eggs in each reproductive season. It applies to many marine organisms such as oysters (*Hedgecock and Pudovkin, 2011*), and Atlantic cod and its Pacific relatives (*Árnason and Halldórsdóttir, 2019*) that support large fisheries of great economic importance. The dynamics of all these systems can be profitably studied using multiple-merger coalescents (*Freund et al., 2022*), be they generated by random or selective sweepstakes.

## Materials and methods

### Sampling

We randomly sampled adults from our extensive tissue collection (*Árnason and Halldórsdóttir, 2015*; *Halldórsdóttir and Árnason, 2015*) from two localities in Iceland, the South/south-east coast ($n = 68$) and Þistilfjörður on the north-east coast ($n = 71$) (*Appendix 6—figure 1*). The Icelandic Marine Research collected the fish during spring spawning surveys (*Árnason and Halldórsdóttir, 2015*). All fish selected here had running gonads (eggs and milt with maturity index 3), indicating they were spawning at the capture locality.

### Ethics statement

The Icelandic Committee for Welfare of Experimental Animals, Chief Veterinary Officer at the Ministry of Agriculture, Reykjavik, Iceland has determined that the research conducted here is not subject to the laws concerning the Welfare of Experimental Animals (The Icelandic Law on Animal Protection, Law 15/1994, last updated with Law 157/2012). DNA was isolated from tissue taken from dead fish on board research vessels. Fish were collected during the yearly surveys of the Icelandic Marine Research Institute (and other such institutes as already described *Árnason and Halldórsdóttir, 2019*). All research plans and sampling of fish, including the ones for the current project, have been evaluated and approved by the Marine Research Institute Board of Directors, which serves as an ethics board. The Board comprises the Director-General, Deputy Directors for Science and Finance and heads of the Marine Environment Section, the Marine Resources Section, and the Fisheries Advisory Section.

### Molecular analysis

We shipped tissue samples of cod from the South/south-east coast population of Iceland to Omega Bioservices. Omega Bioservices isolated genomic DNA using the E-Z 96 Tissue DNA Kit (Omega Biotek), made picogreen DNA sample quality checks, made sequencing libraries using Kapa Hyper Prep WGS (Kapa Biosystems), used Tapestation (Agilent Technologies) for sizing libraries, and sequenced libraries on a 4000/X Ten Illumina platform with a 2 × 150-bp read format, and returned demultiplexed fastq files.

Genomic DNA was isolated from the tissue samples of Þistilfjörður population using the E-Z 96 Tissue DNA Kit (Omega Biotek) according to the manufacturer's recommendation. The DNA was normalized with elution buffer to 10 ng/μl. The normalized DNA was analyzed at the Bauer Core of

Harvard University. According to the manufacturer's recommendation, the Bauer Core used the Kapa HyperPrep Plus kit (Kapa Biosystems) for enzymatic DNA fragmentation and adapter ligation, except that the reaction volume was 1/4 of the recommended volume. The target insert size was 350 base pairs (bp) with a resulting average of 487 bp. The libraries were indexed using IDT (Integrated DNA Technologies) unique dual 8 bp indexes for Illumina. The Core uses Tapestation (Agilent Technologies) and Picogreen qPCR for sizing and quality checks. Multiplexed libraries were sequenced on NovaSeq (Illumina) S4 lanes at the Broad Institute with a $2 \times 150$ bp read format, and demultiplexed fastq files were returned.

## Bioinformatic analysis

The sequencing centres returned de-multiplexed `fastq` files for different runs of each individual. Data processing followed the Genome Analysis Toolkit (GATK) best practices (*Van der Auwera et al., 2013*) as implemented in the fastq_to_vcf pipeline of Alison Shultz (https://github.com/ajshultz/comp-pop-gen; *Shultz, 2020*). Using the pipeline the raw reads were adapter trimmed using `NGmerge` (*Gaspar, 2018*), the trimmed `fastq` files aligned to the gadMor3.0 chromosome-level reference genome assembly (NCBIaccessionID:GCF_902167405.1) using `bwa mem` (*Li and Durbin, 2009*), and the resulting `bam` files deduplicated, sorted, and indexed with `gatk` (*Van der Auwera et al., 2013*).

The deduplicated bam files were used for population genetic analysis with `ANGSD` (*Korneliussen et al., 2014*). We have sequenced a large sample of cod from various localities in the North Atlantic and performed both principal component (PCA) and admixture analysis using `PCangsd` (*Meisner and Albrechtsen, 2018*) revealing some population substructure and possible admixture (unpublished results). To minimize the effects of potential population substructure we screened the individuals of the two samples in this study and ensured that they are members of the same cluster detected by PCA and assigned to the same population detected with admixture. This filtering also addresses the issue of potential SNP misorientation and ancestral admixture discussed below. In order to polarize sites for estimation of site-frequency spectra outgroup fasta sequences were generated with `-dofasta 3`, which chooses a base using an effective depth algorithm (*Wang et al., 2013*). A high coverage specimen (*Árnason and Halldórsdóttir, 2019*) from each of Pacific cod *Gadus macrocephalus* (labelled Gma), walleye pollock, also from the Pacific, *G. chalcogrammus* (labelled Gch), Greenland cod *G. ogac* (labelled Gog), and Arctic cod *Boreogadus saida* (labelled Bsa) were each taken individually as an outgroup providing independent replicate estimation of site-frequency spectra. We used biallelic sites only with `-skipTriallelic 1` filtering in `ANGSD`, which will leave only sites that have the same exact ancestral state in the outgroup as one of the two alleles in the ingroup. In conjunction with multiple outgroups this filtering addresses some issues with SNP misorientation. If a particular site can be polarized by outgroup A (e.g. Gma) it means that the state of the site in taxon A is the same as one of the alleles segregating in the ingroup population. If outgroup B (say Gch) has a different state for that site, the site would would be tri-allelic in that comparison and removed by the tri-allelic filtering. We did not use parsimony or consensus to infer the state of ancestral nodes (*Keightley and Jackson, 2018*). Therefore, this filtering will not remove sites which have parallel changes simultaneously in two or three outgroup taxa. To address the potential effects of SNP misorientation from parallel mutation (*Baudry and Depaulis, 2003*; *Hernandez et al., 2007*) or from ancestral introgression (*Schumer et al., 2018*) we generated a 100% consensus sequence (with perl script available from https://github.com/josephhughes/Sequence-manipulation/blob/master/Consensus.pl; *Hughes, 2011*) from walleye pollock (Gch), Pacific cod (Gma), and Arctic cod (Bsa) sequences and used the consensus sequence to polarize sites. There is potentially mutational saturation of transition sites (*Agarwal and Przeworski, 2021*) that complicates polarization of sites. We used the `-rmTrans` flag to remove transitions and study variation at transversion sites only. The effects of SNP misorientation from parallel mutation (*Baudry and Depaulis, 2003*; *Hernandez et al., 2007*) or from low-level ancestral introgression (*Schumer et al., 2018*) will primarily affect the singleton and doubleton as well as the anti-singletons ($n - 1$) and anti-doubletons ($n - 2$) site-frequency classes. We, therefore, also removed these classes of sites and used truncated site-frequency spectra to minimize the effects SNP misorientation and ancestral introgression.

To estimate site-frequency spectra the site allele frequency likelihoods based on genotype likelihoods were estimated using `ANGSD` and polarized with the respective outgroup using the `-anc` flag with `-doSaf 1` and `-doMajorMinor 1` for both genotype likelihoods 1 and 2 (GL1 the SAMtools

genotype likelihood, `-GL 1` and GL2 the GATK genotype likelihood, `-GL 2`). Filtering was done on sequence and mapping quality `-minMapQ 30 -minQ 20`, indel realignment `-baq 1-C 50`, quality checks `-remove_bads 1 -uniqueOnly 1 -only_proper_pairs 1 -skipTrial-lelic 1`, and finally the minimum number of individuals was set to the sample size (e.g. `-minInd 68`) so that only sites present in all individuals are selected. Errors at very low-coverage sites maybe called as heterozygotes. Similarly, sites with very high-coverage (more than twice or three times the average) may represent alignment issues of duplicated regions such that paralogous sites will be called as heterozygous. We addressed the issues of coverage with two steps. First, we screened out individuals with an average genome-wide coverage less than 10× giving samples sizes of $n = 68$ and $n = 71$ for the South/south-east and the Þistilfjörður populations, respectively. This resulted in an average coverage of 16× and 12× for the South/south-east and the Þistilfjörður populations, respectively. Second, we determined the overall coverage of all sites in the genome that passed the quality filtering. We then used the minimum and maximum of the boxplot statistics ($Q_1 - 1.5 \times$ IQR and $Q_3 + 1.5 \times$ IQR, which represent roughly $\mu \pm 2.7\sigma$ for a normal distribution) to filter sites using the `ANGSD` flags `-setMinDepth` $Q_1 - 1.5 \times$ IQR and `-setMaxDepth` $Q_3 + 1.5 \times$ IQR thus removing sites with a boxplot outlier coverage. We did this filtering separately for each chromosome. All our site-frequency spectra are estimated using these flags. The site-frequency spectra of the full data for each chromosome were then generated with `realSFS` using default flags. Site-frequency spectra for genomic regions used the `-sites` flag of `realSFS` with the sample allele frequency files (`saf`) files estimated with the above filtering and was thus based on the same filtering.

We use the logit transformation, the log of the odds ratio $\log(p/(1 - p))$, to analyze the site-frequency spectra. We transform both the derived allele frequency and the normalized site frequency. Under this transformation, the overall shape of the site-frequency spectrum (L-shape, U-shape, V-shape) is invariant. We used the kernel density estimator and functions of the `eks` R package (***Duong, 2022***) to estimate and plot density contours of parameter estimates.

To investigate divergence among gadid taxa we used `ANGSD` to generate beagle likelihoods (`-GL 1`, `-doGlf 2`) and the quality filtering above. We then used `ngsDist` (***Vieira et al., 2015***) to estimate the *p*-distance as nucleotide substitutions per nucleotide site between Atlantic cod and walleye pollock. The number of sites (`--n_sites`) was set to the number of variable sites and the total number of sites (`--tot_sites`) was set equal to the number of sites that passed the quality filtering in the estimation of the site-frequency spectra above (***Appendix 7—table 4***). A tree (***Appendix 6—figure 20***) was generated with `fastME` (***Lefort et al., 2015***) and displayed using `ggtree` (***Yu et al., 2016***).

To evaluate deviations from neutrality, we used `ANGSD` to estimate the neutrality test statistics Tajima's *D* (***Tajima, 1989***), Fu and Li's *D* (***Fu and Li, 1993***), Fay and Wu's *H* (***Fay and Wu, 2000***), and Zeng's *E* (***Zeng et al., 2006***) in sliding windows (window size 100 kb with 20 kb step size).

We generated `vcf` files for the South/south-east population using `GATK` (***Van der Auwera et al., 2013***). We used the genomic features files (`gtf`) of the Gadmor3 assembly to extract sites belonging to different functional groups. We used `ReSeqTools` (***He et al., 2013***) to extract fourfold degenerate sites, `bedtools` (***Quinlan and Hall, 2010***) to extract exon and intron sites using genomic feature files (`gtf`), and we used the `GenomicFeatures Bioconductor package` (***Lawrence et al., 2013***) for extracting other functional regions. We then used the `-sites` flag of `realSFS` to estimate site-frequency spectra from the sample allele frequency (`saf`) files of the entire data for each chromosome, thus keeping the quality and coverage filtering applied to the full data (Bioinformatic analysis). We used `PopLDdecay` (***Zhang et al., 2018***) to estimate the decay of linkage disequilibrium. To perform the McDonald–Kreitman test of selection (***McDonald and Kreitman, 1991***) we used `SnpEff` (***Cingolani et al., 2012***) to estimate the number of polymorphic non-synonymous and synonymous ($P_n$ and $P_s$) sites of protein-coding genes. To estimate the number of fixed non-synonymous and synonymous ($D_n$ and $D_s$) sites, we used a single individual with the highest coverage (32×) from the South/south-east population and a single high coverage (31×) Pacific cod individual and counted sites that are homozygous within species while exhibiting different allelic states between species. We used the neutrality index $NI = (P_n/P_s)/(D_n/D_s)$ (***Rand and Kann, 1996***) transformed as $-\log NI$ as an index of selection with negative values implying negative (purifying and background) selection and positive values implying positive selection (selective sweeps).

We did a principal components (PC) based scan of selection using `PCangsd` (***Meisner and Albrechtsen, 2018***) (`python pcangsd.py -selection`), which implements the fastPCA method

of *Galinsky et al., 2016*. We then removed regions of 500 kb on either side of selective peaks that exceeded $\log_{10} p \geq 4$ (*Appendix 6—figure 8*) to define regions of no selection that we compared with other genomic regions (e.g. *Figure 5*).

We used `OmegaPlus` (*Alachiotis et al., 2012*) and `RAiSD` (*Alachiotis and Pavlidis, 2018*) scanning for selective sweeps genome wide. Both methods use local linkage disequilibrium to detect sweeps (*Nielsen, 2005*) and in addition `RAiSD` uses a local reduction in levels of polymorphism and shifts in the frequencies of low- and high-frequency-derived alleles affecting, respectively, the left and righ tails of the site-frequency spectrum.

## Methods for analyzing coalescent models

This section describes the model-fitting procedure we used for each family of models discussed in Appendix 1. Where possible, we have resorted to documented state-of-the-art simulators and inference packages, though that was not possible in all cases, particularly for the Durrett–Schweinsberg model. A description of various terms is given in *Appendix 7—table 5*. All custom code has been made available via GitHub, with links below.

### Kingman coalescent

There are numerous, well-documented packages for inferring population size profiles from whole-genome data under the Kingman coalescent, typically relying on the sequentially Markovian coalescent approximation (*McVean and Cardin, 2005*). We used `smc++` (https://github.com/popgen-methods/smcpp; *Terhorst et al., 2016*) to produce best-fit profiles. We also used the stairway plot (https://github.com/xiaoming-liu/stairway-plot-v2; *Liu and Fu, 2015*; *Liu and Fu, 2020*; ) that use the site-frequency spectra for a model-flexible demographic inference. Both packages were installed according to their respective documentations, and run using default settings. To treat runs of homozygosity, which may represent centromeric regions, as missing, we set the flag `--missing-cutoff 10` in `smc++` runs.

### $\Xi$-Beta$(2-\alpha, \alpha)$ coalescent

At the time of writing there are no off-the-shelf inference packages capable of estimating $\alpha$ or a population size profile from whole-genome data under the $\Xi$-Beta$(2-\alpha, \alpha)$ coalescent. However, synthetic data from the model can be simulated using `msprime` (*Kelleher et al., 2016*). Hence, we fit our model using ABC, in which model fitting is accomplished by comparing summary statistics of simulated and observed data under various parameters. We used uniform priors adjusted for different situations (*Appendix 7—table 6*).

We used the logit transform of the normalized site-frequency spectrum (SFS) as our summary statistic. The `msprime` package is not well optimized for simulating multiple chromosomes, so we used chromosome 4 as our observed data. To simulate observations, we set the chromosome length to 3.5 Mb, and used respective per-site per-generation mutation and recombination probabilities of $10^{-7}$ and $10^{-8}$, respectively.

A proposed parameter combination was accepted whenever the simulated statistic was within a specified tolerance of the observed statistic. To avoid tuning the tolerance and other hyperparameters, and to focus computational effort on regions of $\Theta_B$ of good model fit automatically, we used the adaptive ABC-MCMC (Approximate Bayesian Computation Markov Chain Monter Carlo) method of *Vihola and Franks, 2020* with a target acceptance rate of 10%, which the authors recommend.

### Durrett–Schweinsberg coalescent

To our knowledge, there are no off-the-shelf inference packages for the Durrett–Schweinsberg model, and also no packages for simulating it. Hence we implemented a basic, single locus simulator in `C++`, based on the exact rejection sampling mechanism which is used in both the `msprime` and `Beta-Xi-Sim` simulation packages (see the Appendix in *Koskela, 2018*). Since the Durrett–Schweinsberg coalescent is a single locus model, we computed an observed site-frequency spectra separately for 25 kb lengths of genome separated by 500 kb gaps. This was done across all 19 non-inversion chromosomes, and the mean of the resulting ensemble was taken to be the observed SFS. Simulated values were calculated as the mean of 10,000 independent, single-locus replicates. This number was

found to be high enough in trial runs to avoid zero entries in the averaged SFS, and hence infinite values in the logit transform.

Then we used the same ABC-MCMC pipeline outlined above for the $\Xi$-Beta$(2 - \alpha, \alpha)$ coalescent to infer an approximate posterior distribution of values for the compound parameter $c$ of the Durrett–Schweinsberg model.

### Computations

The computations in this paper were run on the Odyssey cluster supported by the FAS Division of Science, Research Computing Group at Harvard University. Some computations were run on the Mimir bioinformatics server and the Assa bioinformatics computer at the University of Iceland.

### Code availability

Simulations of background selection were done with SLiM 3 (**Haller and Messer, 2019**) available at https://messerlab.org/slim/. Estimates of population size histories for the Kingman coalescent were produced using the stairwayplot (**Liu and Fu, 2015**; **Liu, 2020**) and smc++ (**Terhorst et al., 2016**) available via Github at https://github.com/xiaoming-liu/stairway-plot-v2 and https://github.com/popgenmethods/smcpp, respectively. Based on the estimated population size histories site-frequency spectra under the Kingman and the $\Xi$-Beta$(2 - \alpha, \alpha)$ coalescents were simulated using msprime, available via GitHub at https://github.com/tskit-dev/msprime, (copy archived at swh:1:rev:-becc7b948123f8683c49ed41480ca2682d979a7f; **Wong, 2022**), with documentation at https://tskit.dev/msprime/docs/stable/. Our msprime simulations also make use of the tskit library, available via GitHub at https://github.com/tskit-dev/tskit, (copy archived at swh:1:rev:575daea4bcd535df7b-c328a7387876eb986daebb; **Jeffery, 2023**), with documentation at https://tskit.dev/tskit/docs/. To our knowledge, no prior implementation of the Durrett–Schweinsberg coalescent is available. Hence, we wrote a simulator, which is available via GitHub at https://github.com/JereKoskela/ds-tree, (copy archived at swh:1:rev:7ee7d9c473278aaf618af7a539fd3cba2735d1e1; **Koskela, 2022**). This repository also contains documentation of the Durrett–Schweinsberg implementation, as well as Python and shell scripts for the (1) ABC pipelines we used to conduct model fitting for both the $\Xi$-Beta$(2 - \alpha, \alpha)$ and Durrett–Schweinsberg coalescents, and (2) the simulation pipelines for sampling site-frequency spectra under the best-fit Kingman, $\Xi$-Beta$(2 - \alpha, \alpha)$, and Durrett–Schweinsberg coalescents. C++code and python scripts implementing the sampling schemes described in https://github.com/eldonb/coalescents; **Eldon, 2021a**. C code using recursions **Birkner et al., 2013b** for computing the exact expected branch length spectrum for Examples 2.3 and 2.4 of the Durrett–Schweinsberg model (**Durrett and Schweinsberg, 2005**) is available at https://github.com/eldonb/Durrett_Schweinsberg_Expected_SFS, (copy archived at swh:1:rev:07a534d2d6b5870762bfe6dd3c79f860eb82494a; **Eldon, 2021b**).

## Acknowledgements

We thank John Wakeley, Fabian Freund, Pierre-Alexandre Gagnaire, and anonymous reviewers for critical comments on the manuscript and Kristján Kristinsson and the Icelandic Marine Research Institute for help in sampling. We acknowledge technical help with the molecular analysis by The Bauer Core Facility at Harvard University. Funding: The work was supported by an Icelandic Research Fund Grant of Excellence no. 185151-051 to EÁ, KH, Alison Etheridge, Wolfgang Stephan, and BE. BE also acknowledges financial support by the Deutsche Forschungsgemeinschaft (DFG, German Research Foundation) Project number 273887127 through DFG grant STE 325/17 to Wolfgang Stephan through DFG Priority Program (SPP) 1819: Rapid evolutionary adaptation, a DFG SPP1819 start-up module grant to JK, Maite Wilke Berenguer, and BE, and JK acknowledges financial support from Engineering and Physical Sciences Research Council (EPSRC) grants EP/R044732/1 and EP/V049208/1.

## Additional information

### Funding

| Funder | Grant reference number | Author |
|---|---|---|
| Rannsóknasjóður. Rannís (Rannsóknamiðstöð Íslands) | 185151-051 | Einar Árnason Katrín Halldórsdóttir Bjarki Eldon |
| Deutsche Forschungsgemeinschaft | 273887127 STE 325/17 | Bjarki Eldon |
| Engineering and Physical Sciences Research Council | EP/R044732/1 and EP/V049208/1 | Jere Koskela |
| Deutsche Forschungsgemeinschaft | DFG Priority Program (SPP) 1819: Rapid evolutionary adaptation and DFG SPP1819 start-up module grant | Bjarki Eldon |

The funders had no role in study design, data collection, and interpretation, or the decision to submit the work for publication.

### Author contributions

Einar Árnason, Katrín Halldórsdóttir, Conceptualization, Resources, Data curation, Formal analysis, Supervision, Funding acquisition, Validation, Investigation, Visualization, Methodology, Writing – original draft, Project administration, Writing – review and editing; Jere Koskela, Conceptualization, Software, Formal analysis, Funding acquisition, Validation, Investigation, Visualization, Methodology, Writing – original draft, Writing – review and editing; Bjarki Eldon, Conceptualization, Resources, Software, Formal analysis, Supervision, Funding acquisition, Validation, Investigation, Visualization, Methodology, Writing – original draft, Project administration, Writing – review and editing

### Author ORCIDs

Einar Árnason http://orcid.org/0000-0003-3686-6407
Jere Koskela http://orcid.org/0000-0002-2836-8777
Katrín Halldórsdóttir http://orcid.org/0000-0002-9682-0286
Bjarki Eldon http://orcid.org/0000-0001-9354-2391

### Decision letter and Author response

Decision letter https://doi.org/10.7554/eLife.80781.sa1
Author response https://doi.org/10.7554/eLife.80781.sa2

## Additional files

### Supplementary files
• MDAR checklist

### Data availability

All data needed to evaluate the conclusions of the paper are presented in the paper, and/or the supplementary materials. The bam files of the whole-genome sequencing of each individual aligned to the Gadmor3 reference genome (NCBI accession ID: GCF_902167405.1) are available from the NCBI SRA Sequence Read Archive under accession number BioProject ID: PRJNA663624 at the time of publication.

The following datasets were generated:

| Author(s) | Year | Dataset title | Dataset URL | Database and Identifier |
|---|---|---|---|---|
| Arnason E, Koskela J, Halldórsdóttir K, Eldon B | 2022 | Sweepstakes reproductive success via pervasive and recurrent selective sweeps | https://dx.doi.org/10.5061/dryad.bcc2fqzgx | Dryad Digital Repository, 10.5061/dryad.bcc2fqzgx |
| Halldórsdóttir K, Árnason E | 2020 | Genomic coalescent-based evidence of sweepstakes reproduction in Atlantic cod | https://www.ncbi.nlm.nih.gov/sra/PRJNA663624 | NCBI Sequence Read Archive, PRJNA663624 |

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

## Appendix 1

## Coalescent models

Our aim is to distinguish between three possible explanations of the pattern of genetic diversity observed in our samples:

1. Reproduction with a low variance offspring distribution is consistent with the classic Wright–Fisher model of genetic evolution, albeit in a population of varying size (the Kingman coalescent with variable demography).
2. Neutral and random sweepstakes reproduction arising from a skewed family size distribution (the neutral $\Xi$-Beta$(2 - \alpha, \alpha)$ coalescent) in a population undergoing exponential growth.
3. Recurrent selective sweeps (the Durrett–Schweinsberg coalescent of recurrent selective sweeps) as models of selective sweepstakes.

All three scenarios predict an excess of singletons in the site-frequency spectrum relative to a standard Kingman coalescent (at least as long as the population size decreases backward in time in scenario 1) and are hence a priori plausible explanation of our data. However, the models differ in their predictions for the rest of the site-frequency spectra given a desired singleton class (*Appendix 6—figure 21*).

In this section, we introduce a class of models to describe each scenario. In the methods section for analyzing coalescent models (Methods for analyzing coalescent models) we fit each family to our data and use the discrepancy between the data and the corresponding prediction of the best-fit model within each class to assess the degree to which each model class can explain the data.

## Appendix 2

### The Kingman coalescent

The Kingman coalescent (*Kingman, 1982*) is a seminal model in population genetics and the default null model for genetic evolution. In brief, the distribution of the genealogical tree connecting $n$ sampled lineages is obtained by merging each pair of lineages at a unit rate, until only one lineage remains.

The Kingman coalescent is derived from individual-based models with a finite population size $N$ by suitably rescaling time and sending $N \to \infty$. To connect a merger time in the Kingman coalescent back to generations, it is necessary to undo the rescaling. The most common choice of individual-based models is the Wright–Fisher model (*Durrett, 2008*), under which one unit of coalescent time corresponds to $N$ generations.

The original formulation of the coalescent assumed a constant population size, but extensions to variable population sizes, resulting in variable pair-merger rates, have subsequently been developed (see e.g. *Donnelly and Tavaré, 1995*). There are many robust and well-established methods for flexibly inferring historical population size profiles from DNA sequence data. Hence, rather than specifying an explicit model for population size under our Kingman coalescent scenario, we use the model specifications to coincide with the implementation of our model-fitting pipeline defined in the methods section for analyzing coalescent models (Methods for analyzing coalescent models).

## Appendix 3

### The $\Xi$-Beta($2 - \alpha, \alpha$) coalescent

The Beta-coalescent (or Beta($2 - \alpha, \alpha$)-coalescent) is a genealogical model arising from a haploid population of fixed size, in which individuals can have family sizes comparable to the total population size with non-negligible probability (*Schweinsberg, 2003*). Large families looking forward in time result in multiple branches merging simultaneously looking backward in time. Specifically, when there are $n$ lineages, each subset of size $n$ lineages merges at a rate

$$\lambda_{n,k}^{(B)} := \int_0^1 x^{k-2}(1-x)^{n-k}\frac{x^{1-\alpha}(1-x)^{\alpha-1}}{B(2-\alpha,\alpha)}dx, \tag{1}$$

where $B(y, z)$ is the beta function.

The extension to diploid population results in a coalescent with up to four simultaneous merger events, arising out of the $2 \leq k \leq n$ merging lineages separating uniformly at random into the four available parental chromosomes (*Birkner et al., 2013a*; *Koskela and Wilke Berenguer, 2019*). The resulting coalescent with simultaneous multiple mergers is known as the $\Xi$-Beta($2 - \alpha, \alpha$) coalescent (*Schweinsberg, 2000*). Extensions to variable population sizes have also been constructed (*Freund, 2020*; *Koskela and Wilke Berenguer, 2019*).

To balance flexibility with computational constraints, we focus on the two-dimensional family of models

$$\Theta_B := \{(\alpha, \beta) \in (1, 2) \times [0, \infty)\},$$

where $\alpha$ determines the skewness of the offspring distribution in (1), and $\beta$ denotes a rate of exponential growth of the population forward in time.

Beta $(2 - \alpha, \alpha)$ and $\Xi$-Beta$(2 - \alpha, \alpha)$ coalescents are also obtained as scaling limits of individual-based models described in *Schweinsberg, 2003*, but unlike the Kingman coalescent, one unit of time in the limiting coalescent model corresponds to $O(N^{\alpha-1})$ generations. This results in a very short timescale when $\alpha$ is close to 1, and can lead to implausibly large predicted population sizes (or, equivalently, mutation rates) in order to match observed levels of diversity. We will sidestep this issue by resorting only to statistics that are insensitive to the total number of mutations in our model-fitting pipeline.

## Appendix 4

### The Durrett–Schweinsberg coalescent

We model the impact of recurrent selective sweeps with the *Durrett and Schweinsberg, 2005* coalescent. The model describes genetic ancestry at a single locus in a haploid population subject to hard selective sweeps due to beneficial mutations arising along the genome. The neutral locus of interest that is linked to a selected site can either merge due to a sweep or escape via recombination. The pattern of mergers and escapes in each of the sweeps results in multiple mergers, where any $2 \leq k \leq n$ of $n$ lineages merge at rate

$$\lambda_{n,k}^{(DS)} := \mathbb{I}_{\{k=2\}} + \frac{\delta s^2}{\gamma} \int_0^1 x^{k-2}(1-x)^{n-k}\frac{x}{1/2}\,\mathrm{d}x, \tag{2}$$

where $\mathbb{I}_A = 1$ if the event $A$ happens and zero otherwise, $\delta \geq 0$ is the population-rescaled rate at which beneficial mutations arise, $s \in [0, 1]$ is a measure of the advantage they provide (a larger value corresponds to a greater fitness advantage), and $\gamma > 0$ is the population-rescaled rate of recombination per link between sites. Specifically, this is the infinite-chromosome model with uniformly distributed locations of both mutations and recombinations described in Example 2.4 of *Durrett and Schweinsberg, 2005*.

Extensions of the Durrett–Schweinsberg coalescent to diploid populations or nonconstant population sizes have not been derived. Because the Durrett–Schweinsberg model features hard selective sweeps and selective advantage manifests on each chromosome individually, we do not expect that a diploid extension would give rise to lineages separating into four groups in each merger, as it did in the $\Xi$-Beta$(2 - \alpha, \alpha)$ coalescent. However, both diploidy and a nonconstant population size likely would affect the merger rates in (2). Development of these generalizations is beyond the scope of this article, and in their absence, we resort to the haploid model with constant population size as a practical, heuristic model.

The timescale of the Durrett–Schweinsberg model coincides with that of the Kingman coalescent. One unit of coalescent time corresponds to $N$ generations regardless of the value of parameters. The duration of a selective sweep is much shorter at $\log N$ generations, which is why they become instantaneous in the coalescent limit, and hence also cannot overlap. However, the Durrett–Schweinsberg model requires very rapid recombination, with a per-generation recombination probability $r_N \sim \rho/\log N$ for a constant $\rho \in [0, \infty)$, unlike the more familiar $r_N \sim \rho/N$ under a Kingman coalescent.

Note that (2) coincides for any two parameter combinations with $\delta'(s')^2/\gamma' = \delta s^2/\gamma$. It is also possible to draw a random selective advantage $S$ independently at each sweep from a fixed distribution, and obtain the same coalescent model as long as $\delta'(s')^2/\gamma' = \delta\mathbb{E}[S^2]/\gamma$. Hence, we define the compound parameter $c := \delta\mathbb{E}[S^2]/\gamma$ and focus on the one-dimensional family of models given by

$$\Theta_{DS} := \{c \in [0, \infty)\}.$$

The derivation of the model in *Durrett and Schweinsberg, 2005* is only valid for a single locus. Genomic generalization is likely to result in a more complex model depending on separations between sites of interest, as well as potential mutation or recombination points. We expect the methodology we develop in the methods section for analyzing coalescent models (Methods for analyzing coalescent models) to be robust to the inconsistency of applying a single-locus model to whole-genome data as it depends only on expected frequencies of segregating sites, rather than their higher moments or joint distributions. To facilitate a comparison of observed and predicted linkage patterns (*Appendix 6—figure 6*), we present the following two-locus generalization of the Durrett–Schweinsberg coalescent, applicable to samples of two haplotypes on which the first site is at position 0, and the second at position $d + 1$.

Because the recombination in the Durrett–Schweinsberg model acts on a timescale of $O(\log N)$ generations while the timescale of evolution is $O(N)$ generations, we assume that the two sites are unlinked between sweeps. When a sweep happens, four things can happen: (1) there are no mergers, (2) there is a merger at site 1 only, (3) there is a merger at site 2 only, or (4) there is a simultaneous pair merger at each site. We obtain the rate of simultaneous pair mergers by adapting

the calculation in *Durrett and Schweinsberg, 2005*, Example 2.4 to consider the rate with which four out of four lineages take part in a merger:

$$\eta([y, 1]) = \delta s \int_{-\infty}^{\infty} \mathbb{1}_{\{e^{-\gamma|x|/s} \geq y\}} \mathbb{1}_{\{e^{-\gamma|x-d|/s} \geq y\}} \mathrm{d}x, \tag{3}$$

where $\eta$ is defined in *Durrett and Schweinsberg, 2005*, Theorem 2.2. The integrand is positive on the interval $\left( d - \frac{-s \log y}{\gamma}, \frac{-s \log y}{\gamma} \right)$,

whereupon (*Equation 3*) is positive whenever $y \leq \exp(-d\gamma/(2s))$. Hence

$$\frac{\mathrm{d}}{\mathrm{d}y} \eta([y, 1]) = \frac{-2\delta s^2}{\gamma y} \mathbb{1}_{\{y \leq \exp(-d\gamma/(2s))\}},$$

so that sweeps which cause a simultaneous pair merger at both sites, in which all four lineages participate, rise at rate

$$\int_0^{\exp(-\gamma d/(2s))} y^{4-2} cy \mathrm{d}y = \frac{c}{4} \exp\left(\frac{-2\gamma d}{s}\right).$$

Since the marginal rate of mergers at each site must be $c/2$, the rate of sweeps which cause a merger at only one particular site is

$$\frac{c}{2} - \frac{c}{4} \exp\left(\frac{-2\gamma d}{s}\right).$$

Ancestral trees with these rates of single and double pair-mergers were simulated, and corresponding linkage disequilibrium patterns were estimated from $n$ replicates via

$$\widehat{\mathrm{LD}} := \frac{\sum_{i=1}^n e^{-\theta(T_1^i + T_2^i)} - n^{-1} \left( \sum_{i=1}^n e^{-\theta T_1^i} \right) \left( \sum_{i=1}^n e^{-\theta T_2^i} \right)}{\left( \sum_{i=1}^n (1 - e^{-\theta T_1^i}) e^{-\theta T_1^i} \right)^{1/2} \left( \sum_{i=1}^n (1 - e^{-\theta T_2^i}) e^{-\theta T_2^i} \right)^{1/2}}, \tag{4}$$

where $T_1^i$ and $T_2^i$ are the estimated TMRCAs at sites 1 and 2 from the ith replicate, and $\theta/2$ is a population-rescaled rate of neutral mutation. The right-hand side of (4) is obtained by computing the correlation in the events in which no mutation separates the two samples at each site, and substituting ensemble estimates for the resulting mean TMRCAs.

The model constructed above has three free parameters: $c$, $\gamma/s$, and $\theta/2$, but predicts an exponential decay to zero of LD as the separation $d$ grows, regardless of the values of parameters as long as they are nonzero. This contradicts the observed positive background level in *Appendix 6—figure 6*. To explain a positive level of background LD, we assumed that a fraction $a \in [0, 1]$ of sweeps were not localized to a mutation at a given position along the chromosome, but rather would cause a sweep along the full length of the chromosome regardless of recombination. A possible interpretation for such a 'sweep' is a population bottleneck. The resulting rate of a simultaneous double pair-merger is

$$\frac{c}{4} \left( a + (1 - a) \exp\left(\frac{-2\gamma d}{s}\right) \right),$$

with a corresponding change in the rate of single-site pair mergers. *Appendix 6—figure 6* shows that the observed linkage data are informative about $a$, but not the other parameters and that the resulting model predictions are not inconsistent with the observed LD profile.

## Appendix 5

### Limits of the models

This section highlights some key ways in which the Durrett–Schweinsberg and $\Xi$-Beta$(2 - \alpha, \alpha)$ coalescent families differ from the classical and well-known Kingman coalescent, as well as from each other.

The Kingman and $\Xi$-Beta$(2 - \alpha, \alpha)$ coalescent families we have considered are models for genome-scale, diploid organisms. In contrast, the Durrett–Schweinsberg coalescent models a haploid (since the chromosomes are treated as separate 'individuals') organism undergoing selective sweeps at unobserved, linked sites some recombinational distance away from the neutral site for which we are tracing the genealogy.

The fact that the model provides such a good fit to our data with only one free parameter despite this discrepancy is a strong motivation for the construction of an analogous model for multi-locus data from a diploid organism, which is not presently available in the literature. We expect such an extension to further improve the model fit, and also very likely render identifiable the various components of the compound parameter $c = \delta\mathbb{E}[S^2]/\gamma$ which cannot be inferred separately using the single-locus Durrett–Schweinsberg coalescent. Because multiple mergers in the Durrett–Schweinsberg coalescent arise from hard selective sweeps which encompass the whole population, we do not anticipate that a diploid extension will simply split multiple mergers into four groups (corresponding to the four ancestral chromosomes available in a reproduction event) as it does in the case of the $\Xi$-Beta$(2 - \alpha, \alpha)$ coalescent (**Birkner et al., 2013a**). In a similar vein, the impact of varying population size on measures of genetic diversity under the Durrett–Schweinsberg coalescent is unknown.

Typical constructions of the Kingman coalescent from a Wright–Fisher population predict that the number of generations between merger events is proportional to the (census) population size $N$. In contrast, the $\Xi$-Beta$(2 - \alpha, \alpha)$ coalescent predicts a timescale proportional to $N^{\alpha-1}$ generations per coalescent time unit (**Schweinsberg, 2003**). With an $\alpha \approx 1$ under the $\Xi$-Beta $(2 - \alpha, \alpha)$ coalescent (as observed), the only way to match the best-fit regime of the $\Xi$-Beta $(2 - \alpha, \alpha)$ coalescent to our data is to postulate a census population size many orders of magnitude larger than the effective size (which under the model is proportional to $N^{\alpha-1}$).

In contrast, the timescale of evolution under the Durrett–Schweinsberg coalescent coincides with that of the Kingman coalescent (i.e. on the order of $N$ discrete time units per coalescent time unit). Selective sweeps under the Durrett–Schweinsberg coalescent govern the rate with which a mutation spreads in a population (proportional to $\log(N)$ time units on average), but sweeps are independent of the timescale on which new favourable mutations arise relative to the Kingman coalescent.

In the $\Xi$-Beta$(2 - \alpha, \alpha)$ coalescent derived for a diploid population, a multiple-merger is due to a single pair of diploid individuals (**Birkner et al., 2018**), picked at random, giving rise to a large family, and the single pair becomes ancestral to a non-negligible fraction of the population in one generation. The offspring of the highly fecund successful pair carry no fitness advantage. In the Durrett–Schweinsberg coalescent, the positive mutation, which initiates a selective sweep, hits a random individual that passes on a fitness advantage to its offspring, and with a certain probability the beneficial mutation sweeps to fixation in $\log N$ generation on average. The probability of a second sweep beginning before the first is complete is proportional to $\log(N)/N$, and hence vanishes as $N \to \infty$. After a sweep ends, a new individual carrying the beneficial type can initiate a new, independent sweep. The recurrent sweeps imply that the environment is forever changing with each sweep climbing a new selective peak in the adaptive landscape. To the best of our knowledge, our work is the first attempt to use whole-genome data to distinguish between families of multiple-merger coalescents to assess the plausibility of their underlying assumptions as explanations for observed data in diploid taxa.

Finally, the Durrett–Schweinsberg coalescent assumes that recombination occurs at approximately $\log N$ time unit intervals, much faster than mutations that arise on a timescale of $N$ time units. Thus, the Durrett–Schweinsberg coalescent predicts a rapid decay of linkage disequilibrium as observed (**Appendix 6—figure 6**). Under the Kingman coalescent, both mutation and recombination act on a timescale of $N$ units, while both act on a timescale of $N^{\alpha-1}$ units under the $\Xi$-Beta$(2 - \alpha, \alpha)$ coalescent. Hence, the Durrett–Schweinsberg model predicts widespread recombination relative

to mutations, which is qualitatively consistent with the very rapid decay of linkage disequilibrium (*Appendix 6—figure 6*).

The Durrett–Schweinsberg model is based on the *Moran, 1958* model of reproduction in which a single individual replicates and another dies instead. The reproducing individual remains active in the population. In the Durrett–Schweinsberg addition, the reproducing individual is fitter. On a coalescent timescale, there is a burst of reproductive activity by the individual and his descendants such that their fit, derived lineage quickly takes over the whole population. The Moran model does not model high fecundity by itself. Still, with this addition, it is as if the Moran process and its associated Kingman coalescent are interrupted by a burst of high fecundity. Thus the Durrett–Schweinsberg model approximates selective sweepstakes by adding recurrent sweeps of a new selectively advantageous mutation each time to the Moran model. The multiple-merger Durrett–Schweinsberg coalescent (Appendix 4) describes the genealogy at a single site, the 'neutral' site, that is linked at some recombinational distance to a site hit by a favorable mutation. The population experiences recurrent, strongly beneficial mutations at sites linked to the neutral site, and it is assumed that the neutral site never experiences mutation. A beneficial mutation sweeps to fixation in $\log N$ time units, where $2N$ is the population size, and the probability of fixation does not depend on the population size. However, a vital component of the Durrett–Schweinsberg model is the assumption of an elevated rate of recombination between the neutral and the mutated site, giving ancestral lineages at the neutral site a chance to escape a sweep via recombination.

## Appendix 6

## Supplementary figures

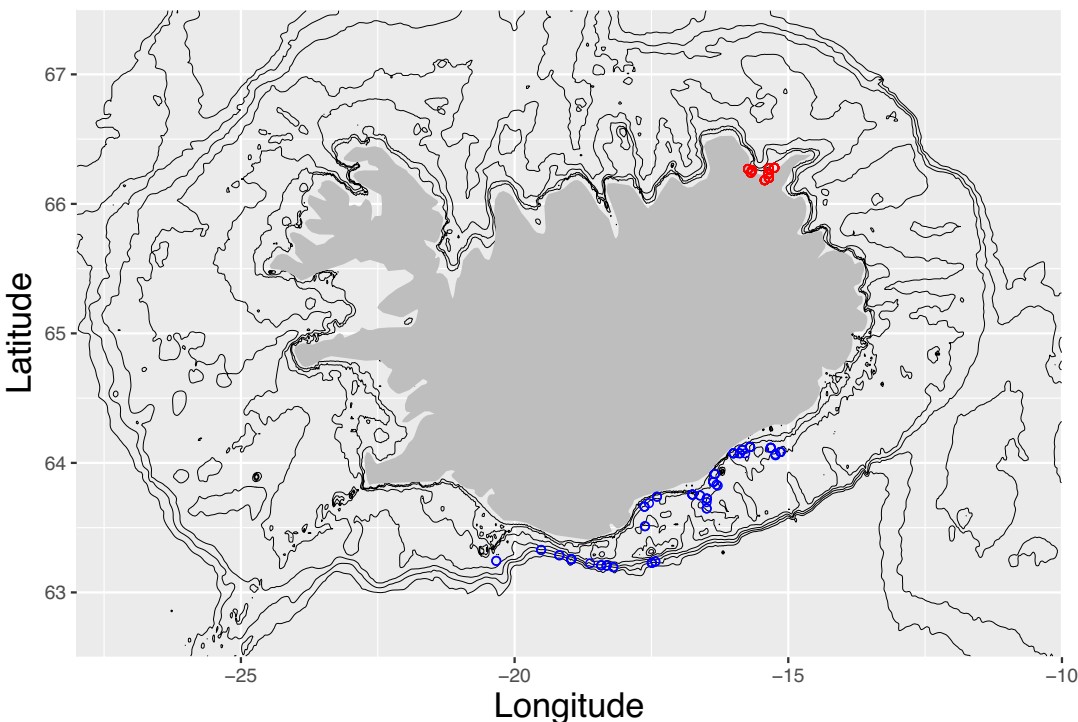

**Appendix 6—figure 1.** Sampling localities at Iceland. Sampling localities ranging from Vestmannaeyjar to Höfn on the south and south-east coast (blue circles, $n = 68$) and Þistilfjörður in the north-east (red circles, $n = 71$) on a map of Iceland. Depth contours are at −25, −50, −100, −200, −400, −600, and −800 m. The two localities serve as statistical replicates, the South/south-east and the Þistilfjörður population, respectively.

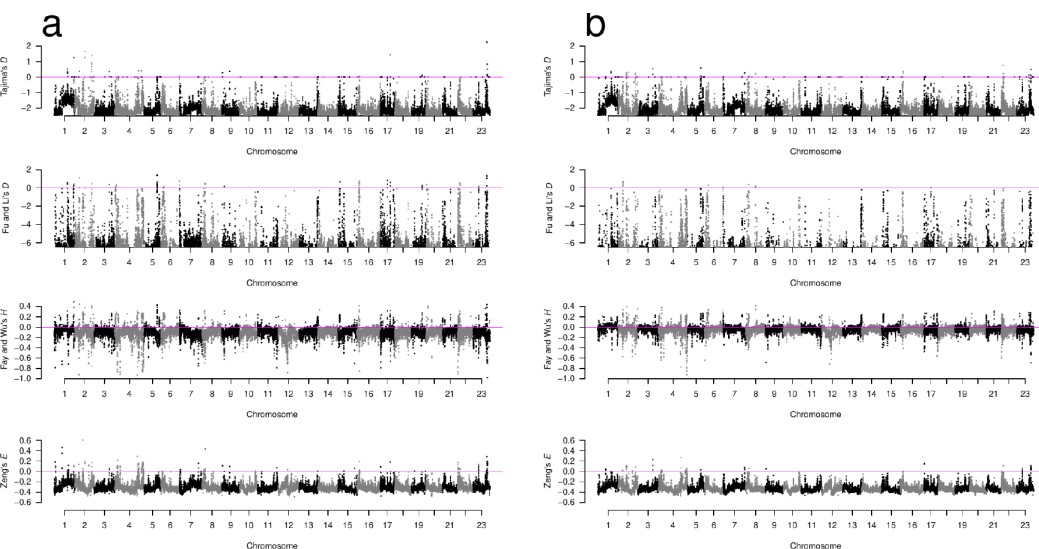

**Appendix 6—figure 2.** Neutrality test statistics in sliding windows across all chromosomes for GL2 estimates. (**a, b**) Manhattan plots of Tajima's *D* (***Tajima, 1989***), Fu and Li's *D* (***Fu and Li, 1993***), Fay and Wu's *H* (***Fay and Wu, 2000***), and Zeng's *E* (***Zeng et al., 2006***) for the South/south-east population and the Þistilfjörður population, respectively. Sliding window estimates (window size 100 kb with 20 kb step size) using GL2 genotype likelihoods. Value of statistic under Kingman coalescent neutrality equilibrium indicated with magenta horizontal line.

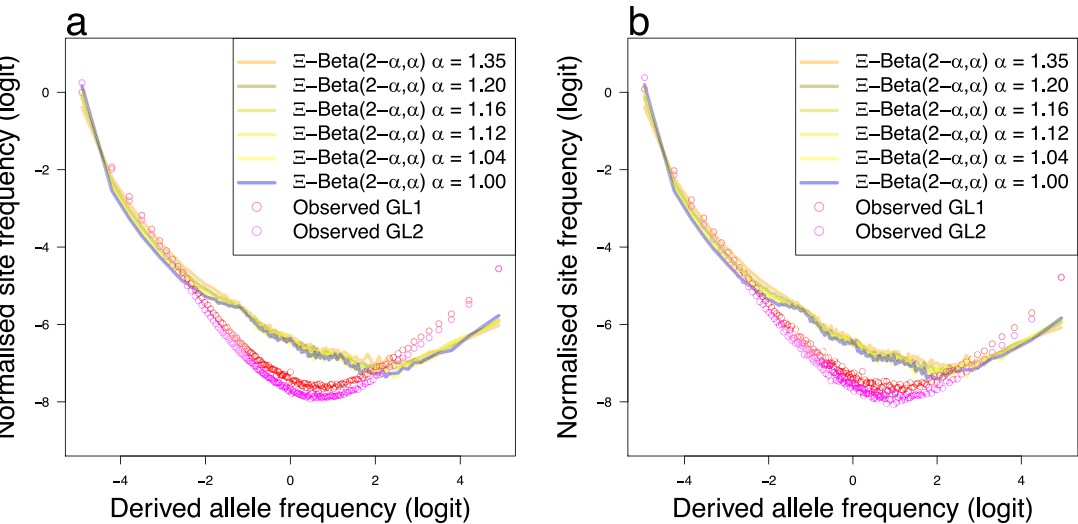

**Appendix 6—figure 3.** The random sweepstakes model. (**a, b**) Observed site-frequency spectra of non-inversion chromosomes and expectations of the $\Xi$-Beta$(2 - \alpha, \alpha)$ coalescent (the random sweepstakes model) for the South/south-east population (sample size $n = 68$) and Þistilfjörður population (sample size $n = 71$), respectively. The observed mean site-frequency spectrum of the non-inversion chromosomes 3–6, 8–11, and 13–23 polarized with Gma as outgroup and estimated under genotype likelihoods GL1 and GL2 and expected site-frequency spectrum of the $\Xi$-Beta$(2 - \alpha, \alpha)$ coalescent with $\alpha = 1.35$, $\alpha = 1.20$, $\alpha = 1.16$, $\alpha = 1.12$, $\alpha = 1.04$, and $\alpha = 1.00$ which are representative samples of the posterior estimates that coincide with the kernel density estimates (*Figure 2*). The $\alpha = 1.16$, $\alpha = 1.12$, and $\alpha = 1.04$, which represent the approximate maximum likelihood best estimates as detailed in *Figure 4a, b* and *Figure 4—figure supplement 1a, b*.

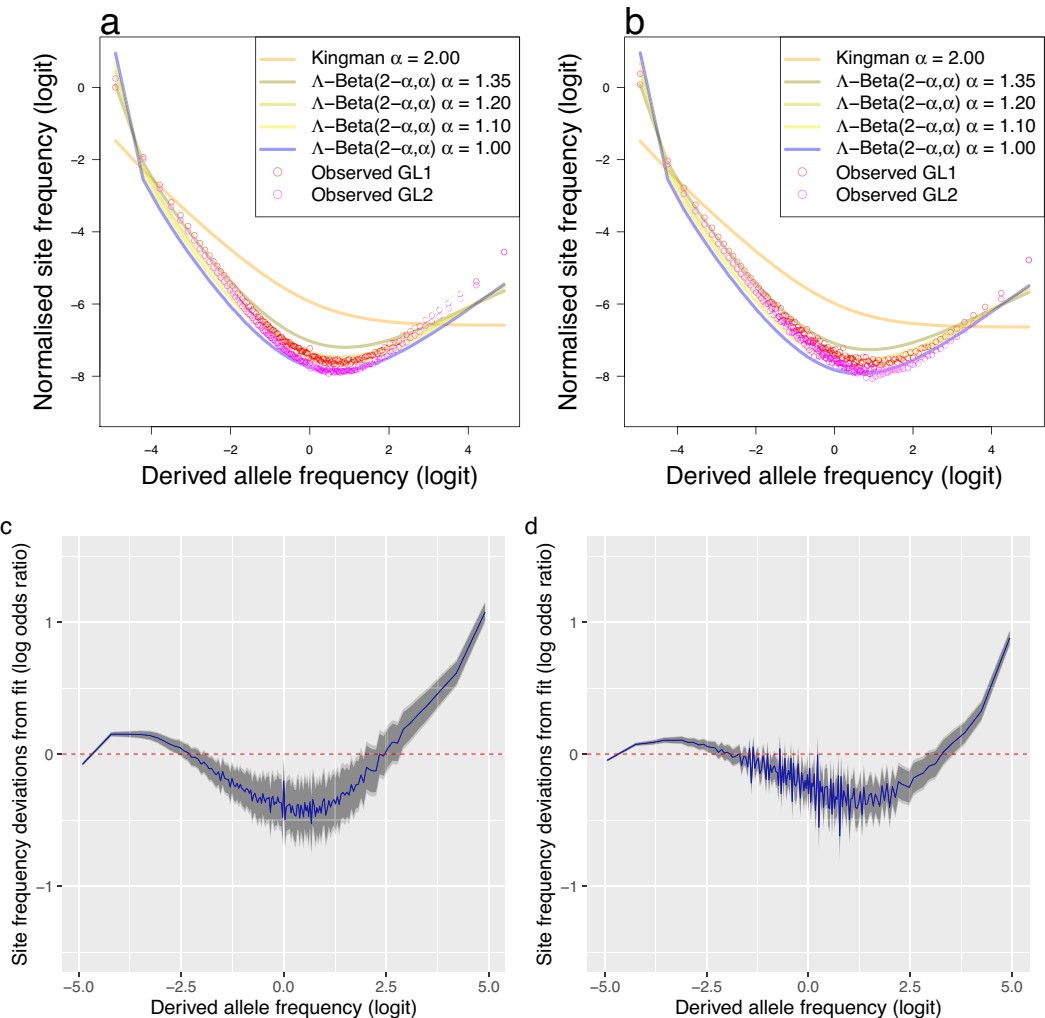

**Appendix 6—figure 4.** Piecewise comparison of expectations of the $\Lambda$-Beta$(2 - \alpha, \alpha)$ coalescent and deviations from fit. The observed mean site-frequency spectrum of the non-inversion chromosomes 3–6, 8–11, and 13–23 polarized with Gma as outgroup and estimated under genotype likelihoods GL1 and GL2 and expected site-frequency spectrum of the $\Lambda$-Beta$(2 - \alpha, \alpha)$ coalescent with $\alpha = 1.35$, $\alpha = 1.20$, $\alpha = 1.10$, and $\alpha = 1.00$. Population South/south-east (sample size $n = 68$) (**a**) and population Þistilfjörður (sample size $n = 71$) (**b**). Deviations from the maximum likelihood estimated expecations of $\Lambda$-Beta$(2 - \alpha, \alpha)$ coalescent for the South/south-east (**c**) and the Þistilfjörður population (**d**).

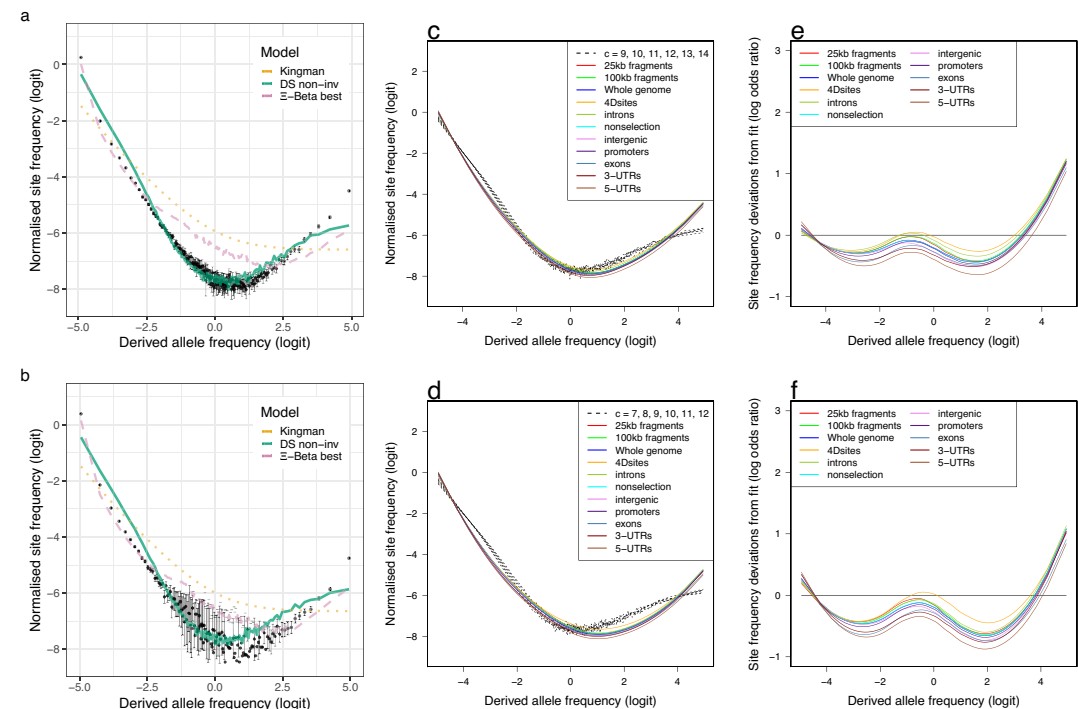

**Appendix 6—figure 5.** Fit to the selective sweepstakes model for GL2 estimated site-frequency spectra. (**a, b**) Mean observed site-frequency spectra for the 19 non-inversion chromosomes combined estimated with GL2 likelihood for the South/south-east (sample size $n = 68$) and Þistilfjörður populations (sample size $n = 71$), respectively. Error bars of observed data are ±2 standard deviations of the bootstrap distribution with 100 bootstrap replicates. Expected site-frequency spectra are the Kingman coalescent (the no sweepstakes model), the best approximate maximum likelihood estimates (**Eldon et al., 2015**) of the Ξ-Beta model (the random sweepstakes model), and the Durrett–Schweinsberg coalescent (DS) (the selective sweepstakes model) approximate Bayesian computation (ABC) estimated for the non-inversion chromosomes (non-inv). (**c, d**) The observed site-frequency spectra of different sized fragments and various functional classes compared to expectations of the Durrett–Schweinsberg coalescent (DS) ABC estimated for the non-inversion chromosomes for the South/south-east population and the Þistilfjörður population, respectively. The compound parameter $c$ ranges from 7 to 14. The different functional groups are fourfold degenerate sites (Dsites), intronic sites, non-selection sites (sites more than 500 kb away from peaks of selection scan, **Appendix 6—figure 8**), intergenic sites, promoters, exons, $3'$-UTR sites (3-UTRs), and $5'$-UTR (5-UTRs) sites in order of selective constraints. (**e, f**) Deviations from expectations of the Durrett–Schweinsberg model of recurrent selective sweeps of different sized fragments and functional groups for the South/south-east population and the Þistilfjörður population, respectively.

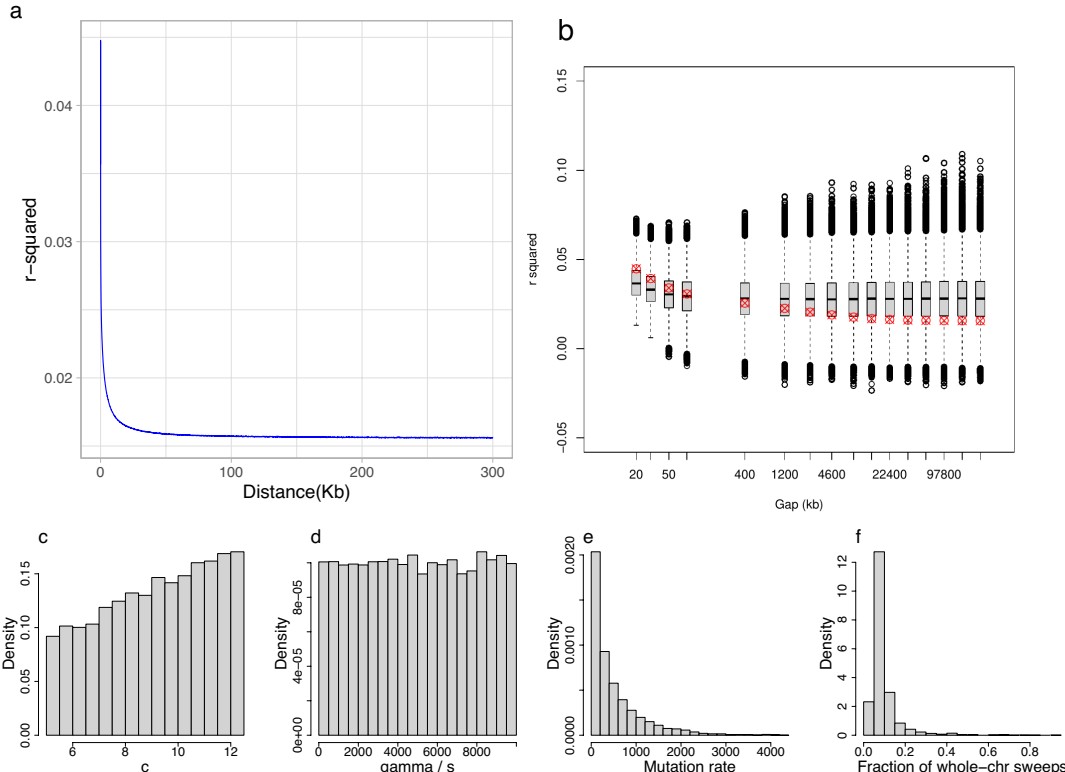

**Appendix 6—figure 6.** Decay of linkage disequilibrium with distance: observed and under an extension of the Durrett–Schweinsberg model. (**a**) Observed linkage disequilibrium (LD), measured as $r^2$, with distance in kb (kilobase). Non-inversion chromosomes from the South/south-east population as an example. LD decays rapidly to background values. (**b**) A subset of the distances from panel **a** (red × in circles) overlaid on the simulated empirical distribution of LD profiles (boxplot) obtained from the extension of the Durrett–Schweinsberg model described in Appendix 4. (**c–f**) Posterior distributions of parameters from which panel **b** has been sampled. The $c$ parameter was constrained to lie between 5 and 12.5 to enforce consistency with the site-frequency spectrum (SFS)-based results in *Figure 3* and *Appendix 6—figure 7*, while $\gamma/s$ and $\theta$ were constrained between 0 and 10,000 to avoid transient approximate Bayesian computation (ABC)-MCMC chains.

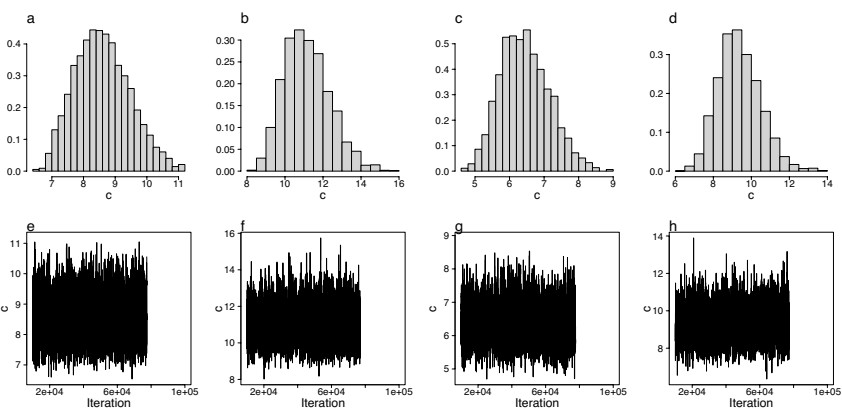

**Appendix 6—figure 7.** Approximate Bayesian computation (ABC) estimation of parameters of the Durrett–Schweinsberg coalescent (***Durrett and Schweinsberg, 2005***). (**a–d**) ABC-posterior densities of the compound parameter $c \in \Theta_{DS}$ using site-frequency spectra computed from likelihood GL1 (**a, c**) and GL2 (**b, d**) for the South/south-east and Þistilfjörður populations, respectively. (**e–h**) Corresponding trace plots demonstrating the good mixing of the ABC-MCMC.

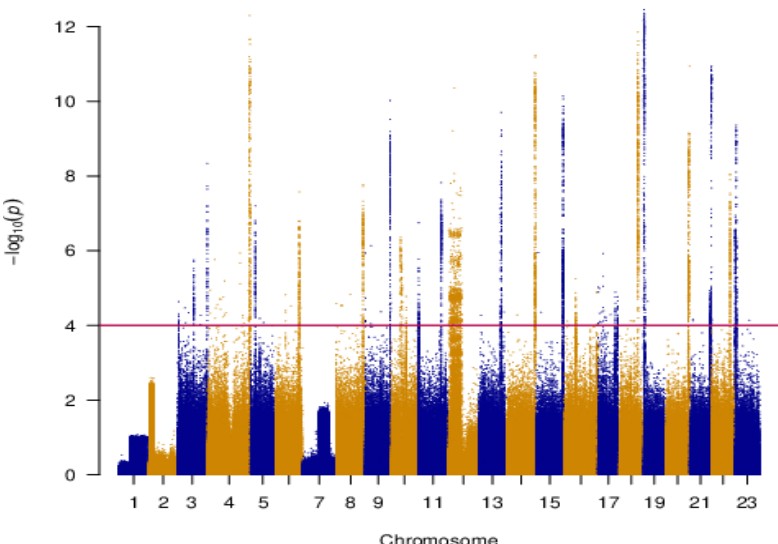

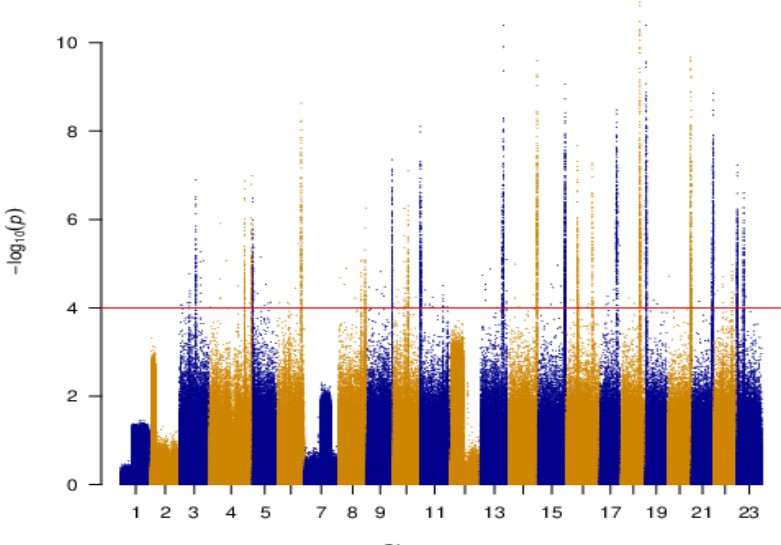

**Appendix 6—figure 8.** Principal components based genomic scan of selection for South/south-east (top) and Þistilfjörður (bottom) populations. Regions of 500 kb on either side of peaks exceeding $-\log 10p \geq 4$ were excluded to define regions of no selection for analysis in *Figures 3 and 5*.

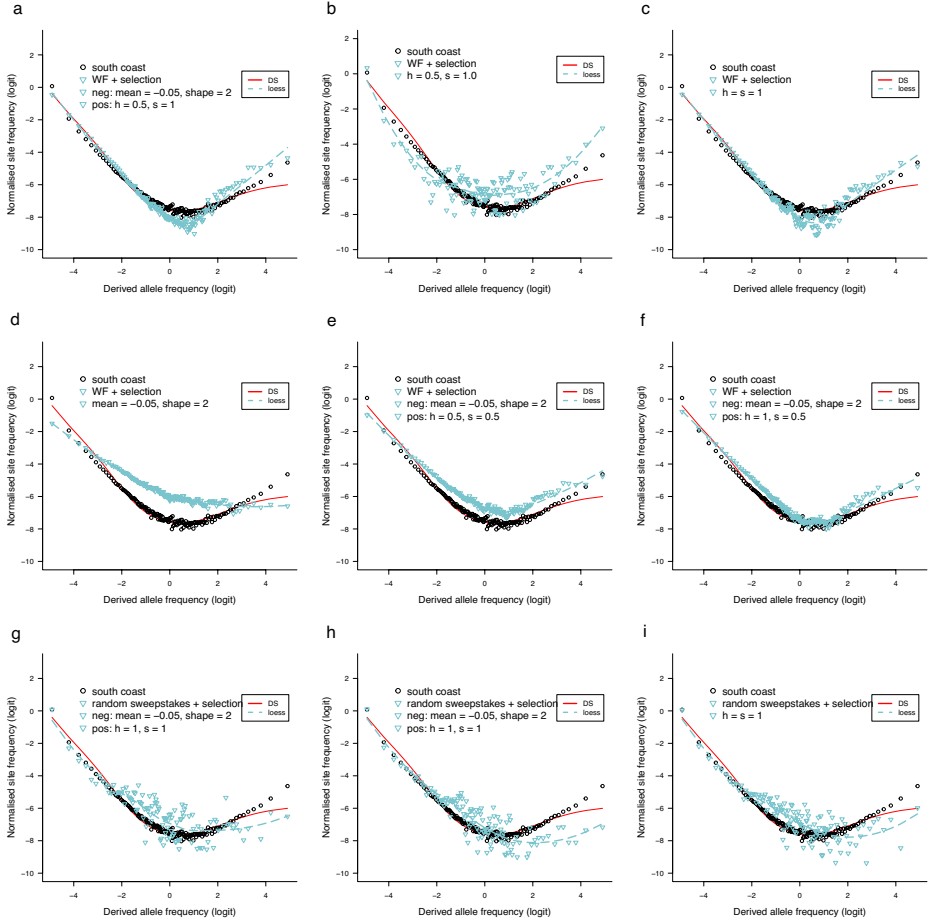

**Appendix 6—figure 9.** Observed site-frequency spectra compared to SLiM simulated site-frequency spectra under no-sweepstakes reproduction and random sweepstakes reproduction with selection. Forward simulation using SLiM (*Haller and Messer, 2019*) of negative (background) selection and positive selection with variable dominance and with no-sweepstakes and random sweepstakes models of reproduction. (**a–f**) The Wright–Fisher no-sweepstakes model (population size $2N = 10^4$) with selection. Negative mutations are modelled as a shifted gamma distribution with mean and shape as shown in each panel and with dominance $h = 1$, and positive mutations with fixed effects with dominance ($h$) and selective advantage (selection coefficient $s$) as shown in each panel. In **b** and **c**, there is no negative selection but only positive mutations of fixed effects with $h$ and $s$ as shown. In **d**, there are only negative mutations with same configuration as in **a**. In **e** and **f**, both positive and negative mutations with configurations as shown. In **g–i**, random sweepstakes using a model in the domain of attraction of the $\Xi$-Beta$(2 - \alpha, \alpha)$-coalescent with population size $2N = 2000$, $\alpha = 1.1$ (**g**) and $\alpha = 1.25$ (**h, i**), with both negative and positive mutations in **g** and **h** with configurations as shown, and only positive mutations in **i**. In all graphs a loess regression curve is fitted to the SLiM data points and compared to predictions of the Durrett–Schweinsberg (DS) coalescent with compound parameter $c = 6$. The circles are site-frequency spectrum of chromosome 3 from the South/south-east coast population estimated with GL1 genotype likelihood. The scripts to generate the graphs are available at https://github.com/eldonb/selective-sweepstakes, (copy archived at swh:1:rev:3235fd1a87f2741b486cb9fe17a15ae85f605d26; *Eldon, 2022b*)

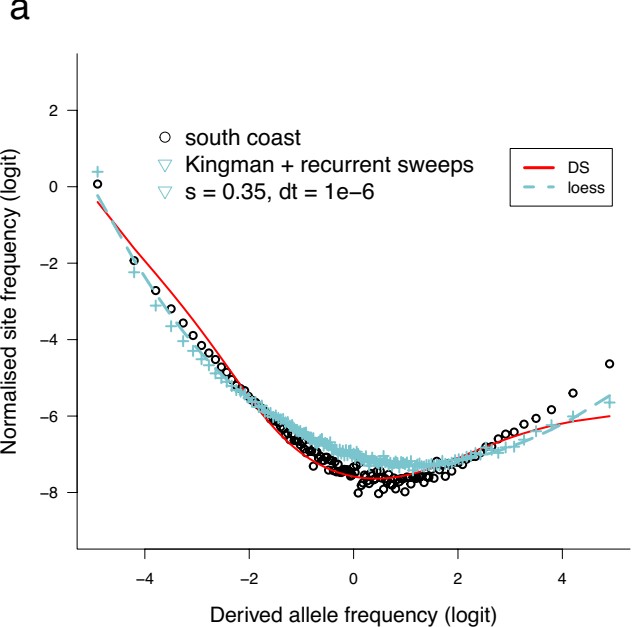

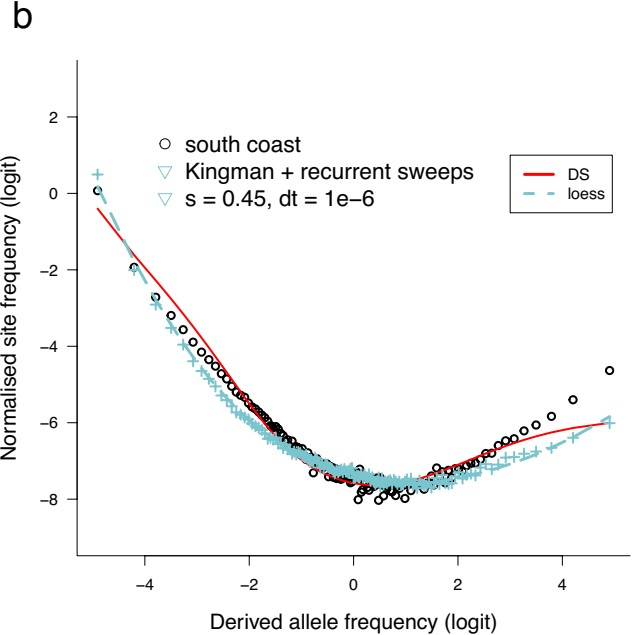

**Appendix 6—figure 10.** Observed site-frequency spectra compared to `msprime` simulated site-frequency spectra under Kingman coalescent with recurrent selective sweeps. Backwards simulation using `msprime` (***Baumdicker et al., 2021***). (**a, b**) The standard Kingman coalescent model interrupted by randomly occurring hard sweeps. Each sweep with a selection coefficient $s$ (and time $dt$ between allele frequency updates) occurs at a random location on a chromosome of length 1 Mbp. `msprime` simulations of the Kingman coalescent and where hard sweeps occur at random times using a structured coalescent approach to model a sweep (***Braverman et al., 1995***), and `msprime` simulates a stochastic sweep trajectory according to a conditional diffusion model (***Kim and Stephan, 2002***; ***Coop and Griffiths, 2004***). See the documentation of `msprime` for further details (tskit.dev/msprime/docs/stable/ancestry.html#sec-ancestry-models-selective-sweeps). The effective population size was $N_e = 10^4$, mutation rate $\mu = 10^{-8}$, and recombination rate. $\gamma = 10^{-7}$ The circles represent the site-frequency spectrum of chromosome 3 (GL1) from the South/south-east coast population, and the red line is the normalized exact expected branch-length spectrum predicted by the Durrett–Schweinsberg coalescent with parameter $c = 6$. The scripts to produce the graphs are available at https://github.com/eldonb.

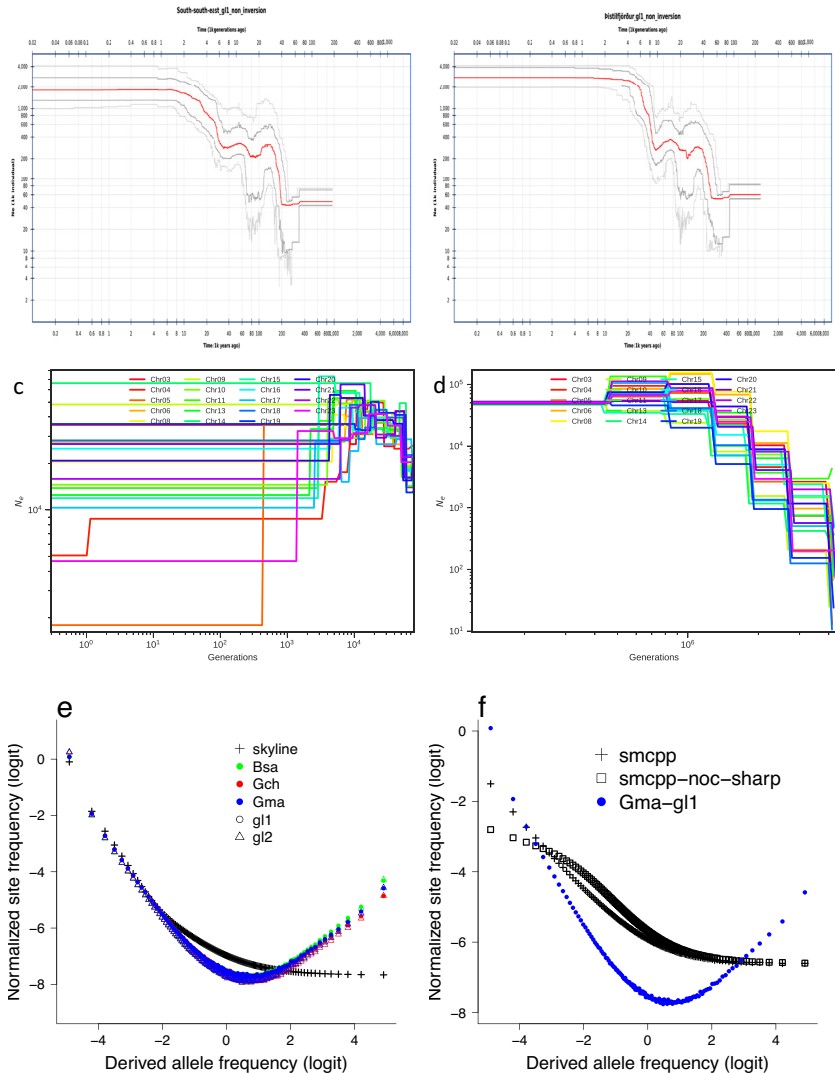

**Appendix 6—figure 11.** Estimated demographic history and frequency spectra from simulated demographic scenarios under the Kingman coalescent. (top, left and right) Demographic history estimated with the stairway plot method (*Liu and Fu, 2015*; *Liu and Fu, 2020*) from the site-frequency spectra of the non-inversion chromosomes estimated with GL1 likelihoods of the South/south-east and Þistilfjörður population, respectively. Population expansion in the distant past and relative stability in more recent times. Demographic history estimated with smc++ (*Terhorst et al., 2016*) for the South/south-east population. smc++ run with default values (**c**) and treating runs of homozygosity as missing with the `--missing-cutoff 10` flag (smcpp-noc-sharp) (**d**). Expected site-frequency spectra simulated using `msprime` (*Kelleher et al., 2016*; *Baumdicker et al., 2021*) based on the demographic scenarios of the stairway plot (**e**) and the smc++ (**f**) for the South/south-east population. The observed site-frequency spectra of the non-inversion chromosomes of the South/south-east population estimated using the GL1 and GL2 likelihoods and polarized using different outgroups (Bsa, Gch, and Gma) (**e**). For the smc ++ comparison the observed data are the average of the non-inversion chromosomes of the South/south-east population estimated using the GL1 genotype likelihood and polarized with Gma as outgroup (**f**).

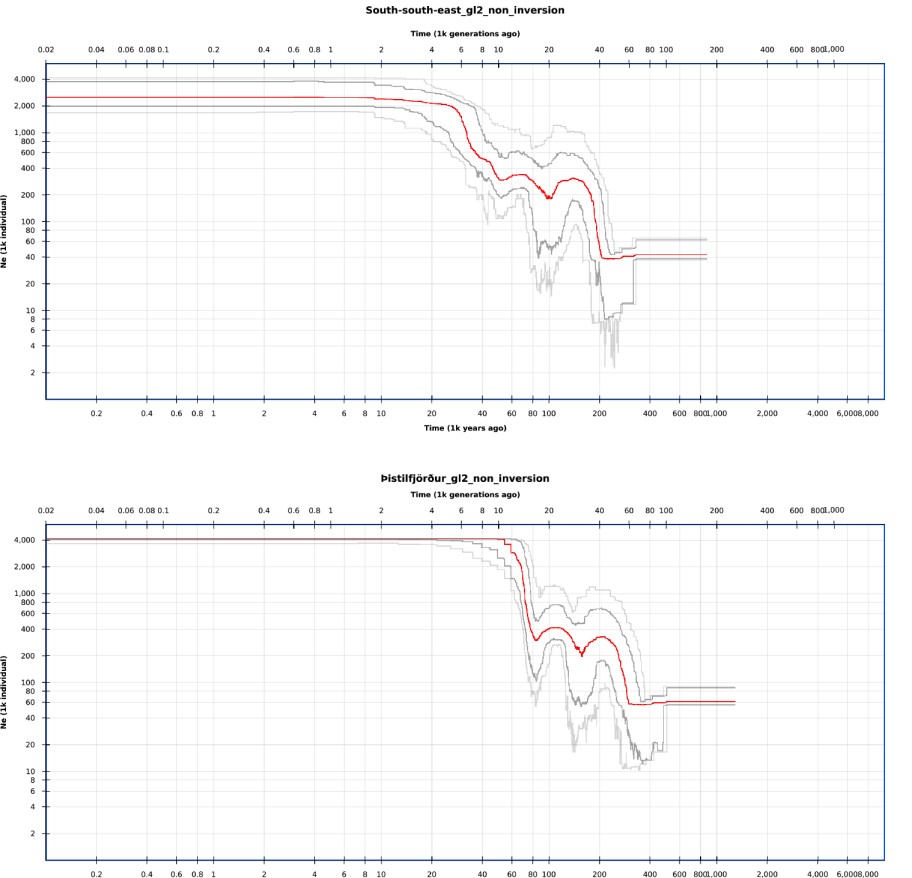

**Appendix 6—figure 12.** Stairway plots of demographic history of the populations of GL2 likelihood data. Demographic history estimated from the site-frequency spectra of the non-inversion chromosomes based on GL2 likelihoods for the South/south-east (top) and the Þistilfjörður (bottom) populations, respectively, with the stairway plot method (*Liu and Fu, 2015*; *Liu and Fu, 2020*).

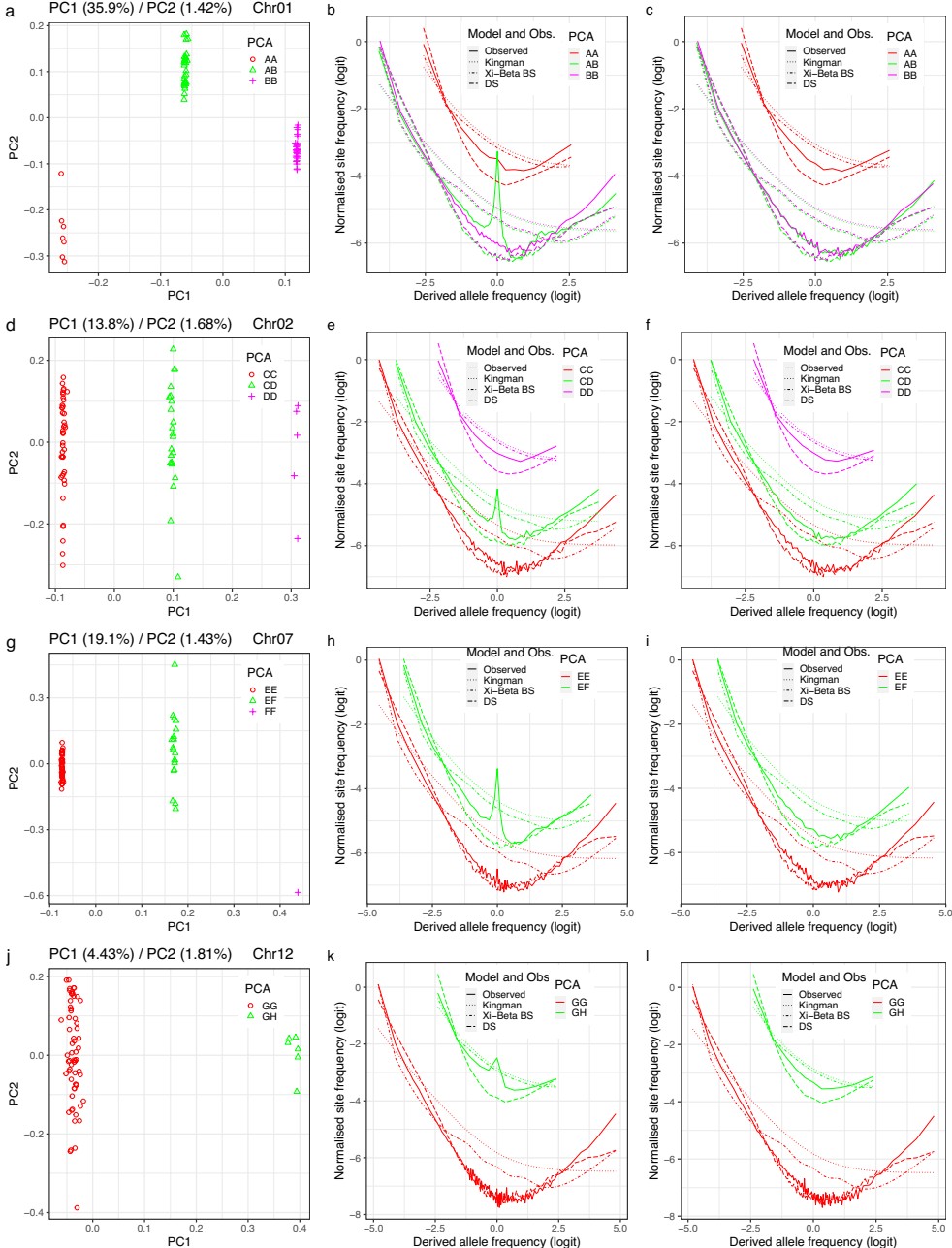

**Appendix 6—figure 13.** Groups from principal component analysis (PCA), conjectured as cryptic population structure, and observed site-frequency spectra compared to coalescent expectations. (**a, d, g, j**) Groups revealed by PCA of variation at inversion chromosomes Chr01, Chr02, Chr07, and Chr12, respectively, conjectured to represent cryptic population structure that should extend to the whole genome. (**b, e, h, k**) The site-frequency spectra estimated for the groups of the respective inversion chromosome using variation at each inversion chromosome. (**c, f, i, l**) The site-frequency spectra estimated for the groups of the respective inversion chromosome using variation at the 19 non-inversion chromosomes. Observed site-frequency spectra compared to expectations based on the *Kingman, 1982* (no-sweepstakes), the $\Xi$-Beta$(2-\alpha, \alpha)$ coalescent (*Schweinsberg, 2000*) (random sweepstakes) with $\alpha = 1$ (the Bolthausen–Sznitman coalescent, BS), and the Durrett-Schweinsberg coalescent (*Durrett and Schweinsberg, 2005*) of recurrent selective sweeps (DS) approximate Bayesian computation (ABC) estimated for the PCA groups of each chromosome. Data from the South/south-east population. Non-inversion chromosomes show no peaks at intermediate frequencies as expected under the conjecture. The conjecture of cryptic population structure is rejected.

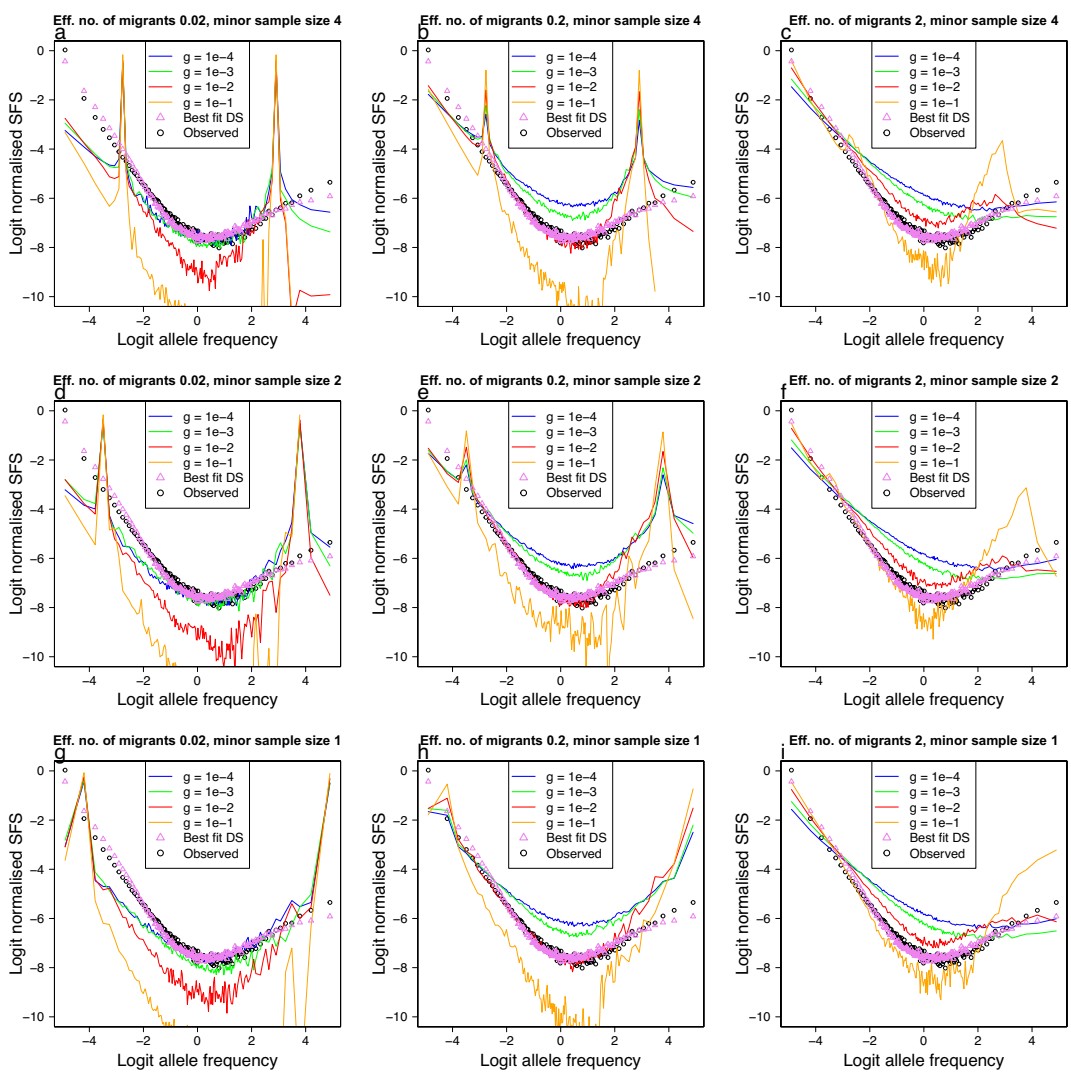

**Appendix 6—figure 14.** Population structure, isolation with migration, and population growth under the Kingman coalescent. (a–i) Simulations using `msprime` (***Kelleher et al., 2016***; ***Baumdicker et al., 2021***) of the effects of a mixed sample of two divergent populations evolving under the Kingman coalescent (no sweepstakes model) with population growth on the expected site-frequency spectrum. A two island model with migration rate $m$ and per-generation population growth rate $g$. The effective number of migrants ($N_e m$) increases from 0.02 to 2, and hence the degree of isolation decreases, going from left to right. The sample size of the minor populations (the population from which fewer individuals are sampled) decreases from top to bottom (minor sample size 4…1). The effects of population growth $g$ displayed with different colours. Simulated model expectations (solid lines) compared to observed data (circles) of chromosome 4 estimated with GL1 and Gma as outgroup from the South/south-east population for comparison. Also included are the expectation of the Durrett–Schweinsberg coalescent (***Durrett and Schweinsberg, 2005***) (selective sweepstakes model, best-fit DS: triangle) to observations. (i) Only a minor sample size of one combined with the highest rate of migration and particular growth rates gives closest resemblance to observations.

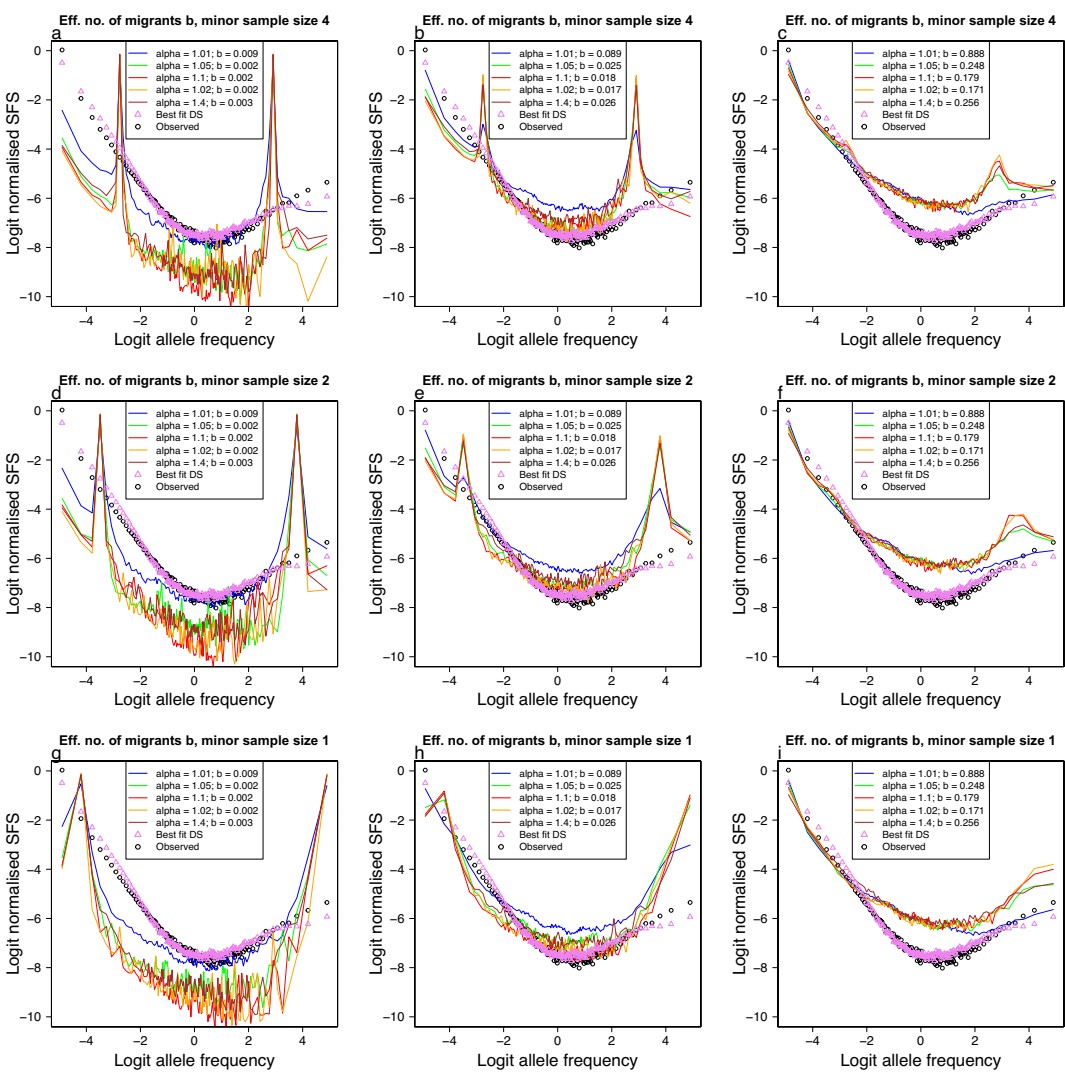

**Appendix 6—figure 15.** Population structure, isolation with migration, and population growth under the Xi-Beta coalescent. (**a-i**) Simulations using `msprime` (***Kelleher et al., 2016***; ***Baumdicker et al., 2021***) of the effects of a mixed sample of two divergent populations evolving under the $\Xi$-Beta$(2-\alpha, \alpha)$ coalescent (the random sweepstakes model) on the expected site-frequency spectrum. A two island model with migration and different values of the $\alpha$ parameter (displayed with different colours). The effective number of migrants $b$ (comparable to $N_em$ in ***Appendix 6—figure 14***) increases, and hence the degree of isolation decreases, going from left to right. The sample size of the minor populations decreases from top to bottom (4–1). Simulated model expectations (solid lines) compared to observed data (circles) of chromosome 4 estimated with GL1 and Gma as outgroup from the South/south-east population for comparison. Also included are the expectation of the Durrett–Schweinsberg coalescent (***Durrett and Schweinsberg, 2005***) (selective sweepstakes model, best-fit DS: triangle) to observations. (**i**) Only a minor sample size of one combined with the highest rate of migration and particular values of $\alpha$ gives closest resemblance to observations.

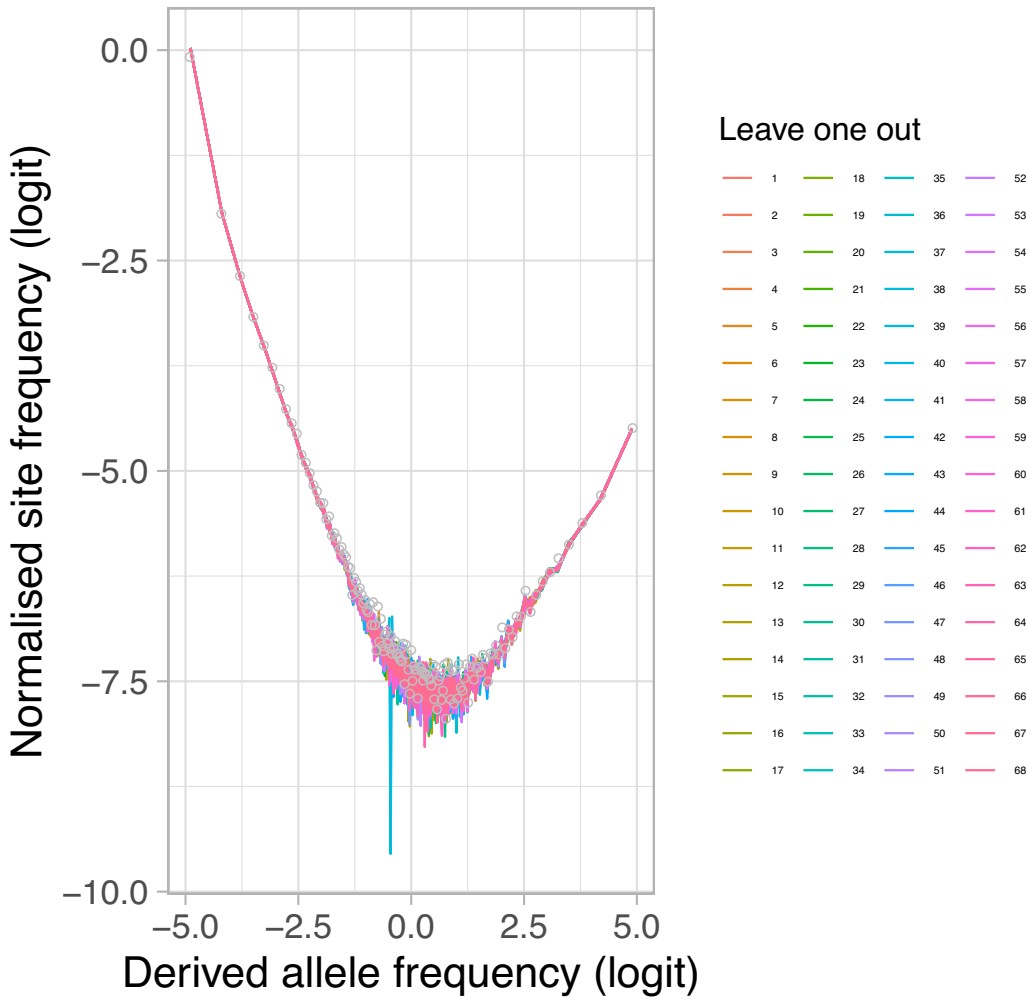

**Appendix 6—figure 16.** Estimated site-frequency spectra with a leave-one-out approach. Estimated site-frequency spectra for chromosome 4 of 67 individuals leaving out each individual in turn from the 68 individuals of the South/south-east population. Circles are site-frequency spectrum of the original sample of 68 individuals. Based on the simulations results in *Appendix 6—figure 14* and *Appendix 6—figure 15* that a minor sample size of one can resemble model expectations, one of the leave-one-out samples should be divergent if the sample of 68 individuals is composed of 67 individuals from one population and a single individual from a divergent population. None of the leave-one-out samples is off so this conjecture is rejected.

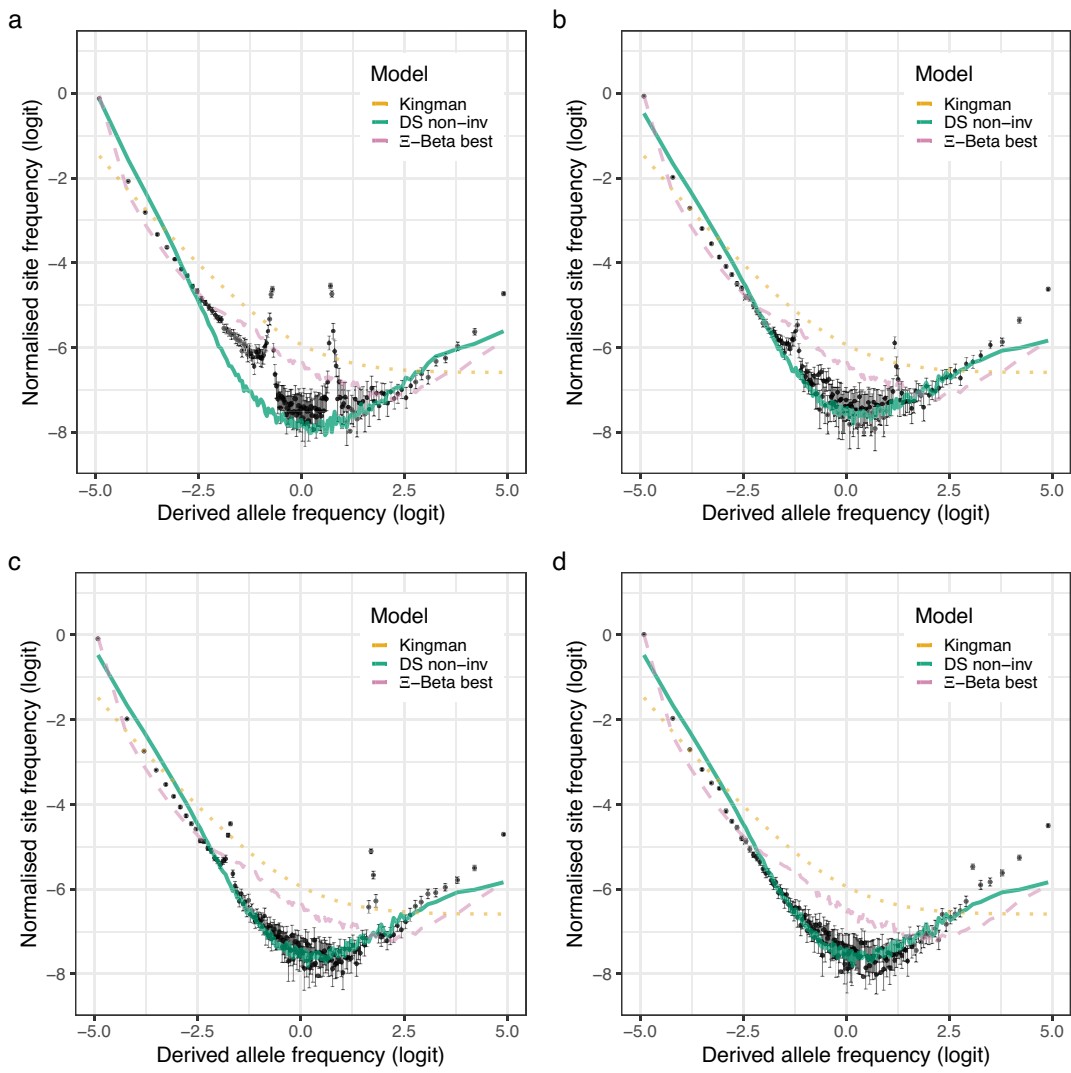

**Appendix 6—figure 17.** Observed site-frequency spectra at inversion chromosomes and coalescent expectations for the South/south-east population. (**a–d**) Observed site-frequency spectra estimated with GL1 for the four inversion chromosomes, chromosome 1 (Chr01), chromosome 2 (Chr02), chromosome 7 (Chr07), and chromosome 12 (Chr12), respectively in the South/south-east population (sample size $n = 68$). Error bars are ±2 standard deviations of the bootstrap distribution with 100 bootstrap replicates. Expected site-frequency spectra are the Kingman coalescent (the no sweepstakes model), the Durrett–Schweinsberg (DS) coalescent (selective sweepstakes model) approximate Bayesian computation (ABC) estimated for the non-inversion chromosomes (DS non-inv), and the best approximate maximum likelihood estimates (**Eldon et al., 2015**) of the Ξ-Beta model (the random sweepstakes model). The best estimated $\hat{\alpha}$ values were $\hat{\alpha}_\Xi = 1.16$, $\hat{\alpha}_\Xi = 1.16$, $\hat{\alpha}_\Xi = 1.16$, and $\hat{\alpha}_\Xi = 1.12$, for chromosomes 1, 2, 7, and 12, respectively.

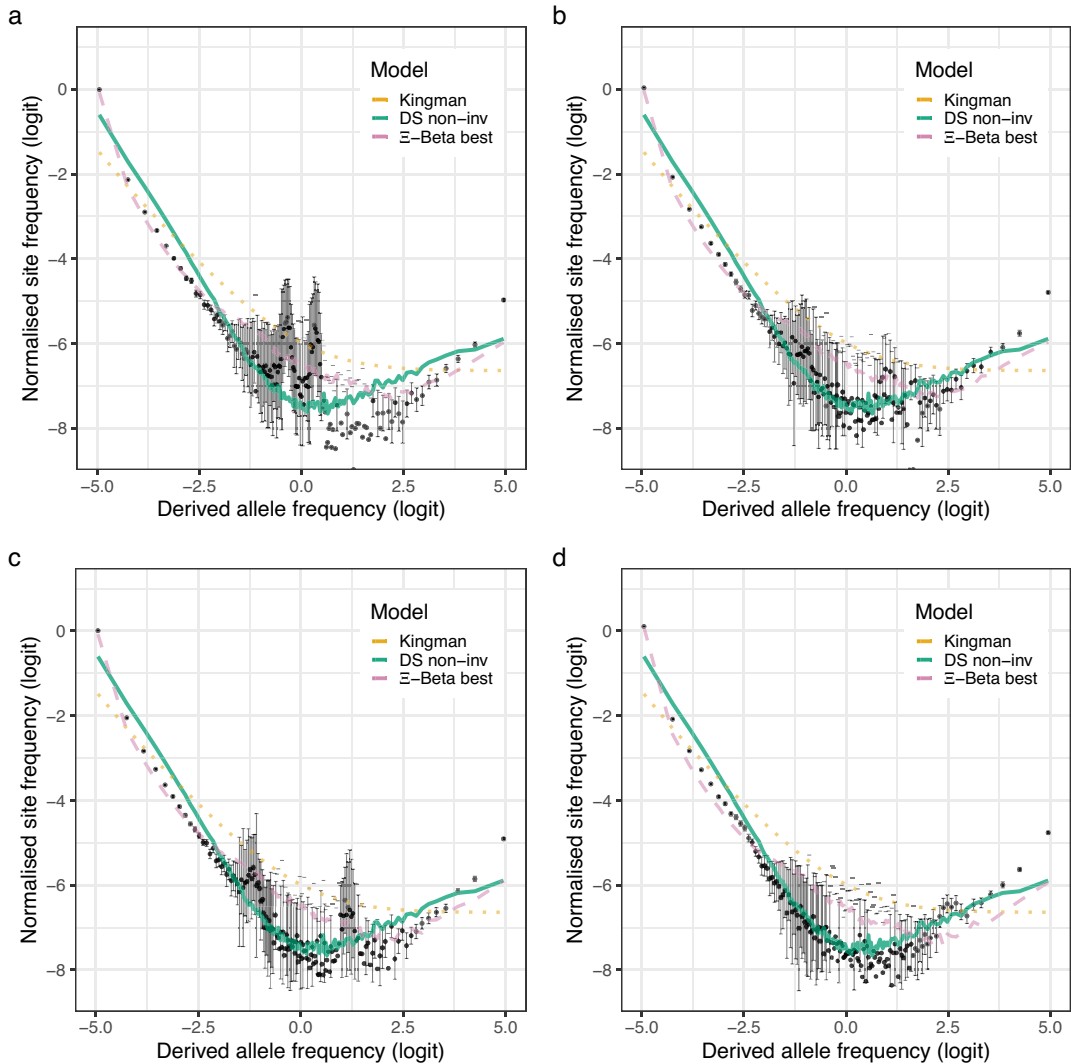

**Appendix 6—figure 18.** Observed site-frequency spectra at inversion chromosomes and coalescent expectations for the Þistilfjörður population. (**a–d**) Observed site-frequency spectra estimated with GL1 for the four inversion chromosomes, chromosome 1 (Chr01), chromosome 2 (Chr02), chromosome 7 (Chr07), and chromosome 12 (Chr12), respectively for the Þistilfjörður population (sample size $n = 71$). Error bars are ±2 standard deviations of the bootstrap distribution with 100 bootstrap replicates. Expected site-frequency spectra are the Kingman coalescent (the no sweepstakes model), the Durrett–Schweinsberg (DS) coalescent (selective sweepstakes model) approximate Bayesian computation (ABC) estimated for the non-inversion chromosomes (non-inv), and the best approximate maximum likelihood estimates (**Eldon et al., 2015**) of the Ξ-Beta model (the random sweepstakes model). The best estimated $\hat{\alpha}$ values are $\hat{\alpha}_\Xi = 1.18$, $\hat{\alpha}_\Xi = 1.16$, $\hat{\alpha}_\Xi = 1.17$, and $\hat{\alpha}_\Xi = 1.08$, for chromosomes 1, 2, 7, and 12, respectively.

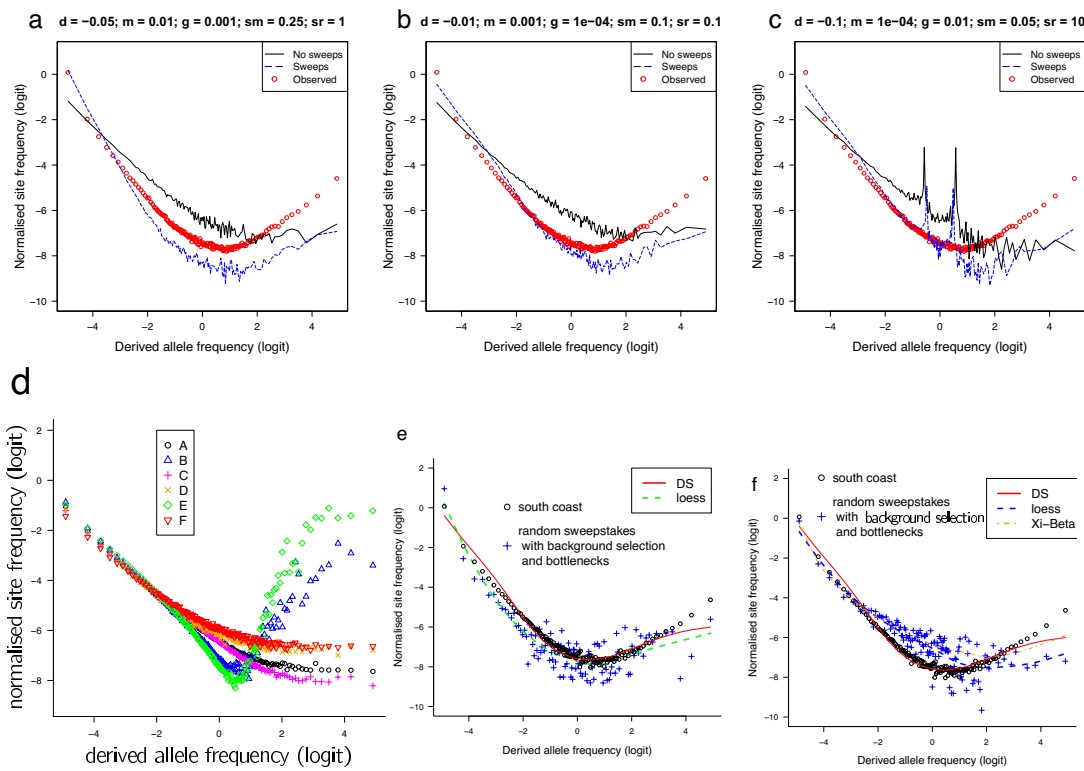

**Appendix 6—figure 19.** Observed site-frequency spectra compared to SLiM forward simulated site-frequency spectra based on demographic scenarios with and without selective sweeps and with background selection and recurrent bottlenecks. Forward simulation using SLiM (*Haller and Messer, 2019*). (**a–c**) Each scenario has two islands of initial population size 300. Both islands undergo exponential growth at per-generation rate $g$ until a total size of 1000. The per-generation migration probability is $m$. The SLiM simulation is run until the whole population has a MRCA, at which point 136 haploid genomes (as the sample from the South/south-east population) are drawn from the population. Each scenario is simulated 1000 times to estimate the mean normalized site-frequency spectrum. The genome length is set to 100 kb, and the recombination and mutation rates are $10^{-8}$ per site per generation. The 'No sweeps' scenario undergoes deleterious mutations with fitness effects described by a gamma distribution with mean $d$ and shape parameter 0.2. The 'Sweeps' scenario has the same deleterious mutations, and also beneficial mutations with a fixed fitness effect of $sm$. The relative rate of these positive mutations to the deleterious ones is $sr$. The observed site-frequency spectrum is the mean of the 100 kb fragments across all non-inversion chromosomes. Only sweeps scenarios show U-shaped site-frequency spectra. (**d**) Results of simulations of background selection. In all cases a population of size $N = 10^5$ evolves according to the Wright–Fisher model assuming a chromosome segment of size $10^5$ bp with recombination rate $10^{-7}$ per site per generation that collects neutral or negative mutations with frequency $\mu = 10^{-7}$ per site per generation. Negative mutations were modelled as Gamma-distributed with a negative sign, with mean −0.1 (**a, b, d**) or −0.05 (**c, e, f**) all with shape parameter 0.2. The relative frequency of negative versus neutral mutations was 1:1 for (**a, b, d**) and 1:9 for (**c, e, f**). The points represent the logits of the normalized site-frequency spectrum of a random sample of 136 chromosomes (corresponding to the sample size of the South/south-east population) averaged over 100 (**a, b, d, e**) and 10 (**d, f**) replicates and taken after $10^5$ generations (**b, e**), $2 \times 10^5$ generations (**a, c**), and $10^6$ generations (**d, f**). U-shaped site-frequency spectra only found for short runs (**b, e**). (**e, f**) Simulations were produced by the C++ simulation code `forward` available at https://github.com/eldonb/forward; *Eldon, 2022a* for individual-based forward-in-time simulations with random sweepstakes, randomly occurring bottlenecks, and selection. Haploid model in **e** and diploid model in **f**.

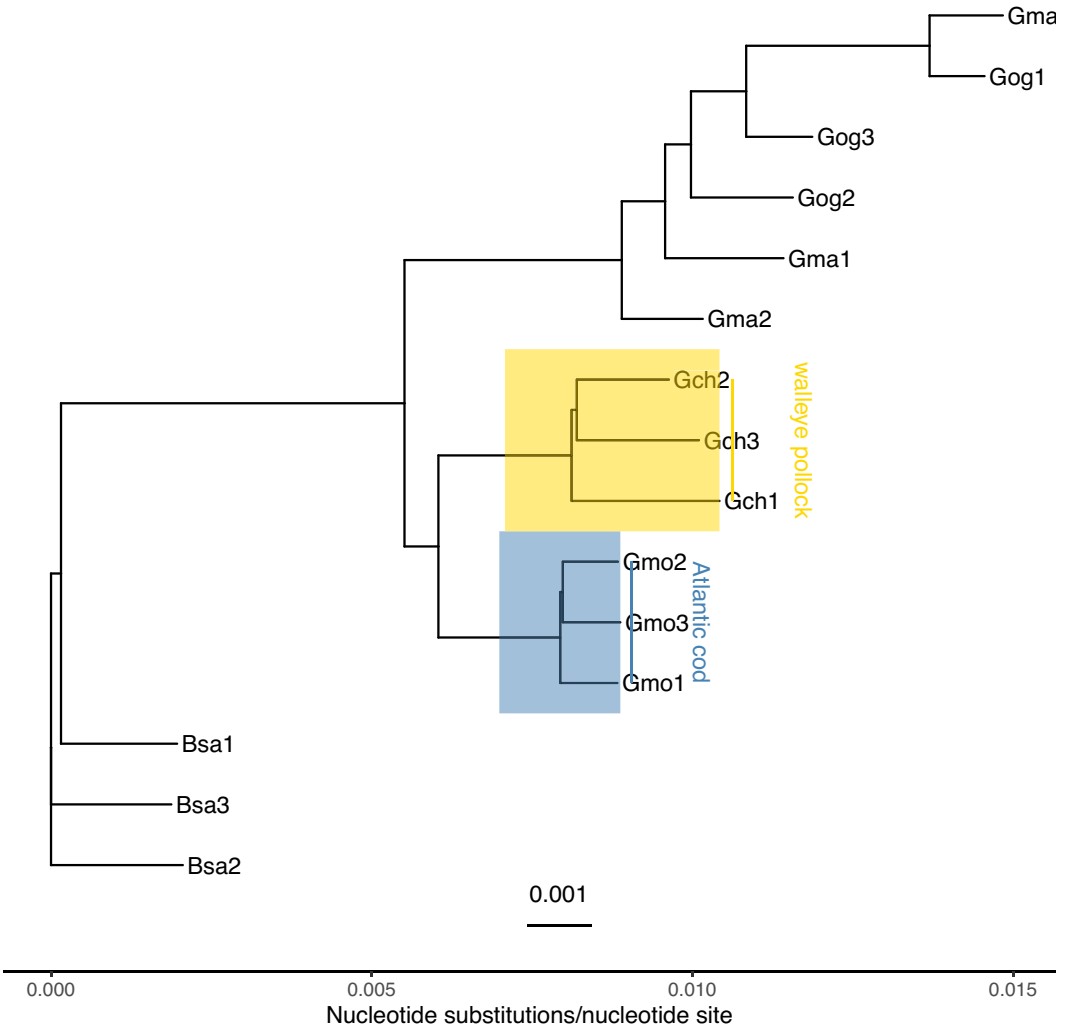

**Appendix 6—figure 20.** Neighbour joining tree of gadid taxa. Based on $p$-distance (nucleotide substitutions per nucleotide site) of whole genome among the gadid taxa Atlantic cod (*Gadus morhua*, Gmo), walleye pollock (*G. chalcogramma*, Gch), Pacific cod (*G. macrocephalus*, Gma), Greenland cod (*G. ogac*, Gog), and Arctic cod (*Boreogadus saida*, Bsa). Under the assumption that the focal taxa, Atlantic cod and walleye pollock, diverged $3.5 \times 10^6$ years ago (*Vermeij, 1991*; *Vermeij and Roopnarine, 2008*; *Coulson et al., 2006*; *Carr and Marshall, 2008*), the distance between these taxa is used for mutation rate estimation in *Appendix 7—table 4*.

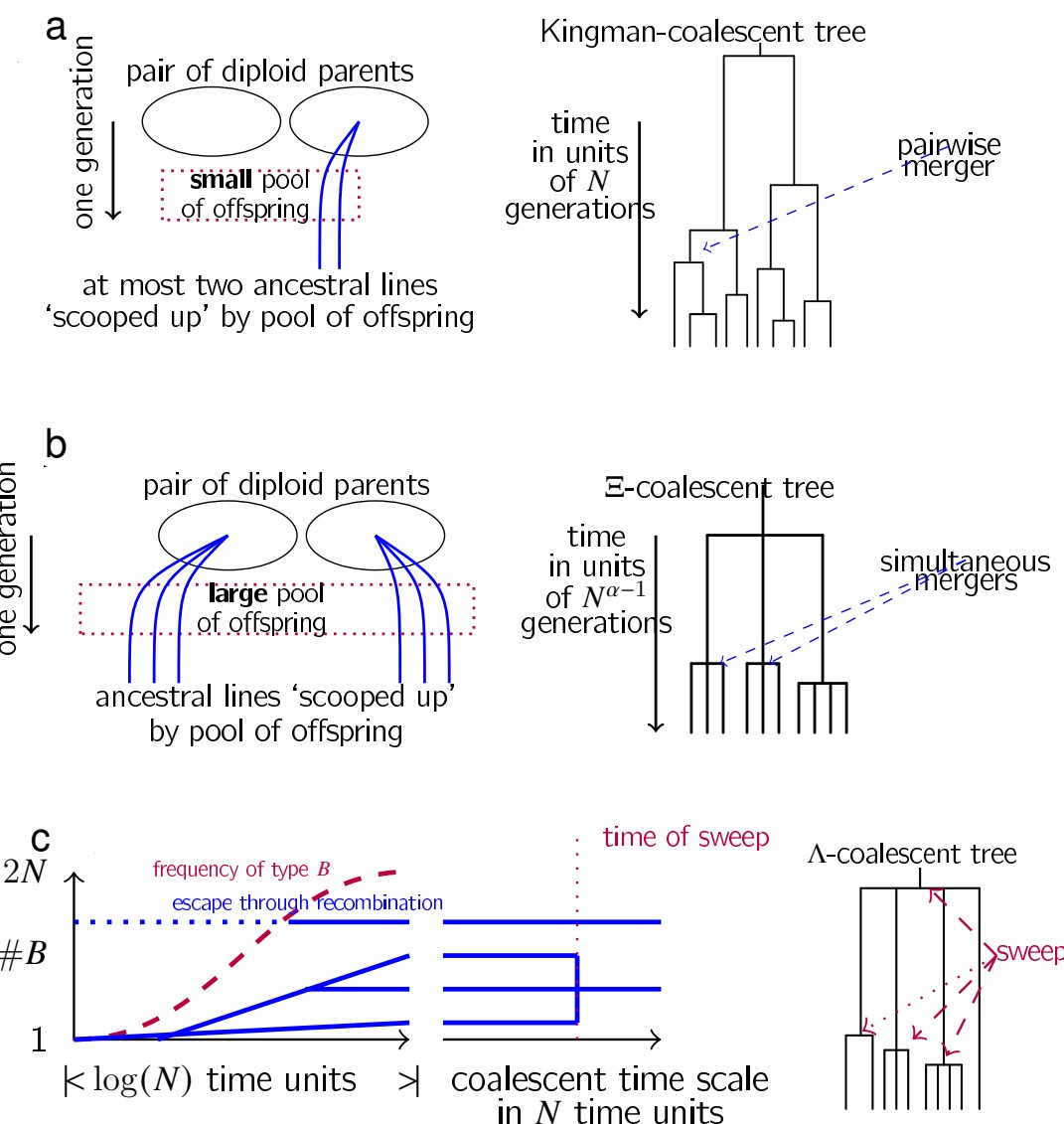

**Appendix 6—figure 21.** Schematic illustration of the three coalescent models, Kingman (no sweepstakes), Xi-Beta (random sweepstakes), and DS (selective sweepstakes). (**a**) In each generation, any given pair of diploid parents in a low-fecundity population produces only a small number of offspring, a no-sweepstakes scenario. At most two ancestral lineages (shown as blue lines) of a sample can, therefore, be involved in a given family with non-negligible probability in a large population, leading to at most two ancestral lineages merging each time when the ancestral tree is viewed on a coalescent timescale of $N$ generations per coalescent time unit. (**b**) In a highly fecund population reproducing according to random sweepstakes reproduction, a given pair of diploid parents may produce a huge number of offspring, scooping up a number of ancestral lineages of a sample (shown as blue lines) in an instance of random sweepstakes. The resulting gene genealogy may include multiple and simultaneous multiple mergers of ancestral lineages of a sample. (**c**) An example of the effects of selective sweepstakes through repeated selective sweeps on the genealogy of a neutral site. Shown is a hypothetical history of ancestral lineages of a sample (blue lines) at the neutral site during a sweep of the beneficial allelic type $B$ at a site different from the neutral site. At the start of a sweep a single chromosome not ancestral to the sample experiences a mutation to type $B$. During the sweep one of the ancestral chromosomes has several descendants while another (shown in dotted blue lines) manages to 'escape' a sweep by recombining onto a 'b' background. At the end of the sweep all chromosomes have a 'B' background, however, not all of the ancestral lineages will trace back to the initial $B$ chromosome. Since we are only interested in the genealogy at the neutral site only the ancestral relations of the neutral site are shown (blue lines). Viewed on a coalescent timescale of $N$ time units per one coalescent time unit, the sweep happens instantaneously, and thus appears as an instantaneous merger of three lineages in the genealogy of the neutral site.

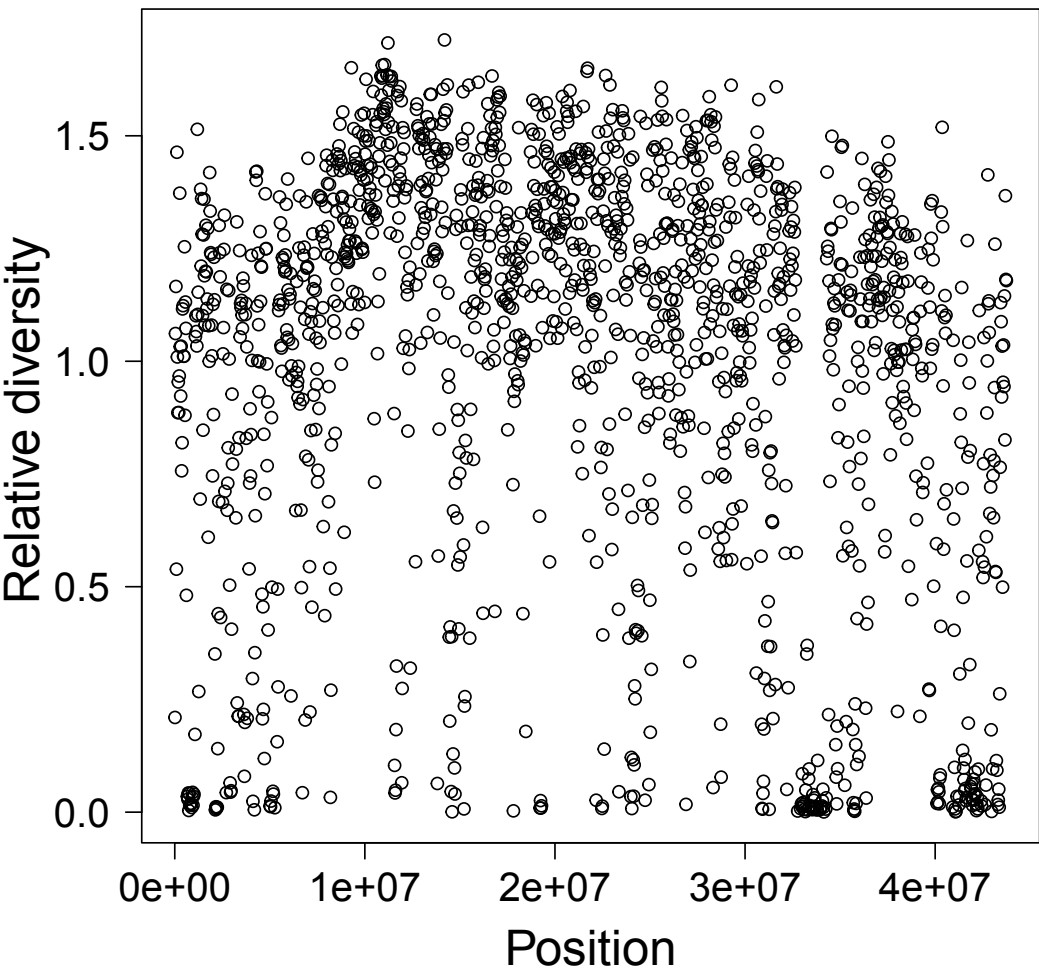

**Appendix 6—figure 22.** Relative diversity and the compound parameter $c$ along chromosome 4. The compound parameter $c = \delta s^2/\gamma$ of the Durrett–Schweinsberg model measures the rate of selective sweeps ($\delta$) times the squared selection coefficient ($s^2$) of the beneficial mutation over the recombination rate ($\gamma$) between the selected site and the neutral site of interest. The compound parameter $c$ can be considered to be essentialy the density of selection per map unit along a chromosome (*Aeschbacher et al., 2017*). The number of single nucleotide polymorphisms (SNPs) in a 25 k fragment is proportional to branch length, which again is proportional to the compound parameter $c$. The relative diversity is the number of SNPs normalized by the mean number of SNPs on chromosome fragment location is indicative of the density of selection along a chromosome.

## Appendix 7

## Supplementary tables

**Appendix 7—table 1.** Diversity and neutrality test statistics for the South/south-east population. Watterson's estimator of the population scaled mutation rate per nucleotide site $\theta_W$, the pairwise nucleotide diversity per nucleotide site $\theta_\pi$, Tajima's $D_T$, Fu and Li's $D_F$, and number of nucleotide sites based on GL1 and GL2 likelihoods (sample size $n = 68$).

| | GL1 likelihood | | | | | GL2 likelihood | | | | |
|---|---|---|---|---|---|---|---|---|---|---|
| | $\theta_W$ | $\theta_\pi$ | $D_T$ | $D_F$ | nSites | $\theta_W$ | $\theta_\pi$ | $D_T$ | $D_F$ | nSites |
| Chr01 | 0.0046 | 0.0024 | −1.64 | −5.77 | 18332422 | 0.0056 | 0.0025 | −1.84 | −6.71 | 18332093 |
| Chr02 | 0.0050 | 0.0020 | −1.98 | −6.00 | 15828347 | 0.0060 | 0.0022 | −2.11 | −6.84 | 15828079 |
| Chr03 | 0.0053 | 0.0020 | −2.09 | −6.22 | 20202769 | 0.0063 | 0.0021 | −2.21 | −6.98 | 20202435 |
| Chr04 | 0.0054 | 0.0020 | −2.08 | −6.03 | 22584280 | 0.0065 | 0.0022 | −2.19 | −6.79 | 22583924 |
| Chr05 | 0.0053 | 0.0020 | −2.10 | −6.22 | 15542562 | 0.0064 | 0.0021 | −2.22 | −6.99 | 15542313 |
| Chr06 | 0.0052 | 0.0019 | −2.11 | −6.33 | 17720989 | 0.0062 | 0.0021 | −2.22 | −7.09 | 17720709 |
| Chr07 | 0.0056 | 0.0022 | −2.01 | −5.88 | 21080002 | 0.0066 | 0.0024 | −2.13 | −6.64 | 21079620 |
| Chr08 | 0.0054 | 0.0020 | −2.09 | −6.09 | 18353883 | 0.0065 | 0.0022 | −2.21 | −6.85 | 18353624 |
| Chr09 | 0.0053 | 0.0019 | −2.13 | −6.42 | 18195728 | 0.0063 | 0.0021 | −2.25 | −7.16 | 18195437 |
| Chr10 | 0.0051 | 0.0019 | −2.09 | −6.27 | 17450729 | 0.0061 | 0.0020 | −2.21 | −7.06 | 17450432 |
| Chr11 | 0.0050 | 0.0018 | −2.14 | −6.54 | 20138893 | 0.0059 | 0.0019 | −2.26 | −7.32 | 20138619 |
| Chr12 | 0.0043 | 0.0016 | −2.14 | −6.32 | 19448827 | 0.0053 | 0.0017 | −2.26 | −7.18 | 19448580 |
| Chr13 | 0.0049 | 0.0018 | −2.14 | −6.38 | 18651575 | 0.0059 | 0.0019 | −2.26 | −7.18 | 18651311 |
| Chr14 | 0.0053 | 0.0019 | −2.14 | −6.34 | 20704894 | 0.0063 | 0.0020 | −2.25 | −7.09 | 20704623 |
| Chr15 | 0.0054 | 0.0019 | −2.17 | −6.41 | 18100213 | 0.0064 | 0.0020 | −2.27 | −7.15 | 18099944 |
| Chr16 | 0.0051 | 0.0019 | −2.09 | −6.13 | 22233178 | 0.0061 | 0.0021 | −2.21 | −6.93 | 22232862 |
| Chr17 | 0.0053 | 0.0020 | −2.06 | −5.99 | 11813809 | 0.0063 | 0.0022 | −2.18 | −6.78 | 11813609 |
| Chr18 | 0.0053 | 0.0019 | −2.11 | −6.23 | 15931558 | 0.0063 | 0.0021 | −2.23 | −7.01 | 15931312 |
| Chr19 | 0.0055 | 0.0020 | −2.10 | −6.23 | 13858302 | 0.0065 | 0.0022 | −2.21 | −6.98 | 13858066 |
| Chr20 | 0.0050 | 0.0018 | −2.15 | −6.56 | 16371168 | 0.0059 | 0.0019 | −2.27 | −7.33 | 16370967 |
| Chr21 | 0.0052 | 0.0019 | −2.10 | −6.29 | 14440220 | 0.0062 | 0.0021 | −2.22 | −7.07 | 14440024 |
| Chr22 | 0.0054 | 0.0020 | −2.08 | −6.12 | 13838440 | 0.0065 | 0.0022 | −2.19 | −6.89 | 13838214 |
| Chr23 | 0.0052 | 0.0020 | −2.08 | −6.27 | 14698719 | 0.0062 | 0.0021 | −2.20 | −7.05 | 14698473 |
| All | 0.0052 | 0.0019 | −2.08 | −6.22 | 17631370 | 0.0062 | 0.0021 | −2.20 | −7.00 | 17631099 |

**Appendix 7—table 2.** Diversity and neutrality test statistics for the Þistilfjörður population. $\theta_W$ Watterson's estimator of the population scaled mutation rate per nucleotide site, $\theta_\pi$ the pairwise nucleotide diversity per nucleotide site, Tajima's $D_T$, Fu and Li's $D_F$, and number of nucleotide sites based on GL1 and GL2 likelihoods (sample size $n = 71$).

| | GL1 likelihood | | | | | GL2 likelihood | | | | |
|---|---|---|---|---|---|---|---|---|---|---|
| | $\theta_W$ | $\theta_\pi$ | $D_T$ | $D_F$ | nSites | $\theta_W$ | $\theta_\pi$ | $D_T$ | $D_F$ | nSites |
| Chr01 | 0.0068 | 0.0037 | −1.51 | −5.99 | 16159362 | 0.0090 | 0.0040 | −1.84 | −7.55 | 16159148 |
| Chr02 | 0.0069 | 0.0030 | −1.86 | −6.18 | 14306627 | 0.0092 | 0.0034 | −2.10 | −7.65 | 14306351 |
| Chr03 | 0.0073 | 0.0029 | −1.99 | −6.38 | 18283815 | 0.0096 | 0.0033 | −2.19 | −7.76 | 18283555 |
| Chr04 | 0.0074 | 0.0030 | −1.97 | −6.14 | 20435443 | 0.0097 | 0.0034 | −2.17 | −7.52 | 20435122 |
| Chr05 | 0.0073 | 0.0029 | −2.00 | −6.36 | 13933982 | 0.0096 | 0.0032 | −2.20 | −7.74 | 13933752 |
| Chr06 | 0.0072 | 0.0028 | −2.00 | −6.46 | 16048768 | 0.0094 | 0.0032 | −2.21 | −7.84 | 16048531 |

*Appendix 7—table 2 Continued on next page*

*Appendix 7—table 2 Continued*

| | GL1 likelihood | | | | | GL2 likelihood | | | | |
|---|---|---|---|---|---|---|---|---|---|---|
| | $\theta_W$ | $\theta_\pi$ | $D_T$ | $D_F$ | nSites | $\theta_W$ | $\theta_\pi$ | $D_T$ | $D_F$ | nSites |
| Chr07 | 0.0076 | 0.0034 | −1.83 | −6.06 | 19008270 | 0.0099 | 0.0038 | −2.05 | −7.46 | 19007926 |
| Chr08 | 0.0074 | 0.0030 | −1.98 | −6.20 | 16559106 | 0.0097 | 0.0033 | −2.18 | −7.59 | 16558861 |
| Chr09 | 0.0073 | 0.0028 | −2.03 | −6.59 | 16381498 | 0.0096 | 0.0032 | −2.23 | −7.93 | 16381249 |
| Chr10 | 0.0070 | 0.0028 | −1.98 | −6.42 | 15789838 | 0.0093 | 0.0032 | −2.19 | −7.83 | 15789584 |
| Chr11 | 0.0069 | 0.0026 | −2.04 | −6.73 | 18211081 | 0.0091 | 0.0029 | −2.24 | −8.12 | 18210846 |
| Chr12 | 0.0061 | 0.0024 | −2.03 | −6.52 | 17597347 | 0.0082 | 0.0027 | −2.24 | −8.07 | 17597135 |
| Chr13 | 0.0068 | 0.0026 | −2.04 | −6.58 | 16846892 | 0.0090 | 0.0029 | −2.24 | −8.01 | 16846697 |
| Chr14 | 0.0073 | 0.0028 | −2.04 | −6.52 | 18699877 | 0.0095 | 0.0031 | −2.23 | −7.89 | 18699625 |
| Chr15 | 0.0074 | 0.0028 | −2.06 | −6.54 | 16349327 | 0.0097 | 0.0031 | −2.25 | −7.86 | 16349118 |
| Chr16 | 0.0070 | 0.0028 | −1.98 | −6.27 | 20259494 | 0.0092 | 0.0032 | −2.19 | −7.71 | 20259231 |
| Chr17 | 0.0072 | 0.0030 | −1.93 | −6.09 | 10667396 | 0.0095 | 0.0033 | −2.15 | −7.52 | 10667225 |
| Chr18 | 0.0072 | 0.0029 | −2.00 | −6.39 | 14305479 | 0.0095 | 0.0032 | −2.21 | −7.79 | 14305261 |
| Chr19 | 0.0075 | 0.0030 | −1.98 | −6.33 | 12465223 | 0.0098 | 0.0034 | −2.18 | −7.68 | 12465024 |
| Chr20 | 0.0069 | 0.0026 | −2.06 | −6.73 | 14829191 | 0.0091 | 0.0029 | −2.25 | −8.11 | 14829009 |
| Chr21 | 0.0071 | 0.0029 | −1.99 | −6.43 | 13014009 | 0.0094 | 0.0032 | −2.20 | −7.83 | 13013813 |
| Chr22 | 0.0074 | 0.0030 | −1.97 | −6.30 | 12407034 | 0.0097 | 0.0034 | −2.17 | −7.70 | 12406815 |
| Chr23 | 0.0072 | 0.0029 | −1.97 | −6.40 | 13273011 | 0.0094 | 0.0032 | −2.18 | −7.81 | 13272801 |
| All | 0.0071 | 0.0029 | −1.97 | −6.37 | 15905742 | 0.0094 | 0.0032 | −2.18 | −7.78 | 15905508 |

**Appendix 7—table 3.** Demographic statistics, correction factor, $C$, and generation length, $G$, of female component of Atlantic cod in Iceland.

Age-specific survival rate, $l_i$, was based, respectively, on the average and the 1948–1952 and the 1963–1967 instantaneous mortality estimated from tagging experiments of Icelandic cod (*Jónsson, 1996*). Age-specific fecundity based on the average age-specific weight in catch (*Anonymous, 2001*) and fecundity by weight relationships (*Marteinsdottir and Begg, 2002*) and similar relationships for Newfoundland cod for comparison (*May, 1967*). The $C$ and $G$ are, respectively, the correction factor for the effects of overlapping generations and generation time based on demographic estimation (*Jorde and Ryman, 1995*; *Jorde and Ryman, 1996*; *Laikre et al., 1998*) and iteration of Equations 5–9 in *Jorde and Ryman, 1996*. Table is truncated at Age class 15 for lack of population data on older age classes.

| Age | Age class | $\bar{l}_i$ | '48–'52 $l_i$ | '63–'67 $l_i$ | GM $b_i \times 10^6$ | May $b_i \times 10^6$ |
|---|---|---|---|---|---|---|
| 0 | 1 | 1.0000 | 1.0000 | 1.0000 | 0.00 | 0 |
| 1 | 2 | 0.3396 | 0.4966 | 0.2369 | 0.00 | 0 |
| 2 | 3 | 0.1153 | 0.2466 | 0.0561 | 0.00 | 0 |
| 3 | 4 | 0.0392 | 0.1225 | 0.0133 | 0.38 | 0.52 |
| 4 | 5 | 0.0133 | 0.0608 | 0.0032 | 0.62 | 0.78 |
| 5 | 6 | 0.0045 | 0.0302 | 0.0007 | 1.01 | 1.15 |
| 6 | 7 | 0.0015 | 0.0150 | 0.0002 | 1.59 | 1.67 |
| 7 | 8 | 0.0005 | 0.0074 | 0.0000 | 2.37 | 2.31 |
| 8 | 9 | 0.0002 | 0.0037 | 0.0000 | 3.28 | 3.03 |
| 9 | 10 | 0.0001 | 0.0018 | 0.0000 | 4.24 | 3.73 |

*Appendix 7—table 3 Continued on next page*

*Appendix 7—table 3 Continued*

| Age | Age class | $\bar{l}_i$ | '48–'52 $l_i$ | '63–'67 $l_i$ | GM $b_i \times 10^6$ | May $b_i \times 10^6$ |
|---|---|---|---|---|---|---|
| 10 | 11 | 0.0000 | 0.0009 | 0.0000 | 5.30 | 4.48 |
| 11 | 12 | 0.0000 | 0.0005 | 0.0000 | 6.41 | 5.24 |
| 12 | 13 | 0.0000 | 0.0002 | 0.0000 | 7.68 | 6.07 |
| 13 | 14 | 0.0000 | 0.0001 | 0.0000 | 8.79 | 6.78 |
| 14 | 15 | 0.0000 | 0.0001 | 0.0000 | 10.42 | 7.79 |
| C | | 10.5 | 7.9 | 17.6 | | 20.0 |
| G | | 5.1 | 6.3 | 4.6 | | 4.6 |
| C/G | | 2.1 | 1.3 | 3.8 | | 3.8 |

**Appendix 7—table 4.** Genetic divergence between the Atlantic cod and walleye pollock sister taxa and rate of evolution.

The $p$-distance, the proportion of sites per nucleotide site that differ between the sister taxa Atlantic cod and walleye pollock (*Appendix 6—figure 20*) estimated with `ngsDist` (*Vieira et al., 2015*) setting the total number of sites (`--tot_sites`) equal to the number of sites that pass quality filtering in the estimation of site-frequency spectra (*Appendix 7—table 7*). The mutation rate $\mu$ which is the $p$-distance per nucleotide site per year are calculated under the assumption that these taxa diverged $3.5 \times 10^6$ years ago (*Vermeij, 1991*; *Vermeij and Roopnarine, 2008*; *Coulson et al., 2006*; *Carr and Marshall, 2008*). The number of substitutions per year, based on the number of sites in each chromosome (chromosomal length, last column), and its inverse, the number of years per substitution, are the rates for either lineage. Also given are the average over the chromosomes, and the whole-genome numbers. Based on the overall $p$-distances between the Atlantic cod sample from the South/south-east population (sample size $n = 68$) and a sample of 36 walleye pollock from a single locality in the Gulf of Alaska.

| Chromosome | $p$ per site | $\mu = p$ per site per year | Number of substitutions per year | Number of years per substitution | Number of sites |
|---|---|---|---|---|---|
| Chr01 | 0.00504 | $7.21 \times 10^{-10}$ | 0.022 | 45 | 30875876 |
| Chr02 | 0.00500 | $7.14 \times 10^{-10}$ | 0.021 | 49 | 28732775 |
| Chr03 | 0.00492 | $7.03 \times 10^{-10}$ | 0.022 | 46 | 30954429 |
| Chr04 | 0.00490 | $7.00 \times 10^{-10}$ | 0.031 | 33 | 43798135 |
| Chr05 | 0.00512 | $7.31 \times 10^{-10}$ | 0.018 | 54 | 25300426 |
| Chr06 | 0.00508 | $7.25 \times 10^{-10}$ | 0.020 | 50 | 27762770 |
| Chr07 | 0.00511 | $7.29 \times 10^{-10}$ | 0.025 | 40 | 34137969 |
| Chr08 | 0.00497 | $7.11 \times 10^{-10}$ | 0.021 | 47 | 29710654 |
| Chr09 | 0.00518 | $7.40 \times 10^{-10}$ | 0.020 | 51 | 26487948 |
| Chr10 | 0.00513 | $7.33 \times 10^{-10}$ | 0.020 | 50 | 27234273 |
| Chr11 | 0.00505 | $7.22 \times 10^{-10}$ | 0.022 | 45 | 30713045 |
| Chr12 | 0.00495 | $7.08 \times 10^{-10}$ | 0.022 | 46 | 30948897 |
| Chr13 | 0.00523 | $7.47 \times 10^{-10}$ | 0.022 | 46 | 28829685 |
| Chr14 | 0.00508 | $7.26 \times 10^{-10}$ | 0.021 | 47 | 29586942 |

*Appendix 7—table 4 Continued on next page*

*Appendix 7—table 4 Continued*

| Chromosome | $p$ per site | $\mu = p$ per site per year | Number of substitutions per year | Number of years per substitution | Number of sites |
|---|---|---|---|---|---|
| Chr15 | 0.00499 | $7.13 \times 10^{-10}$ | 0.020 | 49 | 28657694 |
| Chr16 | 0.00498 | $7.12 \times 10^{-10}$ | 0.025 | 40 | 34794352 |
| Chr17 | 0.00502 | $7.16 \times 10^{-10}$ | 0.016 | 64 | 21723002 |
| Chr18 | 0.00513 | $7.33 \times 10^{-10}$ | 0.018 | 55 | 24902675 |
| Chr19 | 0.00529 | $7.56 \times 10^{-10}$ | 0.017 | 60 | 22015597 |
| Chr20 | 0.00506 | $7.23 \times 10^{-10}$ | 0.018 | 56 | 24843429 |
| Chr21 | 0.00521 | $7.45 \times 10^{-10}$ | 0.017 | 60 | 22358821 |
| Chr22 | 0.00516 | $7.37 \times 10^{-10}$ | 0.018 | 57 | 23744039 |
| Chr23 | 0.00529 | $7.56 \times 10^{-10}$ | 0.019 | 52 | 25242006 |
| Average | 0.00508 | $7.26 \times 10^{-10}$ | 0.021 | 49 | 28406758 |
| Genome | 0.00507 | $7.25 \times 10^{-10}$ | 0.474 | 2 | 653355439 |

**Appendix 7—table 5.** A list of key terms and a brief description.

| Term | Description |
|---|---|
| High fecundity | The ability of organisms (e.g. broadcast spawners) to produce huge numbers of offspring, or on the order of the population size |
| Sweepstakes reproduction | High variance and high skew in the distribution of number of offspring, where most of the time individuals produce small (relative to the population size) number of offspring, but occasionally a few individuals contribute the bulk of the offspring forming a new generation of reproducing individuals |
| Random sweepstakes | A chance matching of reproduction in a highly fecund population with favorable environmental conditions; random sweepstakes is one example of a mechanism turning high fecundity into sweepstakes reproduction |
| Selective sweepstakes | A mechanism turning high fecundity into sweepstakes reproduction, in which juveniles pass through selective filters during their development, resulting in highly skewed offspring distribution |
| Moran model | A population model of genetic reproduction, in which a single random individual produces one offspring replacing another individual that perishes to keep the population size constant |
| Genealogy | The ancestral relations of a sample of gene copies (see *Appendix 6—figure 21*) |
| Coalescent | A probabilistic model of the random ancestral relations of a hypothetical sample of gene copies |
| Multiple-merger coalescent | A coalescent process in which a random number of ancestral lineages merges each time (see *Appendix 6—figure 21*) |
| $\Xi$-Beta $(2 - \alpha, \alpha)$-coalescent | A multiple-merger coalescent derived from a model of random sweepstakes |
| Durrett–Schweinsberg model | A model of recurrent selective sweeps of a new beneficial mutation each time approximating selective sweepstakes |
| Durrett–Schweinsberg coalescent | A coalescent model for the genealogy at a single site linked to a site experiencing beneficial mutation; during a sweep some lineages of the neutral site may escape a sweep through recombination (see *Appendix 6—figure 21*) |

**Appendix 7—table 6.** Approximate Bayesian computation (ABC) priors of parameter for various analysis.

| Parameter | ABC prior |
|---|---|
| $\alpha$ for the Beta $(2 - \alpha, \alpha)$-coalescent | Uniform between 1.01 and 1.99 |
| $\beta$, the growth rate for the Beta $(2 - \alpha, \alpha)$-coalescent with population growth | Improper, uniform prior on the whole positive half-line |
| $c$ for the single-locus DS model | Improper, uniform prior on the whole positive half-line |

*Appendix 7—table 6 Continued on next page*

*Appendix 7—table 6 Continued*

| Parameter | ABC prior |
| --- | --- |
| $c$ for the DS model with recombination | Uniform between 10 and 25 (to force consistency with the posterior in the single-locus analysis) |
| $\gamma/s$, the ratio of the recombination rate and the selection coefficient, in the DS model with recombination | Uniform between 0 and 10,000 |
| $\theta$, the mutation rate in the DS model with recombination | Uniform between 0 and 10,000 |
| Fraction of whole-chromosome sweeps in the DS model with recombination | Uniform between 0 and 1 |

**Appendix 7—table 7.** Genetic divergence between the Atlantic cod and walleye pollock sister taxa and rate of evolution from GL1 estimated site-frequency spectra.

The site-frequency spectrum of the South/south-east population of Atlantic cod estimated with `ANGSD` and genotype likelihood GL1 (*Korneliussen et al., 2014*), using walleye pollock (Gch) as outgroup to polarise the spectrum, gives all sites that pass quality filtering $L$, the number of invariant sites $I$, the number of segregating sites $S$, and the number of fixed sites $F$ between the focal population and the outgroup taxon. The number of substitutions per year and the number of years per substitution are calculated from fixed sites under the assumption that these taxa diverged $3.5 \times 10^6$ years ago (*Vermeij, 1991*; *Vermeij and Roopnarine, 2008*; *Coulson et al., 2006*; *Carr and Marshall, 2008*). The average over the chromosomes and the whole-genome numbers are also given. Compare to *Appendix 7—table 4* and *Appendix 7—table 8*.

| Chromosome | All sites, $L$ | Invariant sites, $I$ | Segregating sites, $S$ | Fixed sites, $F$ | Substitutions per year | Years per substitution |
| --- | --- | --- | --- | --- | --- | --- |
| Chr01 | 18350418 | 17736728 | 468247 | 145443 | 0.042 | 24 |
| Chr02 | 15850624 | 15269222 | 437440 | 143962 | 0.041 | 24 |
| Chr03 | 20231166 | 19467361 | 592044 | 171761 | 0.049 | 20 |
| Chr04 | 22623179 | 21742567 | 673837 | 206775 | 0.059 | 17 |
| Chr05 | 15557754 | 14963290 | 457852 | 136612 | 0.039 | 26 |
| Chr06 | 17738562 | 17090577 | 506727 | 141258 | 0.040 | 25 |
| Chr07 | 21107906 | 20282169 | 645738 | 180000 | 0.051 | 19 |
| Chr08 | 18381649 | 17681336 | 549023 | 151290 | 0.043 | 23 |
| Chr09 | 18212083 | 17528065 | 533448 | 150571 | 0.043 | 23 |
| Chr10 | 17472145 | 16829837 | 491408 | 150899 | 0.043 | 23 |
| Chr11 | 20157683 | 19439102 | 550466 | 168115 | 0.048 | 21 |
| Chr12 | 19475709 | 18838352 | 465219 | 172138 | 0.049 | 20 |
| Chr13 | 18669907 | 18002288 | 504278 | 163341 | 0.047 | 21 |
| Chr14 | 20723905 | 19946397 | 605101 | 172407 | 0.049 | 20 |
| Chr15 | 18123369 | 17435024 | 538832 | 149513 | 0.043 | 23 |
| Chr16 | 22268819 | 21460587 | 624520 | 183712 | 0.052 | 19 |
| Chr17 | 11831346 | 11376461 | 344921 | 109964 | 0.031 | 32 |
| Chr18 | 15955850 | 15348766 | 461840 | 145244 | 0.041 | 24 |
| Chr19 | 13869827 | 13314508 | 421341 | 133978 | 0.038 | 26 |
| Chr20 | 16390870 | 15807585 | 448550 | 134735 | 0.038 | 26 |
| Chr21 | 14455156 | 13911966 | 414247 | 128943 | 0.037 | 27 |
| Chr22 | 13854159 | 13314972 | 413965 | 125222 | 0.036 | 28 |
| Chr23 | 14714440 | 14154540 | 424496 | 135403 | 0.039 | 26 |
| Mean | 17652892 | 16997465 | 503197 | 152230 | 0.043 | 23 |
| Genome | 406016526 | 390941701 | 11573540 | 3501286 | 1.000 | 1 |

**Appendix 7—table 8.** Genetic divergence between the Atlantic cod and walleye pollock sister taxa and rate of evolution from GL2 estimated site-frequency spectra.

The site-frequency spectrum of the South/south-east population of Atlantic cod estimated with `ANGSD` and genotype likelihood GL2 (*Korneliussen et al., 2014*), using walleye pollock (Gch) as outgroup to polarize the spectrum, gives all sites that pass quality filtering $L$, the number of invariant sites $I$, the number of segregating sites $S$, and the number of fixed sites $F$ between the focal population and the outgroup taxon. The number of substitutions per year and the number of years per substitution are calculated from fixed sites under the assumption that these taxa diverged $3.5 \times 10^6$ years ago (*Vermeij, 1991*; *Vermeij and Roopnarine, 2008*; *Coulson et al., 2006*; *Carr and Marshall, 2008*). The average over the chromosomes and the whole-genome numbers are also given. Compare to *Appendix 7—table 4* and *Appendix 7—table 7*.

| Chromosome | All sites, $L$ | Invariant sites, $I$ | Segregating sites, $S$ | Fixed sites, $F$ | Substitutions per year | Years per substitution |
|---|---|---|---|---|---|---|
| Chr01 | 18350189 | 17645066 | 561297 | 143825 | 0.041 | 24 |
| Chr02 | 15850406 | 15184885 | 523368 | 142153 | 0.041 | 25 |
| Chr03 | 20230947 | 19356488 | 704986 | 169473 | 0.048 | 21 |
| Chr04 | 22622912 | 21614191 | 804994 | 203726 | 0.058 | 17 |
| Chr05 | 15557576 | 14877918 | 544750 | 134908 | 0.039 | 26 |
| Chr06 | 17738379 | 16995520 | 603445 | 139414 | 0.040 | 25 |
| Chr07 | 21107635 | 20164813 | 765533 | 177289 | 0.051 | 20 |
| Chr08 | 18381450 | 17578929 | 653357 | 149164 | 0.043 | 23 |
| Chr09 | 18211894 | 17429901 | 633282 | 148712 | 0.042 | 24 |
| Chr10 | 17471932 | 16736760 | 586177 | 148995 | 0.043 | 23 |
| Chr11 | 20157484 | 19333140 | 658299 | 166045 | 0.047 | 21 |
| Chr12 | 19475530 | 18739806 | 565949 | 169775 | 0.049 | 21 |
| Chr13 | 18669720 | 17904983 | 603304 | 161433 | 0.046 | 22 |
| Chr14 | 20723717 | 19835886 | 717441 | 170390 | 0.049 | 21 |
| Chr15 | 18123186 | 17334782 | 640824 | 147580 | 0.042 | 24 |
| Chr16 | 22268589 | 21341070 | 746271 | 181248 | 0.052 | 19 |
| Chr17 | 11831198 | 11309988 | 412831 | 108379 | 0.031 | 32 |
| Chr18 | 15955648 | 15261569 | 550677 | 143402 | 0.041 | 24 |
| Chr19 | 13869662 | 13237797 | 499507 | 132359 | 0.038 | 26 |
| Chr20 | 16390728 | 15721255 | 536397 | 133077 | 0.038 | 26 |
| Chr21 | 14454994 | 13833280 | 494317 | 127397 | 0.036 | 27 |
| Chr22 | 13853971 | 13237823 | 492661 | 123487 | 0.035 | 28 |
| Chr23 | 14714263 | 14074517 | 506140 | 133606 | 0.038 | 26 |
| Mean | 17652696 | 16902190 | 600252 | 150254 | 0.043 | 23 |
| Genome | 406012010 | 388750367 | 13805807 | 3455837 | 0.987 | 1 |

**Appendix 7—table 9.** Hardy–Weinberg test of PCA groups as inversion genotypes.

Observed $O$ and Hardy–Weinberg expected $E$ haplotype frequencies, allele frequency $p$, $X^2$ test statistic distributed as $\chi^2$, and probability $P$ of test statistic. Arranged by chromsome and by population. Based on the assumption that groups revealed by principal componenet analysis (PCA) represent composite genotypes of inversion haplotypes.

| Chromosome | PCA group | South/south-east | | | | | Þistilfjörður | | | | |
|---|---|---|---|---|---|---|---|---|---|---|---|
| | | $O$ | $E$ | $p$ | $X^2$ | $P$ | $O$ | $E$ | $p$ | $X^2$ | $P$ |
| Chr01 | AA | 7 | 7.44 | 0.33 | 0.06 | 0.80 | 31 | 28.52 | 0.63 | 1.60 | 0.21 |
| Chr01 | AB | 31 | 30.11 | | | | 28 | 32.96 | | | |

*Appendix 7—table 9 Continued on next page*

*Appendix 7—table 9 Continued*

| Chromosome | PCA group | South/south-east | | | | | Þistilfjörður | | | | |
|---|---|---|---|---|---|---|---|---|---|---|---|
| | | $O$ | $E$ | $p$ | $X^2$ | $P$ | $O$ | $E$ | $p$ | $X^2$ | $P$ |
| Chr01 | BB | 30 | 30.44 | | | | 12 | 9.52 | | | |
| Chr02 | CC | 41 | 30.76 | 0.76 | 0.69 | 0.41 | 36 | 39.56 | 0.75 | 4.99 | 0.03 |
| Chr02 | CD | 22 | 24.47 | | | | 34 | 26.87 | | | |
| Chr02 | DD | 5 | 3.76 | | | | 1 | 4.56 | | | |
| Chr07 | EE | 48 | 48.62 | 0.85 | 0.33 | 0.56 | 42 | 43.38 | 0.78 | 0.92 | 0.36 |
| Chr07 | EF | 19 | 17.76 | | | | 27 | 24.23 | | | |
| Chr07 | FF | 1 | 1.62 | | | | 2 | 3.38 | | | |
| Chr12 | GG | 62 | 61.13 | 0.96 | 0.14 | 0.70 | 62 | 61.35 | 0.93 | 1.38 | 0.24 |
| Chr12 | GH | 6 | 5.74 | | | | 8 | 9.30 | | | |
| Chr12 | HH | 0 | 0.13 | | | | 1 | 0.35 | | | |

**Appendix 7—table 10.** Genetic diversity and background selection simulations.
The genetic variation accumulated under different cases in SLiM (*Haller and Messer, 2019*) simulations of background selection (*Appendix 6—figure 19d*). In all cases a population of size $N = 10^5$ evolves according to the Wright–Fisher model assuming a chromosome segment of size $10^5$ bp with recombination rate $10^{-7}$ per site per generation that collects neutral or negative mutations with frequency $\mu = 10^{-7}$ per site per generation as now specified. Negative mutations were modelled as Gamma-distributed with a negative sign, with mean −0.1 (A, B, D) or −0.05 (C, E, F) all with shape parameter 0.2. The relative frequency of negative versus neutral mutations was 1:1 for (A, B, D) and 1:9 for (C, E, F). The points represent the logits of the normalized site-frequency spectrum of a random sample of 136 chromosomes (corresponding to the sample size of the South/south-east population) averaged over 100 (A, B, C, E) and 10 (D, F) replicates and taken after $10^5$ generations (B, E), $2 \times 10^5$ generations (A, C), and $10^6$ generations (D, F).

| Case | Average number of segregating sites | Average$\Pi$ | Average π per seg site |
|---|---|---|---|
| A | 8934.5 | 1257.0 | 0.14 |
| B | 7765.2 | 872.2 | 0.11 |
| C | 15568.8 | 2248.7 | 0.14 |
| D | 9896.6 | 1574.0 | 0.16 |
| E | 13001.8 | 1426.9 | 0.11 |
| F | 18857.7 | 3370.9 | 0.18 |

## Appendix 8

### Classic tests of neutrality

The classic Kingman coalescent, derived from the Wright–Fisher (or similar) model of low-variance reproduction, is the no-sweepstakes model. Several tests of a neutral equilibrium under the Wright–Fisher model of reproduction and the Kingman coalescent use a standardized difference of different estimators of $\theta = 4N_e\mu$ the mutation-rate scaled by population size (*Tajima, 1989*; *Fu and Li, 1993*; *Fay and Wu, 2000*; *Zeng et al., 2006*; *Przeworski, 2002*). These tests are sensitive to mutations on different parts of the genealogy and thus of different frequency classes of the site-frequency spectrum that also may be influenced by demography, background selection, and selective sweeps (*Tajima, 1989*; *Fu and Li, 1993*; *Fay and Wu, 2000*; *Zeng et al., 2006*; *Przeworski, 2002*). A negative Tajima's $D$ indicates an excess of low frequency over intermediate frequency alleles, and a negative Fu and Li's $D$, which contrasts mutations on internal and external branches of a genealogy, indicates an excess of singletons. Thus, these statistics are sensitive to deviations from neutrality affecting the left tail of the site-frequency spectrum, such as population expansion and background selection (*Nielsen, 2005*). In contrast, negative values of Fay and Wu's $H$ (*Fay and Wu, 2000*) and Zeng's $E$ (*Zeng et al., 2006*) statistics, which weigh the frequency of high-frequency derived alleles, are sensitive to deviations from neutrality affecting the right tail of the site-frequency spectrum such as positive selection and selective sweeps (*Fay and Wu, 2000*; *Przeworski, 2002*; *Nielsen, 2005*).

## Appendix 9

### Lifetable and generation time for Atlantic cod

Demographic life-table statistics for the female segment of the Atlantic cod in Iceland are presented in *Appendix 7—table 3*. $C$ and $G$ are, respectively, the correction factors for the effects of overlapping generations and generation time based on demographic estimation (*Jorde and Ryman, 1995*; *Jorde and Ryman, 1996*; *Laikre et al., 1998*). Age-specific survival rate, $l_i$, was based, respectively, on the average and the 1948–1952 and the 1963–1967 instantaneous mortality estimated from tagging experiments of Icelandic cod (*Jónsson, 1996*). These periods showed differences in estimated instantaneous survival and represented variation in demographic statistics. Mortality may be underestimated, particularly for Age class 1. The method assumes that the probability of survival from one year to the next is the same for all age classes (*Jónsson, 1996*). Age-specific fecundity is based on the average age-specific weight in the catch (*Anonymous, 2001*) and fecundity by weight relationships (*Marteinsdottir and Begg, 2002*) and similar relationships for Newfoundland cod for comparison (*May, 1967*). The demographic statistics were used to estimate generation time $G$ and the correction factor $C$ for the effect of overlapping generation by iteration of Equations 5–9 (*Jorde and Ryman, 1996*).

Few data are available on individuals older than 15 years so they are not included in the table. However, large fish up to 180 cm long, weighing up to 50 kg, and as old as 25 years are regularly caught. The annual fecundity of a 50-kg female is predicted to be between 3 and $4 \times 10^8$ eggs. Coupled with the low (type III) survivorship, large older females may contribute disproportionally to the variance in offspring numbers.

## Appendix 10

## SNP misorientation from parallel mutation or low-level ancestral admixture

In estimating site-frequency spectra as in this paper, potential effects of SNP misorientation from parallel mutation (*Baudry and Depaulis, 2003*; *Hernandez et al., 2007*) or low-frequency ancestral introgression (*Schumer et al., 2018*) can cause similar problems in the data, except that introgression is more genome-wide. We deal with these issues together. A parallel mutation in the outgroup to the same state as a derived biallelic polymorphic state in the ingroup will flip the orientation of a site with the ancestral state being considered derived. Singleton and doubleton sites are the most common sites and such sites would flip to the $n - 2$ and $n - 1$ class thus increasing the right tail of the SFS. Under low-level ancestral introgression sites that were formerly monomorphic in the ingroup would flip at sites that have a derived state in the outgroup. Introgression would not affect the right tail of the SFS at polymorphic ingroup sites where the outgroup carries the ancestral allele. Instead, such sites would have a higher frequency of ancestral allele and push derived alleles in the ingroup to a lower frequency and pull up the left side of the site-frequency spectrum. Ancestral introgression will affect ingroup sites that were fixed for the derived allele before introgression by making such sites polymorphic and contributing to the right tail of the site-frequency spectrum. To minimize the potential effects of introgression, we also screened the individuals sampled to ensure that they belong to the same groups revealed by principal component (PCA) and admixture analysis.

The tri-allelic filtering of sites that we apply in conjunction with multiple independent outgroups goes some way towards addressing these issues. This will only leave sites that have the same exact ancestral state in the outgroup as one of the two alleles in the ingroup. If a particular site can be polarized by outgroup A (e.g. Gma) it means that the state of the site in taxon A is the same as one of the alleles segregating in the ingroup population. If outgroup B (say Gch) has a different state for that site, the site would be tri-allelic in that comparison and removed by the tri-allelic filtering. We did not use parsimony or consensus to infer the state of ancestral nodes (*Keightley and Jackson, 2018*). Therefore, this filtering will not remove sites that have parallel changes simultaneously in two or three outgroup taxa.

To further address these issues, we reasoned that SNP misorientation and low-level ancestral introgression will mostly affect singletons and doubletons as well as the anti-singletons ($n - 1$) and anti-doubletons ($n - 2$) classes. The singletons and doubletons together comprise 62–66% of sites (depending on which genotype likelihood was used) whereas the anti-singletons ($n - 1$) and anti-doubletons ($n - 2$) classes comprise less than 1% of sites (see e.g. *Appendix 6—figure 3*). The right tail of the site-frequency spectrum is, therefore, sensitive to low levels of misorientation among singletons and doubletons. However, the truncated site-frequency spectrum compared to the full site-frequency spectrum and to the respective expectations of the Durrett–Schweinsberg model (*Figure 3—figure supplement 4*) does not change the overall pattern.

Second, we reasoned that sites with transition variation are more likely than transversions to be saturated with mutations (*Agarwal and Przeworski, 2021*) complicating the polarization of sites. Parallel changes are more likely at such sites leading to SNP misorientation. To address this, we removed transitions and fitted the Durrett–Schweinsberg model to transversions only. This had minuscule effects compared to full data (*Figure 3—figure supplement 4*).

Third, we studied the effect of outgroups. Instead of maximum parsimony across the cod species tree, we used a 100% consensus sequence of walleye pollock (Gch), Pacific cod (Gma), and Arctic cod (Bsa) as an outgroup. Thus only sites at which the three outgroup taxa agree are used. Under parsimony, an agreement among two out of three would be used, but here three out of three are required (*Figure 3—figure supplement 1*). It is worth noting that the very right tail of the site-frequency spectrum is lower and more in line with the Durrett–Schweinsberg model for both the transversions and the 100% consensus data compared to the full data (*Figure 3*). This probably indicates that SNP misorientation had some effect on the original full analysis. However, the effect does not change the results qualitatively.

## Appendix 11

### Can processes other than selective sweeps better explain the patterns?

The effects of demography (changes in population size, population structure, and migration) can be hard to distinguish from various forms of selection (*Nielsen, 2005*). Different forms of selection can affect the various parts of the site-frequency spectrum in similar ways. We now consider whether processes other than selective sweeps can provide a better explanation for the observed patterns.

## Appendix 12

### Historical demography and low variance reproduction

Our estimated demographic history (*Appendix 6—figure 11* and *Appendix 6—figure 12*) shows population expansion in the distant past leading to the relative stability of population size in the recent past to modern times. In some cases, an apparent population crash in recent times (*Appendix 6—figure 11c*), which is chromosome specific, is an exception to this. Demography produces genome-wide effects and, thus, this is likely a peculiarity of runs of homozygosity of some chromosomes (such as centromeric regions, for example) and does not reflect historical size changes of the population. Based on these population growth curves (*Appendix 6—figure 11* and *Appendix 6—figure 12*) we generated population size change scenarios for simulating site-frequency spectra using `msprime` (*Kelleher et al., 2016*; *Baumdicker et al., 2021*). The results (*Appendix 6—figure 19*) show monotonically decreasing frequency with the size of the mutation or L-shaped site-frequency spectra that neither capture the singleton class nor the upswing of the right tail of the observed site-frequency spectra (*Figure 3*, *Appendix 6—figure 3*, and *Appendix 6—figure 5*). Thus, there is no evidence in our results for a low-variance no-sweepstakes mode of reproduction modelled by the Kingman coalescent, even taking demographic histories of population expansion or collapse into account. Our simulations are in line with the theoretical proof (see Appendix 6 of *Sargsyan and Wakeley, 2008*), showing that the normalized expected site-frequency spectrum of a Kingman coalescent under arbitrary population size history is L-shaped.

## Appendix 13

### Potential confounding due to cryptic population structure

Here we examine alternative explanations for our observations. In particular, are the site-frequency spectra influenced by cryptic population structure?

The effect of hidden population structure on the site-frequency spectra is expected to look similar to the patterns seen for the inversion chromosomes. These are chromosomes Chr01, Chr02, Chr07, and Chr12 known to carry large inversion (*Kirubakaran et al., 2016*; *Berg et al., 2016*). They show two peaks in the site-frequency spectrum (*Appendix 6—figure 18* and *Appendix 6—figure 18*) at the frequency of the variants' haplotype frequency and show a block of values for neutrality statistics (*Figure 1* and *Appendix 6—figure 2*). If a sample of size $n$ diploid organisms is composed of two cryptic reproductively isolated populations (sample sizes $n_1$ and $n_2$) we expect to see peaks in the site-frequency spectra at the relative frequencies of the two groups. If $n_1 = n_2 = n$ we expect a sharp peak at $n/(2n)$. This peak would include all fixed sites in both populations ($n_1/n$ and $n_2/n$) and spread over neighbouring frequency classes ($(n_1 - 1)/n, (n_1 - 2)/n, (n_2 - 1)/n, (n_2 - 2)/n$ and so on). If the frequencies of the two groups differ ($n_1 \neq n_2$) two peaks will appear, but are expected to be narrow. They will always include all sites fixed in either population (because fixed sites in either population will appear to be segregating in the sample as a whole).

To study the potential effects of population structure, we used `msprime` (*Kelleher et al., 2016*; *Baumdicker et al., 2021*) to simulate the Kingman coalescent with two isolated populations exchanging a varying number of migrants under population growth as determined by the growth parameter $\beta$. Thus, we examined the effects of cryptic structure on the site-frequency spectrum by varying the growth rate and the effective number of migrants between subpopulations ($4N_e m$), and varied the number of individuals sampled from the population with fewer individuals represented (referred to as the minor population). Parameters of the simulations were the number of individuals from the minor population ($k \in \{4, 3, 2, 1\}$), the migration rate ($m = 10^{-5} \ldots 10^{-3}$), and the growth rate ($g = 10^{-4} \ldots 10^{-1}$). The effective size was set at $N_e = 500$ and thus the effective number of migrants per subpopulation per generation was $4N_e m = 0.02 \ldots 2$.

We use a two-island model with exponential growth under the Kingman coalescent as a simple tool for assessing the qualitative, joint effect of demography and substructure on the site-frequency spectrum (*Appendix 6—figure 14* and *Appendix 6—figure 15*). Two narrow peaks at opposite allele frequencies are evident (much like the two narrow peaks for the inversion chromosomes, *Appendix 6—figure 18* and *Appendix 6—figure 18*) becoming smaller with increasing migration. If the sample contained only a single individual from the minor population, is there a remote resemblance to the observed data (*Appendix 6—figure 14g, h, j*). Nevertheless, even in this case, doublets are more common than singletons, and it is hard to find combinations of growth and migration rates to mimic the observed data. We used the Xi-Beta coalescent for similar simulations (*Appendix 6—figure 15*) and got the same results qualitatively. Therefore, population structure in a population evolving according to the Wright–Fisher (or a similar) low-fecundity model or in a population evolving under a neutral sweepstakes model is an improbable explanation for our results. Both simulations (*Appendix 6—figure 14* and *Appendix 6—figure 15*) show that only for a minor sample size of one diploid individual do the models show a remote resemblance to our data. To further address this issue, we, therefore, estimated the site-frequency spectra with a leave-one-out approach (*Appendix 6—figure 16*). The leave-one-out approach is model free: whichever model is correct, one of the leave-one-out samples should behave differently if a cryptic population structure with a minor sample size of one is present in our data. None of them do. There is no indication that our sample from the South/south-east coast is composed of 67 individuals from one population and a single individual from a divergent population.

To further study the potential effects of cryptic population structure, we note that PCA of variation at each of the four chromosomes harboring large inversions reveals two or three groups that likely represent genotypes of the inversion alleles. There are three groups for Chr01 (which we refer to as Chr01-*AA*, Chr01-*AB*, and Chr01-*BB*), Chr02 (Chr02-*CC*, Chr02-*CD*, and Chr02-*DD*), and Chr07 (Chr07-*EE*, Chr07-*EF*, and Chr07-*FF*), and two groups for Chr12 (Chr12-*GG* and Chr12-*GH*), which has a low frequency of one inversion allele (*Appendix 6—figure 13*, *Appendix 6—figure 18*, and *Appendix 6—figure 18*). If we take these groups as representing the haplotypes of the inversions,

the genotypic frequencies at each chromosome do not deviate from Hardy–Weinberg equilibrium, and there is thus no evidence for breeding structure (no Wahlund effect, *Appendix 7—table 9*). However, as the inversions effectively suppress recombination between the inversion alleles, we can also look at the chromosomes of the inversion genotypes as effectively isolated populations with no recombination (migration) between them and estimate the site-frequency spectra within genotypes for the inversion chromosomes. Furthermore, we conjecture that the PCA groups observed at inversion chromosomes represent reproductively isolated but cryptic populations. Because demography has genome-wide effects, the cryptic structure should be evident in the rest of the genome. We, therefore, estimate the site-frequency spectra for the 19 non-inversion chromosomes (chromosomes 3–6, 8–11, and 13–23) for these groups.

PCA did not show any structure for the non-inversion chromosomes. However, the four inversion chromosomes each showed two narrow peaks at intermediate allele frequencies (*Appendix 6—figure 18* and *Appendix 6—figure 18*) indicative of either balancing selection or cryptic population breeding structure. If this is a breeding structure it should affect the whole genome. To disentangle the effects of balancing selection and potential breeding structure, we used the groups defined by PCA at the inversion chromosomes to investigate the inversion chromosomes themselves and the non-inversion chromosomes. We thus conjecture that the PCA groups represent cryptic breeding units.

PCA revealed three (or two) groups on the first principal axis that explains 4–36% of the variation in the inversion chromosomes (*Appendix 6—figure 13a, d and g*, j). The PCA groups most likely represent genotypes of inversion haplotypes. Taking membership in PCA groups to represent inversion genotype, their frequencies fit the Hardy–Weinberg equilibrium (*Appendix 7—table 9*) and thus there is no evidence of heterozygote deficiency or Wahlund effect (*Wahlund, 1928*) indicative of breeding structure. The only exception is chromosome 7 in the Þistilfjörður population, which shows a slight heterozygote excess (*Appendix 7—table 9*). Furthermore, the site-frequency spectra of the PCA groups (*Appendix 6—figure 13b, e and h, k*) show the same overall V-shape pattern as the site-frequency spectra for the overall data (*Figure 3*). Additionally, the intermediate PCA group shows a sharp peak at a derived allele frequency of $n/(2n)$ (an equal frequency of two types or 0 on the logit scale) as expected for a group composed of heterozygotes only. Similarly, the site-frequency spectra of these PCA groups for the 19 non-inversion chromosomes combined (*Appendix 6—figure 13c, f, i, l*) show a pattern characteristic of the site-frequency spectra for the overall data. There is not the slightest hint of a Kingman coalescent-like behaviour for any of these PCA groups. Similarly, the expectations of the $\Xi$-Beta$(2 - \alpha, \alpha)$ coalescent do not explain the data.

Overall, the shape of the site-frequency spectra for each of the inversion chromosomes (*Appendix 6—figure 18* and *Appendix 6—figure 18*) and the PCA groups of each of the inversion chromosomes (*Appendix 6—figure 13*) is the same as the shape of the site-frequency spectra of the non-inversion chromosomes (*Figure 3*). This shape is well explained for all PCA groups by the Durrett–Schweinsberg coalescent (*Durrett and Schweinsberg, 2005*), for which we estimated the $c$ parameter using ABC for the PCA group of the respective inversion chromosomes (*Appendix 6—figure 13*).

The observed V-shaped site-frequency spectra are inconsistent with an amalgamation of cryptic units reproducing under a Wright–Fisher model. The PCA groups are not cryptic breeding units, and we reject the above conjecture. Instead, we consider them to represent polymorphic inversion genotypes maintained by some form of balancing selection, such as frequency-dependent fitnesses arising from the accumulation of deleterious recessives on homokaryotypes (*Jay et al., 2021*) or other mechanisms of balancing selection (*Faria et al., 2019*). The good fit of the Durrett–Schweinsberg model of selective sweeps to the overall site-frequency spectra of the inversion chromosomes (*Appendix 6—figure 18* and *Appendix 6—figure 18*) and to the PCA groups representing the alternative haplotypes of each of the inversion chromosomes (*Appendix 6—figure 13*) likely indicate recurrent selective sweeps within the alternative haplotypes of chromosomal inversions. It is known that both haplotypes of the *Pan*I locus, which is located close to a breakpoint of the chromosome 01 inversion (*Kirubakaran et al., 2016*), are subject to selective sweeps in action (*Pogson, 2001*).

## Appendix 14

### Inversion polymorphisms

Four chromosomes, Chr01, Chr02, Chr07, and Chr12, are known to carry large inversions (*Kirubakaran et al., 2016*; *Berg et al., 2016*). The two inversions on Chr01 are connected to ecotypic variation, defining a deep-water migratory ecotype and a shallow-water stationary ecotype (*Pampoulie et al., 2007*), and inversions on the other three chromosomes may also be involved (*Berg et al., 2016*). The polymorphic Chr01 inversions likely originated in Iceland to Barents Sea populations as revealed by graph-aware retrieval of selective sweeps (*Refoyo-Martínez et al., 2019*). Similarly, the Chr02 and Chr07 inversions likely originated in Iceland, Faroe Islands, and North Sea populations, while the Chr12 inversion polymorphism originated in the Celtic Sea population (*Refoyo-Martínez et al., 2019*).

The polymorphic inversions on Chr01, Chr02, and Chr07 are segregating at intermediate frequencies (*Appendix 6—figure 18* and *Appendix 6—figure 18*) (and see *Hemmer-Hansen et al., 2013*) in our two sample populations in Iceland (*Appendix 6—figure 1*) while the Chr12 inversion polymorphism is at about 5% versus 95% in the Þistilfjörður population and rarer still in the South/south-east coast population (*Appendix 6—figure 17* and *Appendix 6—figure 18*). The inversion polymorphisms in these chromosomes are likely to be maintained by some form of balancing selection or they are examples of cryptic breeding structure (*Hemmer-Hansen et al., 2013*). To avoid the effects of balancing selection or cryptic breeding structure on our analysis, we exclude these four chromosomes from analysis or analyze them separately. The other 19 chromosomes (Chr03–Chr06, Chr08–Chr11, and Chr13–Chr23) do not seem to harbor large inversions or other significant chromosomal structural variations. We refer to them as non-inversion chromosomes.

## Appendix 15

## Balancing and background selection and functional constraints

Besides the Durrett–Schweinsberg model, various mechanisms of selection may influence the results. Here, we examine the effects of balancing selection, different selective constraints, and background selection.

There are several signs that natural selection affects observed patterns. Balancing selection retains linked neutral or nearly neutral variants at intermediate frequencies. The tighter the linkage and less the recombination, the longer the coalescent time of the neutral variants (*Charlesworth, 2006*). The observed site-frequency of intermediate frequency alleles is higher among the four inversion chromosomes than the 19 non-inversion chromosomes. All comparisons of the four inversion chromosomes and the 19 non-inversion chromosomes show this effect (*Appendix 6—figure 17*, *Appendix 6—figure 18* and *Appendix 6—figure 3*). However, balancing selection does not affect the overall V-shape of the site-frequency spectrum of the inversion chromosomes (*Appendix 6— figure 17* and *Appendix 6—figure 18*).

The PC-based selection scan (*Meisner and Albrechtsen, 2018*) is model-free and is based on finding genes or genomic regions that are outliers relative to the overall genome-wide allele frequencies and taking potential population structure into account. A principal component-based genomic scan of selection (*Appendix 6—figure 8*) showed many peaks that are likely indicative of recent and strong positive selection. Few peaks were population specific, but the two populations share most peaks. Region under a peak ranged from a single site to about 2 Mb. We extracted sites 500 kb or more away from the peaks (referred to as no-selection) and included with genomic regions under different selective constraints. We extracted fourfold degenerate sites, introns, and intergenic sites as less constrained regions, promoter regions, exons, 3′-UTR, and 5′-UTR as more selectively constrained regions. The mean, median, and mode of the estimated compound parameter $c$ of the Durrett–Schweinsberg model for the different genomic regions ranked from least constrained to most constrained sites (*Figure 5*). The ABC-MCMC was well mixed in all cases. There are two possible explanations for the rank order of the compound parameter $c$ with functional genomic regions. First, the more functionally important a region of the genome is, the stronger the selection coefficient of a new advantageous mutation will be as observed for UTRs in *Drosophila* (e.g. *Andolfatto, 2005*; *Sella et al., 2009*). Such mutations will sweep through the population and carry with them tightly linked neutral mutations in these same regions ($c$ being inversely proportional to the recombination rate $\gamma$). Alternatively, different functional regions are preserved and constrained by purifying (negative) selection. If the sites are tightly linked, a positively selected mutation sweeping through will affect neutral, nearly neutral, and even deleterious sites. A tug-of-war between the effects of the sweep and purifying selection at a site results in a net effective selection coefficient for that site. The compound parameter $c$ of the Durrett–Schweinsberg model estimates the net effective selection coefficient squared over the recombination rate or density of selection per map unit (*Aeschbacher et al., 2017*), which may generate the observed rank order. Of course, both explanations may apply to different positive mutations. Thus selective sweeps permeate the genome affecting most if not all sites (*Pouyet et al., 2018*).

To study the effects of background selection, we carried out forward-in-time simulations of the Wright–Fisher model (using SLiM *Haller and Messer, 2019*). Simulations that ran for a relatively short number of generations (on the order of population size) produced V-shaped site-frequency spectra (*Appendix 6—figure 19d*). However, when simulations of the same parameter values ran for a large number of generations (up to 10 times the population size of $10^5$ chromosomes) they accumulated more variation (*Appendix 7—table 10*) and produced monotone L-shaped site-frequency spectra. Thus, only in a narrow window of non-equilibrium between the input of mutation and its removal by purifying selection or loss by drift can background selection site-frequency spectra resemble our observed spectra. In general, however, background selection does not fit our data.

## Appendix 16

### The joint action of several evolutionary mechanisms

The analysis thus has shown that, considered singly, the various factors such as demography and background selection do not provide a good fit, particularly not involving the derived alleles at the right tail of the site-frequency spectrum. Studying the site-frequency spectrum under recurrent selective sweeps, *Kim, 2006* stated that 'the excess of high-frequency derived alleles, previously shown to be a signature of single selective sweeps, disappears with recurrent sweeps.' This effect is sometimes—incorrectly—taken to mean that the site-frequency spectrum is no longer U-shaped under recurrent selective sweeps. However, the excess or deficiency of high-frequency derived alleles is in reference to expectations of the Kingman coalescent (*Kim, 2006*) and how that affects Fay and Wu's $H$ statistic (*Fay and Wu, 2000*). The site frequencies of alleles at intermediate allele frequencies (the alleles contributing most to the variance in fitness) are still reduced under recurrent sweeps (*Kim, 2006*) preserving the U-shaped site-frequency spectrum observed under a single selective sweep. Analyzing the joint action of demography, purifying and background selection with or without random sweepstakes on the genome level is computationally prohibitive. We, therefore, resorted to simulations using SLiM (*Haller and Messer, 2019*) of a sizeable fragment of a chromosome evolving under the joint action of several mechanisms of evolution (*Appendix 6—figure 19*). As is common in complex, multi-component simulations, it may be possible to tweak parameters to obtain results matching the observed data. Nevertheless, a comprehensive model-fitting search is infeasible in our setting. However, the combined effect of negative background selection without selective sweeps did not produce qualitatively accurate, U-shaped site-frequency spectra for any parameter combination we tested. Furthermore, a combination of random sweepstakes, randomly occurring bottlenecks, and background selection (*Appendix 6—figure 19e, f*) did not produce a qualitatively similar U-shaped pattern as the data. Hence, even if best-fit parameters could match the data, we expect the fit would not be robust to small changes in either parameter values or observed data, thus having low predictive and explanatory power. In contrast, scenarios involving selective sweeps routinely produced the right qualitative shape of the site-frequency spectra. Hence, we expect a (hypothetical) best-fit analysis to be far more robust.

## Appendix 17

### Rates of selective sweeps and genomic footprints

The average nucleotide distance between the Atlantic cod and walleye pollock sister taxa was 0.00508 nucleotide substitutions per nucleotide site for the average chromosome and the genome as a whole (*Appendix 6—figure 20* and *Appendix 7—table 4*). On the assumption that Atlantic cod and walleye pollock split at the opening of the Bering Strait 3.5 million years ago (*Vermeij, 1991*; *Vermeij and Roopnarine, 2008*; *Coulson et al., 2006*; *Carr and Marshall, 2008*) the mutation rate is $7.26 \times 10^{-10}$ substitutions per nucleotide site per year, about an order of magnitude lower than the mtDNA rate of $1 \times 10^{-8}$ (*Carr and Marshall, 2008*). This translates into a mutation rate of $3.63 \times 10^{-9}$ nucleotides per nucleotide site per generation. Thus, there are $0.00508 \times 28,406,758/(3.5 \times 10^{6}) = 0.021$ substitutions (where 28,406,758 is the average length of a chromosome *Appendix 7—table 4*) per chromosome per year since the divergence of the two taxa or roughly one substitution per chromosome every 50 years (*Appendix 7—table 4*). For the genome as a whole, there is roughly one-half substitution per year or a substitution every 2 years on average. With a 5-year generation time (Appendix 9 and *Appendix 7—table 3*), the average chromosome has a substitution every 10 generations and the Atlantic cod genome has 2.5 substitutions per generation.

The analysis of site-frequency spectra using `ANGSD` (*Korneliussen et al., 2014*) yields an estimate of the number of invariant sites, the number of segregating sites, and the number of fixed sites between the outgroup used to polarize the spectrum and the focal taxon (*Appendix 7—table 7* and *Appendix 7—table 8*). Based on the number of fixed sites using walley pollock (Gch) as an outgroup we estimate that within Atlantic cod there have been 0.043 substitutions per average chromosome per year and one substitution per genome per year. This translates to one substitution every 23 years or four to five generations for the average chromosome and one substitution in the whole genome per year or five substitutions per generation (*Appendix 7—table 7* and *Appendix 7—table 8*). The rates of evolution obtained with this approach are similar to the rates above using the divergence of the taxa.

We estimated the diversity statistics Watterson's $\theta_W$ and average pairwise nucleotide diversity $\hat{\pi}$, which is an estimate of $\theta_\pi$, as well as the neutrality statistics Tajima's $D_T$ and Fu and Li's $D_F$ for the South/south-east and Þistilfjörður populations respectively (*Appendix 7—table 1* and *Appendix 7—table 2*). The neutrality statistics were significantly negative as expected under the Durrett–Schweinsberg coalescent model of recurrent selective sweeps (*Durrett and Schweinsberg, 2005*). The average pairwise nucleotide diversity is $\hat{\pi} = 0.0019$ per nucleotide site in the South/south-east populations and slightly higher $\hat{\pi} = 0.0029$ for the Þistilfjörður population (*Appendix 7—table 1* and *Appendix 7—table 2*).

The average pairwise nucleotide diversity in the South/south-east population $\hat{\pi} = 0.0019$ (*Appendix 7—table 1*), which is the average number of differences between pairs of sequences looking forward, is a natural proxy for the mean time until a pair of lineages coalesces when looking backwards under both the Kingman and the Durrett–Schweinsberg model. Under the Durrett–Schweinsberg model with the range of parameters that describes our data ($c \approx 6$–19), most of these pairwise mergers are caused by sweeps, and hence the mean time between sweeps will be commensurate to the mean time until a pair coalescence. This is likely to be lower than the rate of all sweeps per chromosome because not every sweep happens close enough to the given site to be likely to cause a merger. In one extreme, the selective advantage might be extreme in comparison to recombination, so that the sweep typically catches most or all of the chromosome, in which case the calculated rate of sweeps is about right. Alternatively, it could be that recombination is more potent than selection, in which case only a short region of genome hitchhikes with each sweep. In this case, the actual rate of sweeps per chromosome would be higher. Identifying a more detailed rate would be equivalent to teasing apart the components of the parameter $c = \delta s^2/\gamma$ of the Durrett–Schweinsberg model.

The Durrett–Schweinsberg model is essentially a model of a single locus on a single chromosome. We estimate that each chromosome in Atlantic cod is affected by a sweep every 23–50 years on average or every 4–10 generations. Since we also see evidence of rapid recombination (*Appendix 6—figure 6*), we expect that one sweep will not have a substantial effect on a large region of a chromosome. Recombination is high enough that linkage disequilibrium decays to the background over 25–100 kb (*Appendix 6—figure 6*). We see similar pattern evidence for selective sweeps for different-sized

chromosomal fragments, for different functional regions, and for all chromosomes (*Figure 5* and *Appendix 6—figure 13 Appendix 6—figure 17* and *Appendix 6—figure 18*). Thus, genomic footprints of all but powerful selective sweeps will be relatively short. There is thus clear evidence that sweeps happen everywhere along the genome, and it is therefore likely that the actual rate of sweeps is even faster than our estimate. For example, if an average sweep were to affect only 10% of a chromosome, then we expect sweeps every year or so on the average chromosome to explain our results. Building a fully quantitative, data-informed picture of the rate of sweeps requires the development of a diploid, genomic version of the Durrett–Schweinsberg model, which is currently absent from the literature, and for which task our results provide strong applied motivation.

The census population size of Atlantic cod in Iceland may be a billion to a trillion ($10^9$ to $10^{12}$) fish. However, the effective population size is much smaller. Suppose a molecular clock dates the most recent common ancestor (MRCA) of two sequences at $X$ years, which results in a Kingman effective population size of $N_e$. The mutation scaled population size $\theta = 4N_e\mu$ is estimated with pairwise nucleotide diversity $\pi = \theta_\pi$ and with Watterson's estimator $\theta_W$ (*Appendix 7—table 1* and *Appendix 7—table 2*). Assuming a mutation rate of $3.62 \times 10^{-9}$ per generation and the Kingman coalescent the effective size of, for example, the South/south-east population ranges from $0.0019/(4 \times 3.62 \times 10^{-9}) \approx 130,000$ to $0.0062/(4 \times 3.62 \times 10^{-9}) \approx 427,000$ depending on which parameter estimate we use (*Appendix 7—table 1*). The corresponding Durrett–Schweinsberg population size is $D = (1 + c) \times N_e$, because selective sweeps increase the pair-merger rate from 1 to $1 + c$. In our case, $D$ is ≈6–19 times larger than $N_e$ of the classical Kingman coalescent, so the discrepancy between the census population size and the population size we need to plug into the model has been reduced (though of course $D$ will still be much less than the census size of a billion). Moreover, if a pair of lineages merges after $X$ years on average, and $c/(1 + c)$ of mergers are caused by selective sweeps, then the rate of effective sweeps (sweeps that cause at least one merger at a given site) is one per $(1 + c)X/c$ years.

We can also approach the question of whether one can plug the parameter estimates obtained using the coalescent model back into the model and recover the genetic variation used to estimate those very parameters. In this vein, we ask what population size $N$ is required to recover the number of segregating sites on an average chromosome under the Durrett–Schweinsberg model. Standard coalescent theory arguments assuming the infinitely many sites mutation model given that the expected number of segregating sites $S$ on an average chromosome is $S = \mu \times L \times (1/c_N) \times \mathbb{E}[\text{tree size in coalescent units}]$. Here, $\mu$ is the mutation rate per site per generation and $L$ is the number of sites of an average chromosome. In the Kingman coalescent, time is rescaled using $1/c_N$ generations as one coalescent time unit where $c_N$ is the probability that two individuals picked at random in a generation are descended from the same parent in the previous generation. Thus $c_N = 1/N_e$ for the Kingman coalescent, where $N_e$ is the effective population size. For the Durrett–Schweinsberg model, we have $1/c_N \approx N$ the actual population size, and we write $\mathbb{E}[\text{tree size in coalescent units}] =: f(n, c)$, where $f(n, c)$ is a function of the sample size $n$ and the compound parameter $c$ of the Durrett–Schweinsberg model. Thus, the minimum population size $N$ required to account for our results is $N = S/(\mu \times L \times f(n, c))$.

The numbers to plug into the equation are $\mu = 3.63 \times 10^{-9}$ per nucleotide site per generation (*Appendix 7—table 4*), the number of segregating sites $S = 503,197$, and the total number of sites $L = 17,652,892$ (*Appendix 7—table 7* and see *Appendix 7—table 8*), and the function $f(n, c) = f(136, 10) = 1.07$ using the sample size of chromosomes $n = 136$ of the South/south-east population and the average $c = 10$ based on a recursion adapted for the Durrett–Schweinsberg model from a recursion for a general $\Lambda$ coalescent (*Birkner et al., 2013b*). Thus, the population size required to account for our observations is $N = 8 \times 10^6$ or roughly $N = 10^7$. This estimate is well within reasonable limits (1 billion cod) and leaves ample room for an even more rapid rate of evolution by selective sweeps than we have observed.

The compound parameter $c = \delta s^2/\gamma$ can be thought of as a density of selection along the chromosome (*Aeschbacher et al., 2017*). The (numerical) derivative of $c$ with respect to the recombination unit would yield an effective local selection coefficient. However, estimating this in windows along the chromosome is too noisy. However, since the number of SNPs is proportional to branch length, which is proportional to $c$, it is legitimate to regard these as factors by which to multiply the single $c$ already estimated in each 25-kb window. *Appendix 6—figure 22* shows

the relative diversity as an indication of the density of selection along chromosome 4 (the largest chromosome). The selective effects are distributed thoughout the chromosome similar to the results of genome scans for selection (*Appendix 6—figure 6*).

