## [Editor Report]

This fundamental work significantly advances our understanding of genetic diversity in highly fecund organisms by showing that the Atlantic cod genome is prone to recurrent selective sweeps. The evidence supporting these conclusions is compelling, with rigorous analysis of the site frequency spectrum providing support for models of selective sweepstakes reproduction and multi-merger coalescents. This work will be of broad interest to evolutionary geneticists and should stimulate future work to determine whether random sweepstakes interact with selective sweepstakes.

---

## [Decision Letter]

**Decision letter after peer review:**

Thank you for submitting your article "Sweepstakes reproductive success via pervasive and recurrent selective sweeps" for consideration by *eLife*. Your article has been reviewed by 3 peer reviewers, including Pierre-Alexandre Gagnaire as Reviewing Editor and Reviewer #3, and the evaluation has been overseen by Molly Przeworski as the Senior Editor. The following individual involved in the review of your submission has agreed to reveal their identity: Fabian Freund (Reviewer #2).

Essential revisions:

The three reviewers were generally enthusiastic. They agreed that the study could have a strong impact in the field of population genetics and a potential for application to a broad range of taxa beyond marine fishes. While the overall assessments were positive, the reviewers also made a series of complementary comments to help you to better support your claims, or instead moderate some of your conclusions in the discussion. While the paper cannot be published as it stands, addressing these comments should be manageable without the need for extensive additional analysis.

Below is a list of essential revisions that should help your to alter the claims or make clear that some conclusions will need supporting data in the future.

1) A priority task concerns the haploid Λ-Β coalescence model of random sweepstakes (reviewers #1 and #2). It is not clear from the manuscript whether the excellent fit of the Λ-Β-Coalescent model is actually better than the fit of the DS model. The authors should test or explain which model fits better.

If it is the Λ-Β-Coalescent model, they should explain why the Λ-Β fit is so good and why the DS model was chosen despite a worse fit (e.g. if other parameters such as LD disagree with Λ-Β) – or restate their main result. Alternatively, if the DS model fits best among all coalescent models, the authors should not argue that the better fit of Λ-Β over Xi-Β is due to selection akin to selection in the DS model (see reviewer's comment #2 on this point).

2) Some of the main findings also need further verification to be more robust. The additional checks and analyses requested include the following:

– Provide a genome-wide estimate of NI, or alternatively alfa, the proportion of adaptive non-synonymous substitutions (Reviewers #1 and #3).

– Evaluate potential issues related to SNP misorientation (Reviewers #1 and #3).

– Assess the impact of low-frequency introgressed ancestry on the shape of the SFS (Reviewer #3).

3) Several comments relate to points that do not require further analysis but rather a clarification in the discussion, including:

– The need to reconcile the role of high fertility in shaping diversity with the assumptions of the DS model, and the related issue of potential effects on the time scaling (Reviewers #2).

– The cost of selection, and the biological meaning of the compound parameter with respect to the nature of selective sweeps (Reviewers #1 and #3).

– Possible limitations of sweepstakes models with respect to characteristics of the Atlantic cod mating system (Reviewer #3).

4) Finally, there are also requests for revision on the grounds of clarity, conciseness, and presentation. The details are provided in the reviews (mostly Reviewer #1), so I only mention the most important points that would require:

– Making the content more accessible to non-experts readers, keeping in the main text only aspects that could be understood without looking at the Sup Mat, and completing with a more thorough and detailed analysis in the Supp Mat when necessary.

– Clarifying definitions and simplifying figure legends, selecting only one representative panel for some figures, and making references to supplementary material easier to follow.

– Reducing the last section of the results that considers alternative explanations that do not fit the data.

*Reviewer #1 (Recommendations for the authors):*

My major comment does not concern the science aspect of this manuscript but encourages the authors to revise in order to ease the reading. I found it sometimes difficult due to numerous references to supplementary material all along the text, that figure legends can be clarified, and sometimes the text lacks a clear definition of words. Furthermore, I found the article quite lengthy in the last section (all the alternative explanations that do not fit the data). I believe that an improvement is possible in making the content more accessible to non-experts so that a wide range of *eLife* readers will be motivated in getting through this article.

Second, did the authors consider a potential issue with misorientation of variants, where some derived are considered ancestral and vice-versa? This has been put forward as a plausible explanation to explain the U-shaped SFS. Although I suspect the fit in the case studied here is so good with \Λ-MMC or S.D.-MMC, that orientation errors are not a real issue. It may however be nice to mention/discuss it somewhere in the text.

Figures: There are often many almost identical figures among the panels. Please consider selecting only one and mention that the other ones are similar. It will also simplify (and clarify) the figure legends.

Figure 3 is quite ugly. Can the authors display some kind of density plot with iso-density lines?

In the following, I list a few suggestions that could help clarify the text to broaden, as much as possible, the audience. But this can be done more generally.

Introduction:

l2 – Individual recruitment success – The manuscript starts with 3 words that sound enigmatic (who is recruiting who and what for?). Can the authors use alternative wording or define right away in what sense they use these words (the same remark applies to abstract)?

l28 – The author qualified the winning genotypes as Sysphean but only later explain why (>l33). The authors used that wording of (a) use a novel term and then (b) only later define it. This does not ease the reading. I suggest doing it the other way around every time it is possible.

l73: "the probability does not depend on N". What probability? Fixation or appearance of beneficial mutation?

Results:

– If all genes are pulled together as if it was a single very long gene, what would be the NI, the overall Tajima's D, and related neutrality tests? In brief, what is the mean pattern?

– What ABC priors did the authors use?

– The fit using \Λ-Β is amazing. Why did the authors keep it in the Supp Mat?

– The discussion about the impact of parameters is sometimes obscure without a deep look at the Supp Mat. Generally, I would recommend the authors keep in the main only aspects that could be understood without looking at the Sup Mat. If they can produce an easy-to-read version in the main, completed with a more thorough and detailed analysis in the Supp Mat, they would have achieved a big step in making their claim very strong.

Discussion:

– Yes indeed, it is tempting to discuss the cost of natural selection here (Haldane's and followers up to Kimura). Can the authors estimate the number of ongoing sweeps and thus estimate the "cost of selection" to see if it can be supported by the codfish population? That'd be an interesting quantity to compute, but it may well be for future work.

General:

– Generally, I would recommend using "processes" or "mechanisms" rather than "forces" for evolution (see the work of Denis M Walsh on the topic, article "Not a sure thing", "Four pillars of statisticalism" among others). This is only for your own culture, not at all required for the publication at stake. You may well disagree with the arguments.

Overall:

In the end, although my comments could be interpreted as annoying, I want to again restate my genuine support for this extremely careful, sound, and interesting work. I really think it is worth making an extra effort so that most non-expert readers of *eLife* will understand what is going on. Asking for help from colleagues outside of the field can be an option. I leave the authors to choose what is best for them.

*Reviewer #2 (Recommendations for the authors):*

Some thoughts of what I would find interesting to discuss in connection to my concern listed in the Public Review that I do see issues with arguing that high fecundity plays a role in shaping diversity when you find a quite good fit of the DS model:

If high fecundity is added to the DS model, should that affect the time scaling (and if so, can this be estimated or at least guesstimated)? Or would it even change the coalescent limit? If adding high fecundity to the model does not change anything, do the points you mention in the discussion paragraph have a measurable effect on the population?

Here, I do not expect a full treatment of these questions, but an outlook and discussion of these points. If possible, some rough approximative calculations resp. some mathematical heuristics would be highly appreciated (at least by me).

*Reviewer #3 (Recommendations for the authors):*

I really enjoyed this paper and I congratulate the authors for their meticulous work in examining alternative scenarios to the selective sweepstakes hypothesis. My main criticisms relate to four points that would need to be clarified in order to strengthen or mitigate some of the conclusions. The first two concern the interpretation of results from models that must simplify complex biological characteristics of mating systems, as well as the dynamics of selection. The last two points deal with the possible impact of other factors that could contribute to the overall V-shape of the cod SFS, including introgression and SNP misorientation.

1. Possible limitations of sweepstakes models with respect to the characteristics of the Atlantic cod mating system.

I was wondering how the Χi-Β(2-α,α) coalescent model assumption that a single pair of parents produces a large number of juvenile offspring could be challenged by a somewhat lower-variance lower-skewness offspring distribution. This might happen for instance because batch-spawning of females could imply pairing with multiple males across batches in the absence of mate fidelity. Also, although the spawning behaviour of the Atlantic cod involves a pairing between two partners to form a ventral mount, fertilization by satellite males has been reported within a single spawning batch. These features of the mating system may reduce the chance that a single pair of parents will occasionally produce a huge number of reproducing offspring, at least for the male parents.

Concerning the DS model, the assumption that only one female hits the jackpot may seem somewhat at odds with the fact that selective sweeps occur mainly at mutations that have already reached a certain frequency (L220-223). At the beginning of the exponential phase of the sweep, several generations will have elapsed since the appearance of the advantageous mutation and several descendant females will carry it. So in other words the sweepstakes effect applies more to the haplotypes that sit on the advantageous mutation than to the genome of the individual in which it initially appeared. Perhaps this distinction should be clarified somewhere (i.e. random sweepstakes increasing the transmission of a winner's entire genome as opposed to selective sweepstakes mainly affecting genomic regions around selected loci).

I also wondered whether a certain amount of random sweepstakes might help new advantageous mutations to escape loss by drift and reach the exponential phase more quickly. Such a mixture of random and selective sweepstakes could be considered as a more realistic scenario, although a mixed model taking them into account simultaneously seems difficult to implement.

For this first set of comments, I do not expect methodological or analytical improvements, but rather clarifications in the text if necessary.

2. Biological meaning of the compound parameter:

The Durrett-Schweinsberg model considers recurrent hard sweeps where new beneficial mutations are driven to fixation by selection. The compound parameter (c = δs²/ϒ) of the DS model integrates three types of information, the rate of sweeps (δ), the squared selection coefficient (s²), and the recombination rate (ϒ). Therefore, the compound parameter can be understood as a selection density parameter (somewhat similar to Aeschbacher et al. 2017, who considered the selection coefficient per base pair) per generation per genetic map unit.

I wondered to what extent this compound parameter could capture the effect of partial or incomplete sweeps in addition to hard sweeps. In essence, a large number of incomplete sweeps could potentially leave a similar footprint as a smaller number of hard sweeps.

Do the authors think that the DS model is more sensitive to recurrent hard sweeps than to even more pervasive incomplete sweeps (or even soft seeps), and why would this be the case? The potential difficulty of interpreting the compound parameter in terms of biologically relevant quantities should lead to caution when estimating the rate of hard sweeps per chromosome from the cod data.

This is notwithstanding that quantitative estimation of the sweepstakes rate would require, as recognized by the authors (e.g. L579), the development of a diploid genomic version of the DS model. My feeling is that at this stage, the main contribution of the proposed approach lies more in the comparison and selection of models for hypothesis testing (i.e. selective versus random sweepstakes) than in the estimation of parameter values. That said, the estimated parameter value under the best-fit DS model can be prudently discussed for its implications in biological terms. In particular, the estimation of c for different functional regions of the genome seems biologically meaningful since it could indicate a higher selection density per generation per genetic map unit in exons and UTRs. For comparison, it would be interesting to estimate the genome-wide average proportion of adaptive non-synonymous substitutions α using 1-NI, since estimates of adaptation rate derived from the McDonald-Kreitman test should not be affected by the type of selective sweeps. The value of α for Atlantic cod could also be compared with that of other species, such as *Drosophila*, known to have frequent sweeps.

Reference:

Aeschbacher, S., Selby, J. P., Willis, J. H., and Coop, G. (2017). Population-genomic inference of the strength and timing of selection against gene flow. Proceedings of the National Academy of Sciences, 114(27), 7061-7066.

3. Impact of admixture and introgressed ancestry

The impact of the demographic history on the shape of the site frequency spectrum was addressed by considering historical changes in effective population size (expansions or contractions) and cryptic breeding structure.

The effect of cryptic population structure was assessed under both the Kingman and Xi-Β-coalescent by considering a two-island model with varying migration rates, and different mixtures of samples from the two differentiated populations. Under both coalescent models, only a minor sample size of one (i.e. rare minor ancestry) combined with the highest rate of migration (i.e. low genetic differentiation) gives the closest resemblance to the V-shaped observed SFS (Suppl. Figure 15-16). These results were used to reject the effect of potential cryptic population structure. However, by considering mixtures of samples from the two differentiated populations without interbreeding (i.e. no admixture), the approach does not explore the potential effect of introgressed ancestry following admixture between two differentiated populations. The leave-one-out approach presented in Suppl. Figure 19 is only suitable for testing the population mixture hypothesis, but it cannot detect signals of recombined ancestries that happen following admixture (because introgressed ancestry would be present across the entire genome).

Results from previous population genomics studies (including Árnason and Halldórsdóttir 2019) have shown that the Atlantic cod is admixed at least with the polar cod and that there is also introgression among genetically differentiated populations within Atlantic cod. Therefore, introgression of foreign ancestry seems a biologically realistic scenario that could generate a V-shaped SFS if introgressed ancestry is present at a low frequency across the genome.

Interestingly, the Manhattan plots of neutrality tests presented in Figure 1 a and b show an excess of low-frequency derived alleles genome-wide, except near chromosome extremities where high-frequency derived alleles are often in excess. Selection against minor introgressed ancestry can lead to this kind of pattern because segments of introgressed ancestry are less efficiently removed by selection in highly recombining regions such as chromosome extremities (e.g. Schumer et al. 2018).

An easy way to consider the effect of (neutral) introgressed ancestry on the shape of the SFS would be to allow migrants to interbreed with the recipient population for e.g. 100 generations in the low migration scenarios.

Reference:

Schumer, M., Xu, C., Powell, D. L., Durvasula, A., Skov, L., Holland, C., … and Przeworski, M. (2018). Natural selection interacts with recombination to shape the evolution of hybrid genomes. Science, 360(6389), 656-660.

4. Impact of variant misorientation on the SFS

Four different species of cod taken from Árnason and Halldórsdóttir 2019 were used as outgroups to perform variant polarization in the Atlantic cod. As I understand it, each species was taken separately as an outgroup, thus generating four possible tables of variant orientation and therefore 4 possible SFS.

Observed SFS of the non-inversion chromosomes that were polarized using three different outgroups (Arctic cod Bsa, walleye pollock Gch, and Pacific cod Gma) are presented in Suppl. Figure 11, and show slightly more high-frequency derived variants using Bsa, compared to Gma and even more compared to Gch. Of these three species, the Arctic cod Bsa is the most divergent from the Atlantic cod, and therefore seems to be the best available outgroup to perform variant orientation while minimizing orientation errors due to shared variation. However, the Pacific cod Gma was the chosen outgroup for most analyses. Why did you choose this species and is there a risk of shared ancestry due to ancient admixture between the Atlantic and the Pacific cod that could affect SNP orientation?

I wondered if the somewhat symmetric distribution of the residuals, which is W-shaped with a more pronounced departure on the right tail of the SFS, could indicate a problem of SNP misorientation. Indeed, observed data show an excess of high-frequency derived alleles compared to the best-fit DS model, which might be the consequence of SNP misorientation.

One possibility to test this hypothesis could be to include a parameter for the rate of variant misorientation when fitting the coalescent models and estimate its value as for other parameters. This is classically done in historical demographic inference based on the SFS.

Alternatively, you could more easily provide a summary of ancestral state consistency, or maximum parsimony across the cod species tree, using the results of variant polarization independently performed with the four different outgroups. If this comparison reveals a very low frequency of non-parsimonious allelic states across the species tree, then variant misorientation is unlikely to be a problem.

---

## [Author Response]

Essential revisions:The three reviewers were generally enthusiastic. They agreed that the study could have a strong impact in the field of population genetics and a potential for application to a broad range of taxa beyond marine fishes. While the overall assessments were positive, the reviewers also made a series of complementary comments to help you to better support your claims, or instead moderate some of your conclusions in the discussion. While the paper cannot be published as it stands, addressing these comments should be manageable without the need for extensive additional analysis.Below is a list of essential revisions that should help your to alter the claims or make clear that some conclusions will need supporting data in the future.1) A priority task concerns the haploid Λ-Β coalescence model of random sweepstakes (reviewers #1 and #2). It is not clear from the manuscript whether the excellent fit of the Λ-Β-Coalescent model is actually better than the fit of the DS model. The authors should test or explain which model fits better.If it is the Λ-Β-Coalescent model, they should explain why the Λ-Β fit is so good and why the DS model was chosen despite a worse fit (e.g. if other parameters such as LD disagree with Λ-Β) – or restate their main result. Alternatively, if the DS model fits best among all coalescent models, the authors should not argue that the better fit of Λ-Β over Xi-Β is due to selection akin to selection in the DS model (see reviewer's comment #2 on this point).

The short answer is that the Λ-Β model does not fit as well as the DS model. The deviations of observations from the MLE of the Λ-Β model are larger than the deviations from the DS model. We now have added plots of deviations from the Λ-Β model to the Supplement. We also still keep this in the Supplement.

R#2 also asks what the Λ-Β model adds and mentions that the Bolthausen-Sznitman coalescent (α = 1) can be interpreted in terms of selection. Our comparison of observations with expectations of the Λ-Β model for various values of α in Supplementary Figure 4 (comparisons to α values that are representative of the posterior estimates for the Xi-Β coalescent in Supplementary Figure 3) show that low frequency alleles (singletons and doubletons) are close to expectations for α = 1.35 and intermediate-frequency alleles close to α = 1.0. Hence our interest in the Bolthausen-Sznitman coalescent as a possible indication of selection. However, this piecewise comparison is not a proper statistical goodness-of-fit and the maximum likelilhood estimate does not support α = 1 of the Bolthausen-Sznitman coalescent. The parameter α determines the skewness of the offspring distribution in the Λ-Β-coalescent and could be construed as a parameter of selection causing that skewness. But that model has no known interpretation for an explicitly selection-driven skewness (except in the special case α = 1, the Bolthausen-Sznitman coalescent). Therefore, we prefer the Durrett-Schweinsberg model (which also is a Λ coalescent) because it both fits better and also because it explicitly incorporates selection and is interpretable in terms of selection. We now clarify this with changes to the section on this matter.

2) Some of the main findings also need further verification to be more robust. The additional checks and analyses requested include the following:– Provide a genome-wide estimate of NI, or alternatively alfa, the proportion of adaptive non-synonymous substitutions (Reviewers #1 and #3).

We now put a vertical line at the mean −log*NI* in Figure 1 and in the caption and text give the mean and the median of −log*NI* (0.27 and 0.21 respectively) which imply that the proportion of adaptive non-synonymous substitutions is 19–24%.

– Evaluate potential issues related to SNP misorientation (Reviewers #1 and #3).– Assess the impact of low-frequency introgressed ancestry on the shape of the SFS (Reviewer #3).

For the purposes of this paper we consider SNP misorientation and low-frequency introgressed ancestry to cause the same kinds of problems in the data except that introgression is more genomewide. We deal with these issues together. A parallel mutation in the outgroup to the same state as a derived biallelic polymorphic state in the ingroup will flip orientation of a site with the ancestral state being considered derived. Singleton and doubleton sites are the most common sites and such sites would flip to the *n* − 1 and *n* − 2 class respectively thus increasing the right tail of the SFS. Under low-level ancestral introgression sites that were formerly monomorphic in the ingroup would flip at sites that have a derived state in the outgroup. Introgression would not affect the right tail of the SFS at polymorphic sites where the outgroup carries the ancestral allele. Instead such sites would have a higher frequency of the ancestral allele pushing derived alleles in the ingroup to a lower frequency pulling up the left side of the SFS. Ancestral introgression will affect ingroup sites that were fixed for the derived allele before introgression by making such sites polymorphic and contributing to the right tail of the SFS. We agree that this is an important issue that we address with the following.

The tri-allelic filtering of sites (-skipTriallelic 1; line 666) that we applied goes some way towards addressing these issues in conjunction with the multiple outgroup taxa. This will leave only sites that have the same exact derived state in the outgroup as one of the two alleles in the ingroup. If a particular site can be polarised by outgroup A (for example Gma) it means that the state of the site in taxon A is the same as one of the alleles segregating in the ingroup population. If outgroup B (say Gch) has a different state for that site, the site would be tri-allelic in that comparison and removed by the tri-allelic filtering. We did not use parsimony or consensus to infer the state of ancestral nodes (c.f. Keightley and Jackson, 2018). Therefore, this filtering will not remove sites which have parallel changes simultaneously in two or three outgroup taxa. We now add statements in the methods to clarify this filtering.

An important point on filtering about potential hidden population structure inadvertently got left out of the manuscript. We now add a statement in the methods about this filtering:

“We have sequenced a large sample of cod from various localities in the North Atlantic and used both PCA and admixture analysis using PCangsd (Meisner and Albrechtsen, 2018) and found some population substructure and possible admixture (unpublished results). To minimize the effects of potential population substructure we screened our samples in this study and ensured that they are members of the same cluster detected by PCA and assigned to the same population detected with admixture. This filtering also addresses the issue of potential SNP misorientation and ancestral admixture.”

To further address the reviewers concerns we did the following:

First, we reasoned that SNP misorientation and low-level ancestral introgression will mostly affect singletons and doubletons as well as the anti-singletons (*n* − 1) and anti-doubletons (*n* − 2) classes. We now include a Figure 2—figure supplement comparing the truncated SFS to full SFS and to respective expectations of the DS model. In short removing these classes does not change the picture. We now add a statement in the methods and include a Figure 2—figure supplement.

Second, we reasoned that sites with transition variation are more likely than transversions to be saturated and thus parallel changes are more likely at those sites leading to SNP misorientation. To address this we removed transistions (-rmTrans flag) and fitted SFS of transversions only. We now add a statement in the methods and include a Figure 2—figure supplement.

Third, we studied the effect of outgroups. Instead of what R#3 mentioned about using maximum parsimony across the cod species tree we decided to use as an outgroup a 100% consensus sequence of walleye pollock (Gch), Pacific cod (Gma), and Arctic cod (Bsa). Thus, only sites at which the three outgroup taxa agree are used. We think this is a strong argument, stronger than parsimony. Three out of three is stronger than two out of three under the parsimony argument. We now add a statement in the methods and include a Figure 2—figure supplement.

It is worth noting that the very right tail of the SFS is lower and more in line with the DS model for both the transversions and the 100% consensus data. This may mean that SNP misorientation had some effect in the original full analysis. However, the effect does not change the results qualitatively. We now make a statement to this effect in the Discussion.

3) Several comments relate to points that do not require further analysis but rather a clarification in the discussion, including:– The need to reconcile the role of high fertility in shaping diversity with the assumptions of the DS model, and the related issue of potential effects on the time scaling (Reviewers #2).

Regarding the Moran model assumption of the DS model and how to integrate high fecundity we add that: In the Moran (1958) model of reproduction a single individual replicates and another dies instead. The reproducing individual remains active in the population. In the Durrett-Schweinsberg addition, the reproducing individual is fitter. On a coalescent time scale, there is a burst of reproductive activity by the individual and his descendants such that their fit, derived lineage quickly takes over the whole population. The Moran model does not model high fecundity by itself. Still, with this addition, it is as if the Moran process and its associated Kingman coalescent are interrupted by a burst of high fecundity. Thus, the Durrett-Schweinsberg model approximates selective sweepstakes by adding recurrent sweeps of a new selectively advantageous mutation each time to the Moran model.

We, furthermore, made changes and now state: The Durrett-Schweinsberg model is haploid and the resulting coalescent is a Λ-coalescent. The Λ-Β(2 − α,α) coalescent for small values of α approximates the Durrett-Schweinsberg coalescent and the fact that the Λ-Β(2 − α,α) fits better to our data than Ξ-Β(2 − α,α) coalescent is further indication for selection. The fact that a haploid Λ-coalescent model fits a diploid organism better than the corresponding diploid Ξ-coalescent is suggestive of natural selection, where fitter offspring descend from one particular parental chromosome out of the available four. The Durrett-Schweinsberg model is a haploid Λ-coalescent model incorporating such selection explicitly. It results in a model fit which is slightly better to the Λ-Β-coalescent (as seen by comparison of residuals), and we regard it as the preferred model because its parameter has an explicit interpretation as a selection coefficient. The parameter α determines the skewness of the offspring distribution in the Λ-Β-coalescent. But that model has no known interpretation for an explicitly selection-driven skewness (except in the special case α = 1, the Bolthausen-Sznitman coalescent, which does not adequately fit our data). Hence, the Λ-Β-coalescentis not an appropriate model for a diploid organism, even under haploid-seeming selection.

– The cost of selection, and the biological meaning of the compound parameter with respect to the nature of selective sweeps (Reviewers #1 and #3).

We now ask Does cod have reproductive capacity to substitute adaptive alleles at a high rate without going extinct from the substitution load (Kimura and Maruyama, 1969)? If each high-fitness fish has *k* offspring which survive long enough to reproduce, and the selective sweep starts from a single individual, then after one generation there are *k* fit fish, after 2 there are k2, after 3 there are k3, and so on. After log*N* generations at the completion of the sweep there are klog⁡N fit descendants. For the sweep to complete, we thus need klog⁡N=N, or k=N1/log⁡N=e*k* = *N*(1∕log*N*) = *e* the base of natural logarithm. As a numerical example it is immaterial whether we assume a a population size of a billion (N=109) and duration of a sweep 20.7 generations or a population size of a trillion (N=1012) fish and duration of a sweep 27.6 generations. The reproductive excess required is 2.71 or approximately three fit offspring that make it to reproduction. In practice the number will have to be higher because not all fit offspring will survive to reproduce, and because our estimated frequency of sweeps was high enough that 20 to 30 generations might be a bit too long. However, there is no reason to think that the cod population would not have the reproductive capacity to support selective substitution at the estimated rate. We add a paragraph to this effect.

The compound parameter c=δs2/γ is essentially the density of selection per map unit along a chromosome (Aeschbacher et al., 2017). To investigate this we note that the number of single nucleotide polymorphisms (SNP) in a chromosomal fragment (e.g. 25k fragment) is proportional to branch length, which is proportional to the compound parameter *c*. It is, therefore, legitimate to regard these as factors by which to multiply the single *c* from each window. The results are presented in a supplemental Figure and briefly in the discussion.

– Possible limitations of sweepstakes models with respect to characteristics of the Atlantic cod mating system (Reviewer #3).

We have added a paragraph in that context of cod mating drawing attention to a model of Birkner et al. 2018 on a two-sex model that may more accurately reflect the mating system of cod. Further development of such models must be left for the future.

4) Finally, there are also requests for revision on the grounds of clarity, conciseness, and presentation. The details are provided in the reviews (mostly Reviewer #1), so I only mention the most important points that would require:

We follow the reviewer comments and clarify the presentation.

– Making the content more accessible to non-experts readers, keeping in the main text only aspects that could be understood without looking at the Sup Mat, and completing with a more thorough and detailed analysis in the Supp Mat when necessary.

We have tried to simplify the text and make it more accessible to non-experts.

– Clarifying definitions and simplifying figure legends, selecting only one representative panel for some figures, and making references to supplementary material easier to follow.

We now reduce the number of panels in the figures and add other panels as Figure supplements using the *eLife* template. We have simplified the captions and clarified references to the supplementary material.

– Reducing the last section of the results that considers alternative explanations that do not fit the data.

We have moved the whole section on alternative explanations to the Supplement. We give a brief account of those results in the Results section and make a statement about those results in the Discussion along the lines suggested by R#3.

Reviewer #1 (Recommendations for the authors):My major comment does not concern the science aspect of this manuscript but encourages the authors to revise in order to ease the reading. I found it sometimes difficult due to numerous references to supplementary material all along the text, that figure legends can be clarified, and sometimes the text lacks a clear definition of words. Furthermore, I found the article quite lengthy in the last section (all the alternative explanations that do not fit the data). I believe that an improvement is possible in making the content more accessible to non-experts so that a wide range of eLife readers will be motivated in getting through this article.

We have moved the section on alternative explanations to the Supplementary material. We have made small changes to the text there and added a subsection dealing with the issue of SNP misorientation and ancestral introgression. We have also added a short account of the alternative explanations to the Results and Discussion sections that we hope are accessible to non-experts. We have tried to follow your advice to make the text more accessible to non-experts throughout. We have tried to remove jargon, improved the English, and tried to soften the tone.

Second, did the authors consider a potential issue with misorientation of variants, where some derived are considered ancestral and vice-versa? This has been put forward as a plausible explanation to explain the U-shaped SFS. Although I suspect the fit in the case studied here is so good with \Λ-MMC or S.D.-MMC, that orientation errors are not a real issue. It may however be nice to mention/discuss it somewhere in the text.

We now add further analysis showing that SNP misorientation may have had some effects, but not so much that it changes the basic results.

Figures: There are often many almost identical figures among the panels. Please consider selecting only one and mention that the other ones are similar. It will also simplify (and clarify) the figure legends.

We now use the *eLife* template and make the figures simpler by selecting a single or few panels and with the additional figures as figure supplements. We think this addresses the concerns of the reviewer.

Figure 3 is quite ugly. Can the authors display some kind of density plot with iso-density lines?

Ugly figure replaced with a plot with kernel-density estimated iso-density contours (25, 50, 75, 90, 95, and 99% contours). A single panel was selected and other panels are presented as a figure supplement to the main figure.

In the following, I list a few suggestions that could help clarify the text to broaden, as much as possible, the audience. But this can be done more generally.Introduction:l2 – Individual recruitment success – The manuscript starts with 3 words that sound enigmatic (who is recruiting who and what for?). Can the authors use alternative wording or define right away in what sense they use these words (the same remark applies to abstract)?

We now clarify and refer to reproductive success instead of recruitment.

l28 – The author qualified the winning genotypes as Sysphean but only later explain why (>l33). The authors used that wording of (a) use a novel term and then (b) only later define it. This does not ease the reading. I suggest doing it the other way around every time it is possible.

Point taken and rephrased

l73: "the probability does not depend on N". What probability? Fixation or appearance of beneficial mutation?

It is probability of fixation. Corrected.

Results:– If all genes are pulled together as if it was a single very long gene, what would be the NI, the overall Tajima's D, and related neutrality tests? In brief, what is the mean pattern?

The overall Tajima’s D, and related neutrality tests are given in SM Table 1. We now put a vertical line at the mean −log*NI* in Figure 1 and in the caption and text give the mean and the median of −log*NI* (0.27 and 0.21 respectively) which imply that the proportion of adaptive non-synonymous substitutions α = 1 − *NI* is 19–24%. Comparing this to organims like *Drosophila* it should noted that we have a more circumscribed population defined to minimize the potential effects of cryptic population structure. When we concatenate all genes of each chromosome, the chromosomal α values range from −0.16 to 0.07 and for the concatenated whole genome α = −0.017. In this respect it is important to remember that singletons are over 50% of all sites and doubletons are another 12% of sites. It is possible and even likely that some of the non-synonymous singleton/doubletons are slightly deleterious inflating the *Pn*∕*Ps* ratio (*Pn*∕*Ps* = 0.64 for the concatenated genome). However, we think that dealing with issues such as bias of *NI* estimators and violations of polymorphic neutrality are beyond the scope of the present study. We therefore present only the α based on the mean value above and leave a full analysis of *NI* and α=1−NI for later.

– What ABC priors did the authors use?

In general we used uniform priors. We now add a Supplementary Table listing the priors for different analysis.

– The fit using \Λ-Β is amazing. Why did the authors keep it in the Supp Mat?

It is indeed a good fit but not as good as the DS to judge from the residuals. As we discuss above the Λ-Β does not have a straightforward interpretation in terms of selection (except for α = 1) and we keep this model in the Supplementary material. We prefer the DS model because it explicitly models selection and recombination.

– The discussion about the impact of parameters is sometimes obscure without a deep look at the Supp Mat. Generally, I would recommend the authors keep in the main only aspects that could be understood without looking at the Sup Mat. If they can produce an easy-to-read version in the main, completed with a more thorough and detailed analysis in the Supp Mat, they would have achieved a big step in making their claim very strong.

We have moved the section on “Other processes” to the Supplement and revised the main text in an attempt to get a more easy-to-read text.

Discussion:– Yes indeed, it is tempting to discuss the cost of natural selection here (Haldane's and followers up to Kimura). Can the authors estimate the number of ongoing sweeps and thus estimate the "cost of selection" to see if it can be supported by the codfish population? That'd be an interesting quantity to compute, but it may well be for future work.

We added a simple but robust argument about the number of fit offspring that an individual must be able to produce to withstand the load (see above).

General:– Generally, I would recommend using "processes" or "mechanisms" rather than "forces" for evolution (see the work of Denis M Walsh on the topic, article "Not a sure thing", "Four pillars of statisticalism" among others). This is only for your own culture, not at all required for the publication at stake. You may well disagree with the arguments.

We have changed forces to processes or mechanisms

Overall:In the end, although my comments could be interpreted as annoying, I want to again restate my genuine support for this extremely careful, sound, and interesting work. I really think it is worth making an extra effort so that most non-expert readers of eLife will understand what is going on. Asking for help from colleagues outside of the field can be an option. I leave the authors to choose what is best for them.

Thank you for your comments. We appreciate your comments and advice.

Reviewer #2 (Recommendations for the authors):Some thoughts of what I would find interesting to discuss in connection to my concern listed in the Public Review that I do see issues with arguing that high fecundity plays a role in shaping diversity when you find a quite good fit of the DS model:

We now add a clarification. In the Moran model a female chosen to reproduce stays around. As stated above one can look at the Moran process as being interrupted by high fecundity. We hope that this answers the concerns.

If high fecundity is added to the DS model, should that affect the time scaling (and if so, can this be estimated or at least guesstimated)? Or would it even change the coalescent limit? If adding high fecundity to the model does not change anything, do the points you mention in the discussion paragraph have a measurable effect on the population?Here, I do not expect a full treatment of these questions, but an outlook and discussion of these points. If possible, some rough approximative calculations resp. some mathematical heuristics would be highly appreciated (at least by me).

This is an important issue for future direction and we add the following to the Discussion:

"Our work motivates the construction of joint models featuring neutral and selective sweepstakes. As a starting point, we expect selective sweeps akin to the Durrett–Schweinsberg model could be incorporated into the (Schweinsberg, 2003) pre-limiting population models giving rise to the Beta coalescent. To affect the infinite-population limit, selective sweeps would have to occur on the same fast timescale of Nα−1 generations as neutral multiple mergers. Even on this timescale, selective sweeps lasting log*N* generations will be instantaneous and result in multiple mergers in the coalescent limit. An intriguing possibility is that the Durrett-Schweinsberg selective sweeps could account for some of the observed deviation from the Kingman coalescent, the combined model, might yield substantially higher best-fit estimates of α than those obtained from the more restrictive Β-coalescent. Low values of α result in implausibly short timescales for evolution, and a combined neutral-and-selective sweepstakes model has the potential to avoid this defect."

Reviewer #3 (Recommendations for the authors):I really enjoyed this paper and I congratulate the authors for their meticulous work in examining alternative scenarios to the selective sweepstakes hypothesis. My main criticisms relate to four points that would need to be clarified in order to strengthen or mitigate some of the conclusions. The first two concern the interpretation of results from models that must simplify complex biological characteristics of mating systems, as well as the dynamics of selection. The last two points deal with the possible impact of other factors that could contribute to the overall V-shape of the cod SFS, including introgression and SNP misorientation.1. Possible limitations of sweepstakes models with respect to the characteristics of the Atlantic cod mating system.I was wondering how the Χi-Β(2-α,α) coalescent model assumption that a single pair of parents produces a large number of juvenile offspring could be challenged by a somewhat lower-variance lower-skewness offspring distribution. This might happen for instance because batch-spawning of females could imply pairing with multiple males across batches in the absence of mate fidelity. Also, although the spawning behaviour of the Atlantic cod involves a pairing between two partners to form a ventral mount, fertilization by satellite males has been reported within a single spawning batch. These features of the mating system may reduce the chance that a single pair of parents will occasionally produce a huge number of reproducing offspring, at least for the male parents.

These are interesting questions that we do not have good answers to at this time. In a monogamous big-bang reproduction eggs might fail to get ferilized (be carried away by currents before sperm hitting them, for example) which would be less likely in a promiscous big-bang swarm reproduction (Nereis like). In a batch spawner like cod with pairing with a particular male at a time and with sneaking males (and being repeated with a different male and sneakers in the next batch), might actually have a higher fertilization success. Ferilization of course is important being the first filtering step to adult offspring. The Xi-Β(2 − α,α) models the simultaneous multiple-mergers of the four parental chromosomes of each sampled individual, In a batch spawner like cod the two maternal chromosomes will get associated with more than two paternal chromsomes. This might mean more frequent simultaneous multiple-mergers. However, there is nothing in our data that allows us to differentiate between these scenarios. We add a sentence drawing attention to how mating system might influence multiple-mergers.

Concerning the DS model, the assumption that only one female hits the jackpot may seem somewhat at odds with the fact that selective sweeps occur mainly at mutations that have already reached a certain frequency (L220-223). At the beginning of the exponential phase of the sweep, several generations will have elapsed since the appearance of the advantageous mutation and several descendant females will carry it. So in other words the sweepstakes effect applies more to the haplotypes that sit on the advantageous mutation than to the genome of the individual in which it initially appeared. Perhaps this distinction should be clarified somewhere (i.e. random sweepstakes increasing the transmission of a winner's entire genome as opposed to selective sweepstakes mainly affecting genomic regions around selected loci).

Yes, this is a good point. We now add a sentence to clarify this that selective sweeps are more locus specific and affect variation within a recombinational unit whereas random sweepstakes is more genome-wide.

I also wondered whether a certain amount of random sweepstakes might help new advantageous mutations to escape loss by drift and reach the exponential phase more quickly. Such a mixture of random and selective sweepstakes could be considered as a more realistic scenario, although a mixed model taking them into account simultaneously seems difficult to implement.

One problem with this is that the allele frequency variance is larger with sweepstakes reproduction so variation also gets lost more frequently. But combined with selection variants that by chance are brought to a high enough frequency to contribute to the variance in fitness can be pulled into the selective mode. It is worth noting in this respect that singletons and doubletons are captured by both the Xi-Β and the Λ-Β models (Supplementary Figures 3 and 4, with α ≈ 1.35) while they are not at all captured by the Kingman coalescent (Figure 2 for example). So one could argue that incoming mutations are behaving more like variation under random sweepstakes than under the classical case and that random sweepstakes may provide raw material for selection to act on. So although it is true (as we already said) that a larger variance means that some new variants are lost other variants are brought to a higher frequency where they contribute to the variance in fitness. We now add a sentence to this effect.

For this first set of comments, I do not expect methodological or analytical improvements, but rather clarifications in the text if necessary.2. Biological meaning of the compound parameter:The Durrett-Schweinsberg model considers recurrent hard sweeps where new beneficial mutations are driven to fixation by selection. The compound parameter (c = δs²/ϒ) of the DS model integrates three types of information, the rate of sweeps (δ), the squared selection coefficient (s²), and the recombination rate (ϒ). Therefore, the compound parameter can be understood as a selection density parameter (somewhat similar to Aeschbacher et al. 2017, who considered the selection coefficient per base pair) per generation per genetic map unit.

Yes, the compound parameter c=δs2/γ is like a density of selection along the chromosome.

And taking the (numerical) derivative of *c* with respect to recombination unit would yield an effective local selection coefficient. However, estimating this in windows along the chromosome is too noisy. Since the number of SNPs is proportional to branch length, which is proportional to *c*, it should be legitimate to regard these as factors by which to multiply the single *c* we already have in each window. We now add a supplementary figure showing this for chromosome 4 and discuss this and give a reference to Aeschbacher et al. in the discussion.

I wondered to what extent this compound parameter could capture the effect of partial or incomplete sweeps in addition to hard sweeps. In essence, a large number of incomplete sweeps could potentially leave a similar footprint as a smaller number of hard sweeps.

The DS is a model of complete hard sweeps. It is of course possible, perhaps likely, that both soft and incomplete sweeps/recombinational breakup occur in cod. Our estimator would pick up effects of such sweeps. Incomplete sweeps and LD-based measures may be important particularly as a future look in connection with the tweaks to the DS model to study LD such as the one that we present in Supplemental Figure 6. We now add statements to clarify this and add a reference to LRH and EHH and Sabeti et al. 2006 and 2007.

Do the authors think that the DS model is more sensitive to recurrent hard sweeps than to even more pervasive incomplete sweeps (or even soft seeps), and why would this be the case? The potential difficulty of interpreting the compound parameter in terms of biologically relevant quantities should lead to caution when estimating the rate of hard sweeps per chromosome from the cod data.

As we already said the DS is a model of complete hard sweeps but the estimator may detect soft and incomplete sweeps.

This is notwithstanding that quantitative estimation of the sweepstakes rate would require, as recognized by the authors (e.g. L579), the development of a diploid genomic version of the DS model. My feeling is that at this stage, the main contribution of the proposed approach lies more in the comparison and selection of models for hypothesis testing (i.e. selective versus random sweepstakes) than in the estimation of parameter values. That said, the estimated parameter value under the best-fit DS model can be prudently discussed for its implications in biological terms. In particular, the estimation of c for different functional regions of the genome seems biologically meaningful since it could indicate a higher selection density per generation per genetic map unit in exons and UTRs. For comparison, it would be interesting to estimate the genome-wide average proportion of adaptive non-synonymous substitutions α using 1-NI, since estimates of adaptation rate derived from the McDonald-Kreitman test should not be affected by the type of selective sweeps. The value of α for Atlantic cod could also be compared with that of other species, such as *Drosophila*, known to have frequent sweeps.

It is possible that there is a higher selection density per generation per genetic map unit in exons and UTRs as the reviewer points out. But recombination also plays a role and we do not have a good handle of differences in recombination in the different functional units. Based on the mean *NI* our estimated proportion of adaptive non-synonymous substitutions α is 19-24%. We already had a reference to Li et al. 2008 comparing our estimated −log*NI* in Figure 1 to *Drosophila* as well as humans and yeast. Our results for the cod are more similar to *Drosophil*a than to other taxa. We also point out that our study is of a circumscribed population from a single locality whereas the *Drosophila* and human studies cast their net wider. We predict that a sample of cod from throughout its distributional range would show more positive selection. We now add statements to this effect and add references to *Drosophila* work (Bierne and Eyre-Walker2004 and Sella et al. 2009).

Reference:Aeschbacher, S., Selby, J. P., Willis, J. H., and Coop, G. (2017). Population-genomic inference of the strength and timing of selection against gene flow. Proceedings of the National Academy of Sciences, 114(27), 7061-7066.3. Impact of admixture and introgressed ancestryThe impact of the demographic history on the shape of the site frequency spectrum was addressed by considering historical changes in effective population size (expansions or contractions) and cryptic breeding structure.The effect of cryptic population structure was assessed under both the Kingman and Xi-Β-coalescent by considering a two-island model with varying migration rates, and different mixtures of samples from the two differentiated populations. Under both coalescent models, only a minor sample size of one (i.e. rare minor ancestry) combined with the highest rate of migration (i.e. low genetic differentiation) gives the closest resemblance to the V-shaped observed SFS (Suppl. Figure 15-16). These results were used to reject the effect of potential cryptic population structure. However, by considering mixtures of samples from the two differentiated populations without interbreeding (i.e. no admixture), the approach does not explore the potential effect of introgressed ancestry following admixture between two differentiated populations. The leave-one-out approach presented in Suppl. Figure 19 is only suitable for testing the population mixture hypothesis, but it cannot detect signals of recombined ancestries that happen following admixture (because introgressed ancestry would be present across the entire genome).Results from previous population genomics studies (including Árnason and Halldórsdóttir 2019) have shown that the Atlantic cod is admixed at least with the polar cod and that there is also introgression among genetically differentiated populations within Atlantic cod. Therefore, introgression of foreign ancestry seems a biologically realistic scenario that could generate a V-shaped SFS if introgressed ancestry is present at a low frequency across the genome.Interestingly, the Manhattan plots of neutrality tests presented in Figure 1 a and b show an excess of low-frequency derived alleles genome-wide, except near chromosome extremities where high-frequency derived alleles are often in excess. Selection against minor introgressed ancestry can lead to this kind of pattern because segments of introgressed ancestry are less efficiently removed by selection in highly recombining regions such as chromosome extremities (e.g. Schumer et al. 2018).An easy way to consider the effect of (neutral) introgressed ancestry on the shape of the SFS would be to allow migrants to interbreed with the recipient population for e.g. 100 generations in the low migration scenarios.Reference:Schumer, M., Xu, C., Powell, D. L., Durvasula, A., Skov, L., Holland, C., … and Przeworski, M. (2018). Natural selection interacts with recombination to shape the evolution of hybrid genomes. Science, 360(6389), 656-660.

We are using the unfolded site frequency spectrum of biallelic sites polarized as ancestral or derived states by reference to variation at four outgroups one at a time. Ancestral introgression only matters for our purposes if it affects polarization af sites that pass filtering. The analysis that we now have added using a 100% consensus of three outgroups, Gch, Gma, and Bsa, shows that the upswing at the right tail (the *n* − 1 and the *n* − 2 classes) is not as high as when using any single outgroup at a time. Thus, the site frequency of the *n* − 1 class is about a logit of −4 to −4.7 for the single outgroup vs. a logit of about −5.8 for the 100% consensus outgroup case, the latter is also very similar to the DS expectation for those two site-frequency classes. These new results, therefore, indicate that there may be some effects of SNP misorientation and introgressed ancestry but not enough to invalidate fit to the DS model. R#1 made statements to that effect also.

4. Impact of variant misorientation on the SFSFour different species of cod taken from Árnason and Halldórsdóttir 2019 were used as outgroups to perform variant polarization in the Atlantic cod. As I understand it, each species was taken separately as an outgroup, thus generating four possible tables of variant orientation and therefore 4 possible SFS.Observed SFS of the non-inversion chromosomes that were polarized using three different outgroups (Arctic cod Bsa, walleye pollock Gch, and Pacific cod Gma) are presented in Suppl. Figure 11, and show slightly more high-frequency derived variants using Bsa, compared to Gma and even more compared to Gch. Of these three species, the Arctic cod Bsa is the most divergent from the Atlantic cod, and therefore seems to be the best available outgroup to perform variant orientation while minimizing orientation errors due to shared variation. However, the Pacific cod Gma was the chosen outgroup for most analyses. Why did you choose this species and is there a risk of shared ancestry due to ancient admixture between the Atlantic and the Pacific cod that could affect SNP orientation?

We used the four outgroups that we have to investigate issues of ancestral states raised for example by Keightley and Jackson (2018). We did not, however, use parsimony or their method but instead used the outgroups singly to provide independent estimates of the SFS. We now add the 100% consensus of three outgroups, Gma, Gch, and Bsa (leaving out Gog because of its close relationship to Gma), which we consider adequately addresses the issue of derived and ancestral states for this study. We chose Pacific cod (Gma) as a primary outgroup because it is ecologically most similar to Atlantic cod and the two have disjunct distribution (Pacific vs. Atlantic).

I wondered if the somewhat symmetric distribution of the residuals, which is W-shaped with a more pronounced departure on the right tail of the SFS, could indicate a problem of SNP misorientation. Indeed, observed data show an excess of high-frequency derived alleles compared to the best-fit DS model, which might be the consequence of SNP misorientation.One possibility to test this hypothesis could be to include a parameter for the rate of variant misorientation when fitting the coalescent models and estimate its value as for other parameters. This is classically done in historical demographic inference based on the SFS.Alternatively, you could more easily provide a summary of ancestral state consistency, or maximum parsimony across the cod species tree, using the results of variant polarization independently performed with the four different outgroups. If this comparison reveals a very low frequency of non-parsimonious allelic states across the species tree, then variant misorientation is unlikely to be a problem.

As we detailed above we have done additional analysis of truncated SFS, of transversions only, and by polarising the SFS using a consensus of the three outgroups Gch, Gma, and Bsa. We prefer that to the parsimony approach. These analysis suggest that, yes, SNP misorientation and ancestral introgression may have had some influence but not to the extent that it invalidates our thesis.

References

Aeschbacher S, Selby JP, Willis JH, Coop G. Population-Genomic Inference of the Strength and Timing of Selection Against Gene Flow. Proceedings of the National Academy of Sciences. 2017 Jun; 114(27):7061–7066. https://doi.org/10.1073/pnas.1616755114, doi: 10.1073/pnas.1616755114.

Andolfatto P. Adaptive Evolution of Non-coding DNA in *Drosophila*. Nature. 2005 Oct; 437(7062):1149–1152. https://doi.org/10.1038/nature04107, doi: 10.1038/nature04107.

Carr SM, Marshall HD. Intraspecific Phylogeographic Genomics From Multiple Complete mtDNA Genomes in Atlantic Cod (Gadus morhua): Origins of the “Codmother, ” Transatlantic Vicariance and Midglacial Population Expansion. Genetics. 2008 Sep; 180(1):381–389. https://doi.org/10.1534/genetics.108.089730, doi: 10.1534/genetics.108.089730.

Coulson MW, Marshall HD, Pepin P, Carr SM. Mitochondrial genomics of gadine fishes: implications for taxonomy and biogeographic origins from whole-genome data sets. Genome. 2006; 49(9):1115–1130. http://dx.doi.org/10.1139/g06-083, doi: 10.1139/g06-083.

Eyre-Walker A. The Genomic Rate of Adaptive Evolution. Trends in Ecology and Evolution. 2006 Oct; 21(10):569– 575. https://doi.org/10.1016/j.tree.2006.06.015, doi: 10.1016/j.tree.2006.06.015.

FayJC, Wyckoff GJ, Wu CI. Testing the Neutral Theory of Molecular Evolution with Genomic Data from *Drosophila*. Nature. 2002 Feb; 415(6875):1024–1026. https://doi.org/10.1038/4151024a, doi: 10.1038/4151024a.

Keightley PD, Jackson BC. Inferring the Probability of the Derived vs. the Ancestral Allelic State at a Polymorphic Site. Genetics. 2018 May; p. genetics.301120.2018. https://doi.org/10.1534/genetics.118.301120, doi:

10.1534/genetics.118.301120.

Kimura M, Maruyama T. The Substitutional Load in a Finite Population. Heredity. 1969 Feb; 24(1):101–114. https://doi.org/10.1038/hdy.1969.10, doi: 10.1038/hdy.1969.10.

Lewontin RC. Evolution and the Theory of Games. Journal of Theoretical Biology. 1961 Jul; 1(3):382–403. https://doi.org/10.1016/0022-5193(61)90038-8, doi: 10.1016/0022-5193(61)90038-8.

Meisner J, Albrechtsen A. Inferring Population Structure and Admixture Proportions in Low Depth NGS Data. Genetics. 2018; http://www.genetics.org/content/early/2018/08/21/genetics.118.301336, doi: 10.1534/genetics.118.301336.

MoranPAP. Random Processes in Genetics. Mathematical Proceedings of the Cambridge Philosophical Society. 1958 Jan; 54(1):60–71. https://doi.org/10.1017/s0305004100033193, doi: 10.1017/s0305004100033193.

Schweinsberg J. Coalescent Processes Obtained from Supercritical Galton–Watson processes. Stochastic Processes and their Applications. 2003 Jul; 106(1):107–139. https://doi.org/10.1016/s0304-4149(03)00028-0, doi:

10.1016/s0304-4149(03)00028-0.

Smith NGC, Eyre-Walker A. Adaptive Protein Evolution in *Drosophila*. Nature. 2002 Feb; 415(6875):1022–1024. https://doi.org/10.1038/4151022a, doi: 10.1038/4151022a.

StrotzLC, Simões M, Girard MG, Breitkreuz L, Kimmig J, Lieberman BS. Getting Somewhere With the Red Queen: chasing a biologically modern definition of the hypothesis. Biology Letters. 2018 May; 14(5):20170734. https: //doi.org/10.1098/rsbl.2017.0734, doi: 10.1098/rsbl.2017.0734.

Van Valen L. A New Evolutionary Law. Evolutionary Theory. 1973; 1:1–30.

Vermeij GJ. Anatomy of an Invasion: the Trans-Arctic Interchange. Paleobiology. 1991; 17:281–307.

Vermeij GJ, Roopnarine PD. The Coming Arctic Invasion. Science. 2008; 321:780–781. http://doi.org/10.1126/ science.1160852, doi: 10.1126/science.1160852.